# Toward Understanding In-context vs. In-weight Learning

**Bryan Chan**[1*]**, Xinyi Chen**[2*]**, András György**[2]**, Dale Schuurmans**[1,2]
[1]University of Alberta     [2]Google DeepMind
bryan.chan@ualberta.ca    {xinyic,agyorgy,schuurmans}@google.com

## Abstract

It has recently been demonstrated empirically that in-context learning emerges in transformers when certain distributional properties are present in the training data, but this ability can also diminish upon further training. We provide a new theoretical understanding of these phenomena by identifying simplified distributional properties that give rise to the emergence and eventual disappearance of in-context learning. We do so by first analyzing a simplified model that uses a gating mechanism to choose between an in-weight and an in-context predictor. Through a combination of a generalization error and regret analysis we identify conditions where in-context and in-weight learning emerge. These theoretical findings are then corroborated experimentally by comparing the behaviour of a full transformer on the simplified distributions to that of the stylized model, demonstrating aligned results. We then extend the study to a full large language model, showing how fine-tuning on various collections of natural language prompts can elicit similar in-context and in-weight learning behaviour.

## 1 Introduction

In-context learning (ICL) is an interesting and useful property exhibited by large language models (LLMs), wherein a model is able to successfully learn and generalize from new information given in its input, without ever having seen related data during training. Radford et al. (2019) and Brown et al. (2020) were among the first to demonstrate the in-context learning capability of large-language models. Since then, numerous empirical studies have attempted to understand the emergence of ICL in LLMs (Olsson et al., 2022; Wei et al., 2023; Agarwal et al., 2024; Shi et al., 2024). However, ICL is not limited to natural language processing (NLP), and many synthetic settings have been constructed to better understand this phenomenon (Garg et al., 2022; Akyürek et al., 2023; Von Oswald et al., 2023; Gupta et al., 2023; Edelman et al., 2024; Wu et al., 2024; Zhang et al., 2024).

A recent line of work (Chan et al., 2022a; Reddy, 2024; Singh et al., 2023) has explored the emergence of ICL through the lens of data-distributional properties. In particular, Chan et al. (2022a) first proposed the importance of common and rare classes as driving in-context learning and in-weight learning (IWL). Both Chan et al. (2022a) and Reddy (2024) empirically identified that ICL can emerge in a transformer model (Vaswani et al., 2017) when there is a large within-class variation and a large number of classes. Chan et al. (2022a;b) further observed that both ICL and IWL can emerge simultaneously when the distribution over classes is Zipfian, while Singh et al. (2023); Reddy (2024); Panwar et al. (2024) subsequently noticed that ICL can become transient in an asymptotic training regime.

The work in this paper is inspired by a similar distributional perspective, but we investigate in greater depth how the distributional properties of the data affect the ability of a transformer model to learn and implement in-weight (IW) and in-context (IC) predictors over different parts of the input space. In particular, we provide insights on the emergence of ICL in synthetic settings with theory and experiments, and bridge the gap to practice by demonstrating that similar phenomena continue to hold in a simple NLP task with a real LLM, Gemini Nano 1 (Gemini Team, Google, 2023).

**Contributions.**   We present a simple theoretical model in which ICL both emerges and becomes transient in the infinite data regime, demonstrating that very simple properties of the training distribution can lead to these phenomena. In particular, following the common vs. rare classes distribution

---

*Equal contribution.

of Chan et al. (2022a), our model requires two types of data: (i) data where similar observations frequently appear in the training set—this type of data will induce IWL; and (ii) a large number of diverse data points, where each sample is predictable from the context, but each appears rarely so that reliable IWL cannot be achieved—this type of data will induce ICL.

Through a combination of a generalization error and regret bound analysis, we identify conditions where ICL and IWL emerge. In particular, our theoretical model shows that the presence of type (i) data (which we refer to as *in-weight* data) induces IWL and the presence of type (ii) data (referred to as *in-context* data) leads to ICL. This is consistent with the empirical findings of Chan et al. (2022a): When a sufficient number of in-context data samples are present, such as samples from rare classes, ICL emerges, and the Zipf distribution provides a good balance between in-context and in-weight data. When the model is trained over more samples, the samples from rare classes (or in-context data) accumulate and the model learns in-weight after observing sufficiently many samples from a given class. That is, the in-context data eventually becomes in-weight data, which improves the performance of IWL. Consequently if IW prediction is better asymptotically, IWL is eventually preferred over ICL and the latter diminishes. This partially explains the recent findings that ICL can be transient (Reddy, 2024; Panwar et al., 2024), leaving open only the question about the case, examined by Singh et al. (2023), where ICL and IWL can achieve the same performance in theory.

We demonstrate empirically that training transformers on synthetic and Omniglot (Lake et al., 2015) data drawn from the stylized distribution in our theoretical model follows the predictions of the developed theory. We further provide examples where ICL is persistent or where ICL and IWL are present in parallel when ICL and IWL can achieve the same performance, which suggests that the transience of ICL is more complicated in this case and it might depend on its finite-time performance (which is much harder to analyze experimentally). Finally, to bridge the gap from theory to practice, we show that fine-tuning an LLM (Gemini Nano v1 Gemini Team, Google, 2023) to memorize certain data can result in a reduction of its ICL ability.

**Additional related work.** We do not investigate how transformers learn circuits that can perform ICL. While such ability can evidently emerge from a vast amount of in-context training data and a sufficiently expressive architecture (function class), several works have investigated mechanistically how transformers can realize ICL (Garg et al., 2022; Olsson et al., 2022; Xie et al., 2022; Akyürek et al., 2023; Hendel et al., 2023; Abernethy et al., 2024; Singh et al., 2024). In particular, Garg et al. (2022) have demonstrated transformers can in-context learn simple regression tasks and Akyürek et al. (2023) mathematically constructed instances in which transformers can implement gradient descent and closed-form regression. Xie et al. (2022) cast ICL as implicit Bayesian inference and have shown that ICL is optimal when provided infinite examples with distinguishable prompts. Hendel et al. (2023) have empirically demonstrated that the transformer can compress context into a task vector and direct the output of the query. These works generally induce ICL by constructing multi-task context sequences. By contrast, we investigate how ICL can emerge with context sequences of only a single task. While we consider the competition of ICL and IWL capabilities of the model, Lin & Lee (2024) examines how the model chooses from different ICL predictors learnt during training in a linear regression setting.

Several works studied ICL from a theoretical perspective. Bai et al. (2023) show that transformers can implement many machine learning algorithms in-context and can perform in-context algorithm selection. They also provide excess loss guarantees for pretraining to perform ICL. Li et al. (2023) formalize ICL as an algorithm learning problem, where during pretraining the transformer learns a prediction function from the training sequences. They further provide generalization bounds from a multi-task learning perspective using algorithmic stability. Several works study the training dynamics with gradient descent or gradient flow and the emergence of ICL during training. Nichani et al. (2024) study ICL by focusing on the ability of a simplified two-layer transformer architecture to learn causal structures and the emergence of induction heads. Edelman et al. (2024); Chen et al. (2024) also study how two-layer transformers learn induction heads for Markovian input sequences, unveiling the multi-phase nature of the learning process. Focusing on single-layer architectures, Sanford et al. (2024) show that the size of a single-layer transformer should be exponentially larger to learn similar induction heads than that of a two-layer architecture, while Zhang et al. (2024) analyze the convergence properties of the training of a single linear attention layer for in-context linear regression tasks. Wu et al. (2024) further study the generalization error under a similar setting, and characterize how many tasks are needed to pretrain the model. They also show that the learned solution is competitive with the Bayes optimal predictor under certain conditions. Our theoretical

work differs in that we focus on the selection of an in-weight and an in-context predictor based on their corresponding test errors, as a model for this mechanism in transformer architectures.

## 2 THEORETICAL ANALYSIS

In this section we present our theoretical model. In essence, we consider a bi-level model that learns a "memorizing" in-weight predictor $g$ as well as an in-context predictor $h$, and for every input $\tilde{x}$ it follows the prediction of the submodel that is expected to be better for that particular data point. Naturally, $g$ can be learned with high fidelity in parts of the input space where there is enough training data, while $h$ can be effective anywhere where the context is useful. The final behavior of the model then will depend on the expected accuracy of $g$ and $h$ for a given input. In the rest of this section we formalize this approach in the setting of tabular classification problems.

### 2.1 SETTING

Let $\mathcal{X} \subset \mathbb{R}^d$ be the finite space of inputs,[1] $C$ be the number of classes, and let the space of labels be $\Delta_C$, the $C$-dimensional simplex. We have access to a training dataset $S$ of $N$ examples, where in addition to the usual (input, label) pairs in a classification problem, each example has a context consisting of $L$ extra (input, label) pairs: $S = \{((x_i^1, y_i^1, x_i^2, y_i^2, \ldots, x_i^L, y_i^L, x_i), y_i)\}_{i=1}^N$. Let $\tilde{x}_i := (x_i^1, y_i^1, x_i^2, y_i^2, \ldots, x_i^L, y_i^L, x_i)$ denote the $i$-th training example, and $\tilde{\mathcal{X}} = (\mathcal{X} \times \Delta_C)^L \times \mathcal{X}$ be the space of examples with context. Suppose $(\tilde{x}_i, y_i)$ are drawn from a ground-truth distribution $\mathcal{D}$; during training the task is to predict $y_i$ given $\tilde{x}_i$, while the final goal is to minimize the prediction error for a new sample $(\tilde{x}, y)$ sampled independently from $\mathcal{D}$ where $\tilde{x} = (x^1, y^1, x^2, y^2, \ldots, x^L, y^L, x)$.

In the latter, the prediction error is measured by the loss function $\ell : \Delta_C \times \Delta_C \to [0, 1]$, where $\ell(\hat{y}, y)$ is the loss of the prediction $\hat{y}$ given the label $y$.

**Data-generating process.**  We consider the setting of inputs with label noise: there exists a mapping $y^* : \mathcal{X} \to \Delta_C$, such that the true label for input $x$ is $y^*(x)$. When we sample $x$ in the context or as the query, we sample $y = e_i$, the $i$-th standard basis vector, with probability $y^*(x)_i$.

**Predictors.**  Below, we formalize the two classes for learning: the in-weight learning (IWL) class $\mathcal{G}$, and the in-context learning (ICL) class $\mathcal{H}$. The *IWL class* of functions $\mathcal{G}$ uses *only the query*, and is a tabular function class: for each $x \in \mathcal{X}$, $g$ can learn a separate mapping from $x$ to $\Delta_C$. We denote $\mathcal{G} = \{g : \tilde{\mathcal{X}} \to \Delta_C, g(\tilde{x}; w) = g(x; w) = w(x) \in \Delta_C, w \in \Delta_C^{|\mathcal{X}|}\}$. The *ICL class* of functions $\mathcal{H}$ uses *only labels in the context* to make a prediction, and let it be parameterized by $u$ belonging to some set $\mathcal{W}$. Inspired by the induction head (Olsson et al., 2022), a circuit that learns to copy tokens in the context to the output, we design $\mathcal{H}$ consisting of functions that output a convex combination of the labels in the context. In addition, we mix in uniform distribution to ensure bounded loss. Let $h' : \tilde{\mathcal{X}} \times \mathcal{W} \to \Delta_L$ be a function that outputs weights for a convex combination (a distribution) given an example, and let $\mathcal{H} = \{h : \tilde{\mathcal{X}} \to \Delta_C, h(\tilde{x}; u) = \epsilon \mathbf{1} + (1 - C\epsilon) \sum_{l=1}^L h'(\tilde{x}; u)_l y_l, u \in \mathcal{W}\}$.

**Simple model.**  We design a model that selects among the in-weight and in-context learners by a function $\alpha$, which can take a different value in $[0, 1]$ for each $\tilde{x} \in \tilde{\mathcal{X}}$. We analyze the following simplified model $f$ to study the emergence of in-context vs. in-weight learning: for each $\tilde{x}$, $f(\tilde{x}; \alpha, w, u) = \alpha(\tilde{x})g(\tilde{x}; w) + (1 - \alpha(\tilde{x}))h(\tilde{x}; u)$.

### 2.2 SELECTION BY TEST ERROR

Previous empirical observations show that ICL emerges when the data distribution has a long tail with many rare classes (Chan et al., 2022a). We investigate the hypothesis that ICL emerges because in-weight learning has large error on classes that have few samples in the training dataset. We first quantify the generalization error of the in-weight learner. Then, we consider a more explicit class of in-context learners and give upper and lower bounds on the errors of the IC predictor. Because of its importance in training classifiers, and LLMs in particular, we concentrate on the cross-entropy loss[2] $\mathbb{CE}(\hat{y}, y) = -\sum_{c \in C} y_c \log \hat{y}_c$; more general losses are considered in Appendix A, and proofs are also relegated to this appendix.

---

[1]The finiteness of $\mathcal{X}$ is only assumed to simplify the exposition; the results can easily be extended to a continuous input space $\mathcal{X}$ (see Appendix A.1).

[2]Throughout the paper, $\log$ denotes the natural logarithm.

We start by quantifying how fast the loss of an IW predictor $g$ can achieve that of the optimal predictor achieving $\min_{\hat{y} \in \Delta_C} \mathbb{E}_y[\ell(\hat{y}, y)]$. In case of the cross-entropy loss, the minimizer $\hat{y}$ above becomes the true distribution $y^*$ and we have $\min_{\hat{y} \in \Delta_C} \mathbb{E}_y[\ell(\hat{y}, y)] = \mathbb{E}_y[\ell(y^*(x), y)]$. Furthermore, one can show that the convergence rate for the cross-entropy loss can be much faster than the usual $O(1/\sqrt{N_x})$ rate—the optimal rate achievable by any predictor $g$ is $\mathbb{E}_y[\mathbb{CE}(g(x), y)] - \mathbb{E}_y[\mathbb{CE}(y^*, y)] \geq \frac{C-1}{2} \frac{\log(N_x)}{N_x} - O(1/N_x)$. Setting $g$ to be the famous Krichevsky–Trofimov estimator, predicting $g^{KT}(x)_i = \frac{N_{i|x} + 1/2}{N_x + C/2}$ where $N_{i|x}$ is the number of samples when we observe label $i$ for input $x$, achieves this bound (see, e.g., Csiszár & Shields, 2004, Theorem 6.4):

$$\mathbb{E}_y[\mathbb{CE}(g^{KT}(x), y)] - \mathbb{E}_y[\mathbb{CE}(y^*, y)] \leq \frac{C-1}{2} \frac{\log(N_x)}{N_x} + O(1/N_x).$$

Inspired by the induction head and attention mechanism in transformers (Olsson et al., 2022), in the rest of this subsection we consider the following function class for in-context learning, which is one of the simplest examples showcasing ICL capabilities and can be used to demonstrate basic properties that may emerge during ICL:

$$\mathcal{H} = \left\{ h(\tilde{x}) = \epsilon \mathbf{1} + (1 - C\epsilon) \sum_{l=1}^{L} \frac{\exp(-\|x_l - x\|_A)}{\sum_{j=1}^{L} \exp(-\|x_j - x\|_A)} y_l, \|A\| \leq B \right\},$$

where $\epsilon \in (0, 1)$, $A$ is a $d \times d$ positive semidefinite matrix, $\|A\|$ denotes the spectral norm of $A$, and $\|x\|_A = \sqrt{x^\top A x}$ for any $x \in \mathbb{R}^d$. The idea is that a function $h \in \mathcal{H}$ implements a simplified induction head, averaging the label predictions $y_l$ for all context vectors $x_l$ weighted by a softmax depending on the similarity of $x_l$ and the query $x$. If the context contains *irrelevant* labels, that is, labels $y_l \neq y$ for some $l \in [L]$, the prediction will be noisy, as shown in the next proposition.

**Proposition 1.** *Given an example sequence $\tilde{x}$ with one-hot label $y$, let $k$ denote the number of irrelevant labels in the context. Suppose for all $x \in \mathcal{X}$, $\|x\| \leq 1$, then the prediction of any $h \in \mathcal{H}$ satisfies*

$$\frac{2k(1 - \epsilon C)}{k + (L - k)\exp(2\sqrt{B})} + 2\epsilon(C - 1) \leq \|h(\tilde{x}) - y\|_1 \leq \frac{2k(1 - \epsilon C)}{L} + 2\epsilon(C - 1).$$

Specializing $\ell(\hat{y}, y)$ to cross-entropy loss $\mathbb{CE}(\hat{y}, y) = \sum_{c \in C} -y_c \log \hat{y}_c$, we obtain the following.

**Corollary 1.** *Assume the labels $y$ are one-hot (deterministic). Let $\tilde{x}$ be an example sequence. If $\tilde{x}$ does not contain a relevant label, then $\mathbb{CE}(h(\tilde{x}), y) = \log \frac{1}{\epsilon}$. If $\tilde{x}$ contains $k$ irrelevant labels, then*

$$\frac{k(1 - \epsilon C)}{k + (L - k)\exp(2\sqrt{B})} + \epsilon(C - 1) \leq \mathbb{CE}(h(\tilde{x}), y) \leq -\log\left((1 - \epsilon C)\frac{L - k}{L} + \epsilon\right).$$

The guarantees above show that the test error of the IW predictor converges to that of the optimal predictor at a rate of $O(1/\sqrt{N_x})$, while the IC predictor has a minimum test error depending on the number of irrelevant labels. In the case where the true labels $y^*(x)$ have low variance, and the best IW predictor achieves lower risk than the IC predictor, the IW predictor will eventually be more accurate than the IC predictor given enough samples. For example, let $B = 1$, $C = 2$. Fix $L$, for any $k \geq 1$, the minimum ICL error is bounded from below by $\frac{1 - 2\epsilon}{1 + (L-1)e^2} + \epsilon$. Under the cross-entropy loss, the minimum error for the IW predictor is the entropy of $y^*(x)$. Since $C = 2$, $y$ has a binary distribution, whose entropy is a continuous function with minimum value 0. Therefore, if $y^*$ concentrates sufficiently on one of the possible outputs, the entropy is smaller than $\frac{1 - 2\epsilon}{1 + (L-1)e^2} + \epsilon$, and under this $y^*(x)$ the minimum error of IWL is smaller than that of the ICL predictor.

We plot the bounds in Figure 1. Indeed, with few samples, the loss of the IC predictor can be smaller than that of the IW predictor. However, the IW predictor eventually converges to a solution with smaller error after seeing many examples with the same query. We can consider an oracle algorithm that selects between the learned IW predictor $\hat{g}$ and IC predictor $\hat{h}$ based on their test error (i.e., how they would perform on new data), and predicts

$$f(\tilde{x}) = \alpha(\tilde{x})\hat{g}(x) + (1 - \alpha(\tilde{x}))\hat{h}(\tilde{x}), \quad \text{where} \quad \alpha(\tilde{x}) = \mathbb{I}\left\{\mathbb{E}_y[\ell(\hat{g}(x), y)] \leq \mathbb{E}_y[\ell(\hat{h}(\tilde{x}), y)]\right\}$$

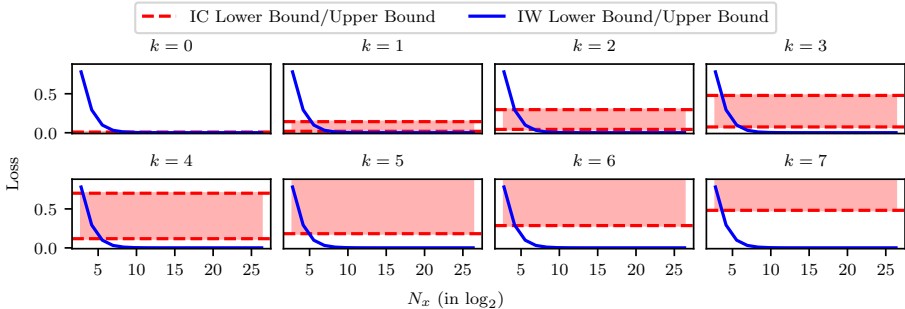

Figure 1: The theoretical error bounds of IC and IW predictors. We set $L = 8$, $\epsilon = 0.001$, $B = 1$, $C = 10$, and $y^*(x) = [0.999, 0.001/9, \ldots, 0.001/9]$. As the number of irrelevant contexts, $k$, increases the lower and upper bounds of IC error also increase, whereas where the number of samples, $N_x$, increases the lower and upper bounds of the IW error decrease. Consequently one can expect ICL to be transient as we observe more samples.

is the indicator function whether the expected test error of $\hat{g}$ does not exceed that of $\hat{h}$ for a particular input $\tilde{x}$. Under this algorithm, it is possible that initially IWL performs worse than ICL due to insufficient in-weight data. For example, when the training data contains a large number of rare classes, and each class is seen only a few times. Eventually, with enough in-weight data, the model can memorize the solution to achieve as good or even better prediction accuracy by using IWL on the query, rather than using the context for prediction and exhibiting ICL. This phenomenon is consistent with the empirical observation that ICL emerges with rare classes in the training data, and can be transient after more training steps with more data (Chan et al., 2022a; Singh et al., 2023; Reddy, 2024; Panwar et al., 2024). We defer further discussion on transience of ICL to Appendix D.

### 2.3 LEARNING TO USE IN-CONTEXT VS. IN-WEIGHT LEARNING

In practice, the model does not observe the test error during training. Here we attempt to bridge the gap between the oracle algorithm and the behavior of the simple model when $f$ is trained using gradient-based methods. Consider the usual sequential training procedure: for each sample $\tilde{x}_t, t = 1, \ldots, N$, the model predicts $f_t(\tilde{x}_t; \alpha_t) = \alpha_t(\tilde{x}_t)g(\tilde{x}_t; w_t) + (1 - \alpha_t(\tilde{x}_t))h(\tilde{x}_t; u_t)$, and the parameters $w_{t+1}, u_{t+1}, \alpha_{t+1}$ are updated based on the observed loss $\ell_t(f_t(\tilde{x}_t)) = \ell(f_t(\tilde{x}_t), y_t)$.

The parameters $w_t, u_t, \alpha_t$ are usually updated using some gradient-based method $\mathcal{A}_f$. Apart from converging to a local optimum of the loss function under technical assumptions (e.g., convexity of the loss function, see, e.g., Zinkevich, 2003; Hazan, 2023), such algorithms often guarantee that their regret grows only sublinearly in $N$, where the regret of algorithm $\mathcal{A}_f$ is defined as

$$\text{Regret}_N(\mathcal{A}_f) = \sum_{t=1}^{N} \ell(f_t(\tilde{x}_t; \alpha_t), y_t) - \min_{w,u,\alpha} \sum_{t=1}^{N} \ell(f(\tilde{x}_t; w, u, \alpha), y_t),$$

the excess loss of the online updated parameter sequence $(w_t, u_t, \alpha_t)_{t=1}^{N}$ relative to that of the optimal predictor selected in hindsight (where $f(\cdot, w, u, \alpha) = \alpha(\cdot)g(\cdot; w) + (1 - \alpha(\cdot))h(\cdot; u)$). Sublinear regret means that the average loss of the algorithm converges to the loss of the optimal predictor; algorithms with this property are called *no-regret* algorithms (as the average regret converges to zero). Note that when minimizing the regret, the parameters used to predict at time step $t$ are computed based on the observed data $(\tilde{x}_1, y_1), \ldots, (\tilde{x}_{t-1}, y_{t-1})$, and are evaluated on a new data point $(\tilde{x}_t, y_t)$; hence the choice of $\alpha_t$ is evaluated on new data, similarly to our recommendation at the end of the previous subsection, where the choice between the IW and IC predictors should be based on their performance on new, unseen data, that is, the test error.

For simplicity, we consider a bi-level parameter update procedure shown in Algorithm 1. Since $g$ is tabular, it can implement the optimal fixed predictor for each input $x$, and hence in the regret definition we can use $g(\cdot; w^*)$ (where $w^*$ is the optimal parameter for $g$). In the following proposition we show that the regret of Algorithm 1 is bounded by the regret of learning $\alpha$ and $g$. If $\mathcal{A}_\alpha$ and $\mathcal{A}_g$ are no-regret algorithms, Algorithm 1 has sublinear regret. Moreover, observe that under the linear

losses $m_t$, the best-in-hindsight $\alpha$ chooses the predictor with the lower total loss for each $\tilde{x}$. Indeed, denote $\alpha^*$ to be the best $\alpha$ in hindsight given the losses $m_t$, it has the following explicit form:

$$\alpha^*(\tilde{x}) = 1 \text{ if } \sum_{t:\tilde{x}_t=\tilde{x}} \ell_t(g(\tilde{x}_t; w_t)) - \ell_t(h(\tilde{x}_t; u_t)) < 0, \quad \alpha^*(\tilde{x}) = 0 \text{ otherwise}$$

---

**Algorithm 1** Bi-level update

---

Input: horizon $N$, no-regret learner $\mathcal{A}_\alpha$, $\mathcal{A}_g$ for $g$, and $\mathcal{A}_h$ for $h$.
**for** $t = 1$ to $N$ **do**
    Receive example $\tilde{x}_t$, predict with $f_t(\tilde{x}_t; \alpha_t) = \alpha_t(\tilde{x}_t)g(\tilde{x}_t; w_t) + (1 - \alpha_t(\tilde{x}_t))h(\tilde{x}_t; u_t)$
    Observe losses $\ell_t(f_t(\tilde{x}_t)), \ell_t(g(\tilde{x}_t; w_t)), \ell_t(h(\tilde{x}_t; u_t))$
    Update $w_{t+1} = \mathcal{A}_g(\ell_1(g(\tilde{x}_t; w_1)), \ldots, \ell_t(g(\tilde{x}_t; w_t)))$
    Update $u_{t+1} = \mathcal{A}_h(\ell_1(h(\tilde{x}_t; u_1)), \ldots, \ell_t(h(\tilde{x}_t; u_t)))$
    Define $m_t(\alpha_t) = \alpha_t(\tilde{x}_t)(\ell_t(g(\tilde{x}_t; w_t)) - \ell_t(h(\tilde{x}_t; u_t)))$
    Update $\alpha_{t+1} = \mathcal{A}_\alpha(m_1, \ldots, m_t)$
**end for**

---

**Proposition 2.** *Let $x_t$ denote the query at time t, let $Regret_N(\mathcal{A}_g) = \sum_{t=1}^{N} \ell(g(x_t; w_t), y_t) - \sum_{t=1}^{N} \ell(g(x_t; w^*), y_t)$ be the regret of learning g, and $Regret_N(\mathcal{A}_\alpha) = \sum_{t=1}^{N} m_t(\alpha_t) - m_t(\alpha^*)$ be the regret of learning $\alpha$. Algorithm 1 satisfies*

$$\sum_{t=1}^{N} \ell(f_t(\tilde{x}_t; \alpha_t), y_t) - \sum_{t=1}^{N} \ell(g(\tilde{x}_t; w^*), y_t) \leq Regret_N(\mathcal{A}_\alpha) + Regret_N(\mathcal{A}_g).$$

Since the problem of learning $\alpha^*$ is linear, common gradient-based algorithms have sublinear regret. For example, both the exponentiated gradient and the AdaGrad method with a diagonal pre-conditioner achieves $O(\sqrt{KN})$ regret, where $K$ is the number of distinct examples seen during the learning process (see, e.g., Cesa-Bianchi & Lugosi, 2006; Hazan, 2023). Further, since $g$ learns a separate (independent) predictor for each $x$, and $\ell$ is convex, the problem of learning $g(x)$ is convex. Many regret-minimizing algorithms, including gradient descent, has $O(\sqrt{K'N})$ regret in this setting, where $K'$ is the number of distinct queries observed. In this case, we have overall sublinear regret against the best in-weight learner $g^*$. This guarantee implies that on average, our loss converges to that of the best in-weight predictor in hindsight. In Appendix A.2, we extend this result to show that Algorithm 1 converges to the best-in-hindsight predictor generally.

Suppose the examples $\{(\tilde{x}_t, y_t)\}_{t=1}^{N}$ are drawn i.i.d. from the population distribution. Then by online-to-batch conversion (Cesa-Bianchi et al., 2001), the expectation of the average loss of $g(\cdot; w_t)$ in Algorithm 1 exhibits similar behavior as the generalization-error guarantee. We give a standard result below. If we consider the same class of ICL predictors, they also will have a positive minimum loss for each $\ell_t$ as in the previous section.

**Proposition 3.** *Fix $x \in \mathcal{X}$. Suppose $\{(\tilde{x}_t, y_t)\}_{t=1}^{N}$ are drawn i.i.d. from $\mathcal{D}$, where $y$ is drawn i.i.d. given the query $x$. Let $N_x$ be the number of examples with $x$ as the query, then*

$$\frac{1}{N_x} \mathbb{E}_{y_t:x_t=x} \left[ \sum_{t:x_t=x} \ell_t(g(x; w_t)) \right] \leq \min_w \mathbb{E}_y[\ell(g(x; w), y)] + \frac{1}{N_x} \mathbb{E} \left[ Regret_{N_x}(\mathcal{A}_g(x)) \right],$$

*where $Regret_{N_x}(\mathcal{A}_g(x))$ is the regret of $\mathcal{A}_g$ on the input x, learned over $N_x$ time steps.*

In other words, although the model in practice does not directly observe the test error during training, the training samples observed at each training iteration can be viewed as a "test" sample, its loss provides a quantity similar to that of the test error.

The above results show that as long as $\mathcal{A}_g$ and $\mathcal{A}_\alpha$ are no-regret algorithms (which are satisfied in this setting by simple gradient descent, see, e.g., Zinkevich, 2003), the performance of the final predictor approximates well the performance of the optimal predictor, and $\alpha_N$ approximates well $\alpha^*$, implying that the learned model will select IC on IW prediction depending on their performance. This shows that the unrealistic step of choosing between an IC or IW predictor based on their (unknown) test error, as suggested at the end of Section 2.2, can be replaced by a decision rule learned during training using an algorithm similar to gradient descent. Note that we have not included the complexity of learning the IC predictor $h$. Nevertheless, we demonstrate experimentally in the next section that the predictions drawn from our simple models are valid in many practical scenarios.

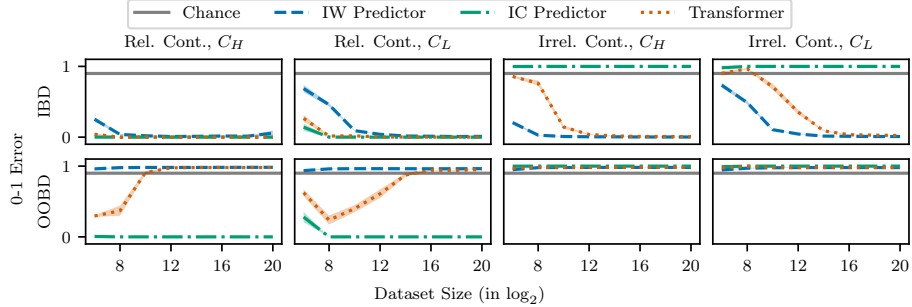

Figure 2: 0-1 validation errors of the IC predictor, IW predictor, and transformer as a function of training set size $N$ on the synthetic data, over five seeds. $L = 1$, $p_{high} = 0.9$, $p_{relevant} = 0.9$ and $\sigma = 0.2$. The columns correspond to test data with relevant/irrelevant context, and classes from the high-frequency (denoted $C_H$) or low-frequency classes (denoted $C_L$). The top row shows IBD error on the specified conditional data distribution, while the bottom row shows OOBD error. ICL diminishes as $N$ increases and IWL and ICL can emerge simultaneously.

## 3    EXPERIMENTS

In this section we demonstrate through experiments that our simple theoretical model is predictive and can help explain qualitatively the relationship between ICL and IWL in practical settings.

### 3.1    SYNTHETIC CLASSIFICATION

We consider a synthetic classification task and investigate how the errors of IC and IW predictors can inform whether a transformer trained end-to-end will exhibit ICL capabilities. How different parameters and other model architectures impact the emergence and transience of ICL are respectively explored in Appendices B.1 and B.2.

**Data.**    We consider a classification problem with an imbalanced data distribution, where there are high- and low-frequency (i.e., common and rare) classes. Let $C_H$ and $C_L$ denote their label sets, respectively, and let $C = C_H \cup C_L$. Each sample is generated by first selecting a target query-response pair $(x, y)$ then generating the accompanying context sequence. An input-output pair $(x, y)$ is sampled from a joint distribution $p(x, y) = p(x|y)p(y)$, which we refer to as the *base distribution*. First we sample a label from the label distribution $p(y)$ which is a mixture of uniform distributions over the high- and low-frequency classes $C_H$ and $C_L$, with weights $p_{high}$ and $p_{low} = 1 - p_{high}$: that is, $p(y) = p_{high}/|C_H|$ if $y \in C_H$ and $p(y) = p_{low}/|C_L|$ if $y \in C_L$; we assume that $p_{high} \gg p_{low} = 1 - p_{high}$. The distribution of queries for class $y \in C$ is parametrized by a prototype vector $x_y$ from the $d$-dimensional unit sphere; for each $y \in C$, $x_y$ is sampled uniformly from the unit sphere and is a fixed parameter of the distribution $p(x|y)$. Then $x$ is obtained by normalizing a perturbed version of the corresponding prototype vector $x_y$, where the perturbation is Gaussian; formally, $x = (x_y + \varepsilon)/\|x_y + \varepsilon\|_2$, where $\varepsilon \sim \mathcal{N}(0, \Sigma)$. To sample the context to a query-target pair $(x, y)$, first with probability $p_{relevant}$ we decide if the context is *relevant* to $(x, y)$, that is, if it should contain at least one query-target pair from the same class $y$. Then the context is obtained by sampling $L$ input-output pairs from $p$ independently under the condition that the resulting $L$-tuple contains a class-$y$ sample if and only if the context should be relevant.

**Experimental setup.**    We conduct experiments by training a transformer (GPT) end-to-end (Radford et al., 2018). The models consist of two transformer decoder layers, each with a single attention head and processes 64-dimensional embeddings. Both the input and output tokenizers are linear projections, where the former is the identity matrix. For prediction we take the last token output from the last transformer block and feed it into a linear layer followed by a softmax. To probe the difficulty of IWL and ICL, we separately train two transformers for these two settings by using data with $p_{relevant} = 0.0$ and $p_{relevant} = 1.0$; we refer to these models as the *IW and IC predictors*, respectively. These transformers only act as proxies for measuring whether it is possible to perform ICL and IWL in the idealized case. A generic model trained on data with $p_{relevant} = 0.9$ is referred to as the *transformer*. Unless specified otherwise, all models are trained using cross-entropy loss for 50K gradient updates with batch size of 32. We set $C_H = \{0, \dots, 4\}$ and $C_L = \{5, \dots, 9\}$, and set the input dimension $d = 64$ with $\Sigma = \sigma^2 I_d$ where $\sigma = 0.2$ and $I_d$ is the $d \times d$ identity matrix. Furthermore, we fix the context length $L = 1$, the probability of sampling high-frequency

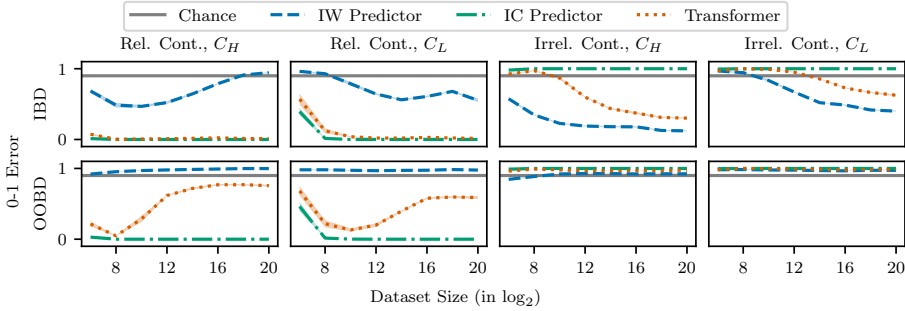

(a) $L = 1$, $p_{high} = 0.9$, $p_{relevant} = 0.9$ and $\sigma = 0.4$. ICL is no longer transient as $N$ increases when the input noise is sufficiently high.

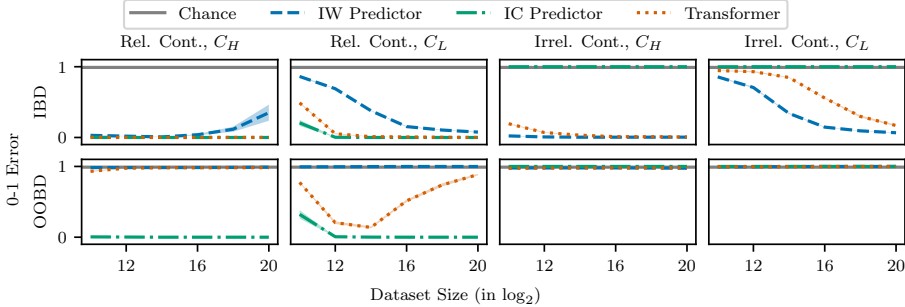

(b) $L = 1$, $p_{high} = 0.9$, $p_{relevant} = 0.9$ and $|C_L| = 95$. ICL persists for larger $N$ when compared to $|C_L| = 5$ (Figure 2) but remains transient with sufficiently large $N$.

Figure 3: 0-1 validation errors of the IC predictor, IW predictor, and transformer as a function of training set size $N$ on the synthetic data, over five seeds.

classes $p_{high} = 0.9$, and vary the number of total samples $N = \{2^6, 2^8, \ldots, 2^{20}\}$. We repeat each experiment five times, and in the figures we show confidence intervals of width 1 standard deviation.

**Evaluation.** To evaluate whether the trained models exhibit IWL or ICL, we evaluate the trained models on *in-base distribution* (IBD) and *out-of-base distribution* (OOBD). The OOBD data is obtained from the base distribution data by cyclically shifting the labels of the high-frequency and low-frequency classes respectively. Since the labels in the IBD and OOBD are different for each query, any purely IW predictor trained on IBD will fail on OOBD data, while the latter can be predicted IC if the context is relevant. A model that exhibits purely IWL will result in low IBD error regardless of context relevance, but will result in high OOBD error. A model that exhibits purely ICL will result in low error on relevant contexts, but will result in high error on irrelevant contexts.

**Main results.** Our first experiments aim to identify conditions in which ICL emerges and/or is transient. Figure 2 shows that slight input perturbation can elicit ICL capabilities from the transformer. As the dataset size $N$ increases, the IW predictor achieves similar IBD error as the IC predictor, which correlates with the transformer losing the ICL capabilities. We further observe that ICL diminishes quicker for high-frequency classes compared to low-frequency classes, as the IW predictor requires more samples to achieve near-zero error for $C_L$. Consequently there is a phase in which the transformer can perform ICL and IWL simultaneously.

Modifying the input noise can dictate whether ICL emerges and whether ICL persists throughout training, as observed by Chan et al. (2022a;b). Consistently, our theory suggests that adding more noise results in a harder learning problem for the IW predictor, and hence we would expect ICL to emerge and be more persistent in the presence of higher noise level. In agreement with this, Figure 3a shows that ICL is never transient for $\sigma = 0.4$; as predicted by our theory, when there is little to no noise, IWL is (nearly) perfect whereas ICL is incorrect with irrelevant context, while at higher noise levels the transformer can no longer rely on IWL and can only perform well with ICL. More results on varying noise level can be found in Figure 7 in Appendix B.1.1.

**Varying distributional parameters.** We further vary the parameters of the data-generating distribution and study the resulting ICL performance in Appendix B.1. Notably, increasing the number of

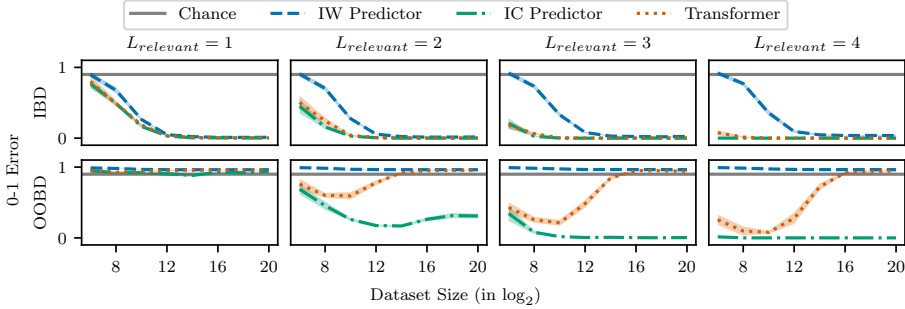

Figure 4: 0-1 validation errors, conditioned on relevant contexts and low-frequency classes $C_L$, of the IC predictor, IW predictor, and transformer as a function of training set size $N$ on the synthetic data, over five seeds. $L = 4$, $p_{high} = 0.9$, $p_{relevant} = 0.9$, and $L_{relevant} = \{1, 2, 3, 4\}$. The transformer exhibits stronger ICL as the number of relevant contexts $L_{relevant}$ increases.

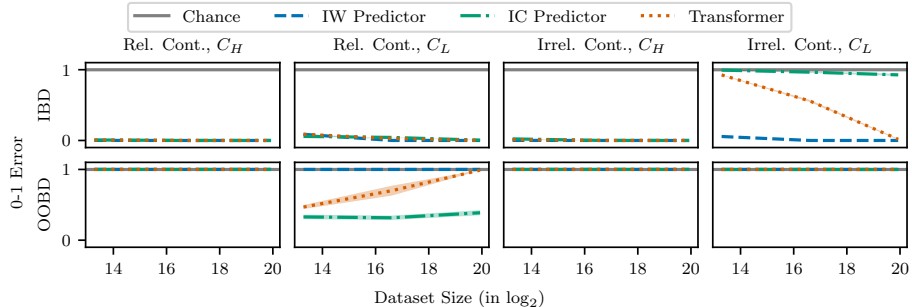

Figure 5: 0-1 validation errors as a function of the dataset size $N$ on Omniglot, over three seeds. $L = 2$, $p_{high} = 0.9$, $p_{relevant} = 0.9$, and $\sigma = 0.0$. The transformer exhibits ICL on low-frequency classes $C_L$ initially but loses said capability as $N$ increases.

low-frequency classes ($|C_L|$) makes it harder to get ICL off the ground (possibly due to the increased complexity to learn the mapping from class embeddings to class labels), but ICL diminishes slower, as it is harder to learn an IW predictor for more classes (Figure 3b).

Based on our theoretical analysis ICL with longer contexts is more difficult to emerge as there can be more distractor contexts, making it harder to identify the relevant context example(s). To test this we fix $L = 4$ and vary the number of relevant contexts $L_{relevant}$ provided to the transformer. Figure 4 shows that when we provide the transformer with a sufficiently small number of relevant contexts it no longer learns to perform ICL, suggesting that ICL is more difficult to learn with less relevant contexts. While transformers might implement ICL differently from our theoretical IC predictor, these findings are consistent with our theoretical analysis.

## 3.2 OMNIGLOT

We now investigate whether our findings in the previous section apply to learning on a natural few-shot learning dataset, Omniglot (Lake et al., 2015), containing images of handwritten characters of different languages. The model is the same as in the previous subsection, except the input tokenizer is a ResNet (He et al., 2016) that maps images to a 64-dimensional embedding. Our data construction follows Section 3.1 and is different from Chan et al. (2022a) and Singh et al. (2023). Specifically, Chan et al. (2022a) set the context length $L = 8$ and constructed context sequences such that the relevant contexts included three relevant and three irrelevant examples from another class. Chan et al. (2022a) also modelled the base-level distribution to be Zipfian. On the other hand, we set the context length to $L = 2$ with a varying number of relevant contexts and input noise $\sigma = 0.1$. The base-level distribution is again a mixture of uniform distributions over high- ($C_H$) and low-frequency classes ($C_L$), respectively. We choose the first 20 classes to be $C_H$ and the remaining 1603 classes to be $C_L$. We vary the number of samples $N = \{10^4, 10^5, 10^6\}$ and train all models for 100k gradient steps. We repeat each experiment for three random seeds. We vary the distributional parameters similar to Section 3.1 and present the complete results in Appendix B.4.

We have found that generally the findings are consistent with that of Section 3.1. However having no input noise appears to not play an important role in this case. Specifically, the transformer trained on Omniglot first exhibits ICL on $C_L$ and ICL becomes transient as $N$ increases even without any input noise (Figure 5), in contrast to the finding in the synthetic experiment. However, this result does not contradict the synthetic experiment—previously we have also shown in the synthetic experiment that with larger number of low-frequency classes the model exhibits stronger ICL—here the number of low-frequency classes is dramatically increased compared to the synthetic experiment.

### 3.3 TRANSIENCE OF ICL.

Our experiments identified several issues related to the transience of ICL. Due to space constraints, a more detailed study of this problem is relegated to Appendix D.

## 4 FINETUNING A REAL LANGUAGE MODEL

We conclude our experiments on studying the effect of IWL on the ICL abilities of a real LLM, Gemini Nano 1, with 1.8B parameters (Gemini Team, Google, 2023). During the experiment we finetuned the language model with a small number of samples, to demonstrate that memorization of these samples reduces ICL capabilities where the response given in the context is in contradiction with the memorized content (i.e., IWL happens more likely as a result of the additional training).

We finetuned the LLM to memorize where certain people live. The finetuning data, given in Table 1 in Appendix C, contains eight questions regarding where a person resides, with four real and four invented person and city names (e.g., Question: *Where does Kaitlyn live? Only give the name of the city.* Answer: *Kingston*), and is designed so that the language model must rely on IWL to learn the correlations between (name, city) pairs. Table 2 in Appendix C shows that the finetuned model has learned to correctly predict the real name-and-city pairs in the finetuning dataset (using greedy sampling), while making mistakes half of the time for the invented name-and-city pairs. In contrast, the base model, naturally, cannot answer any of the questions correctly.

To test ICL capability, before asking the finetuning prompt we also state that the person lives in another city. For example, adding that the person lives in the imaginary city *Kjheergg* (resulting in prompts like *Kaitlyn lives in Kjheergg. Where does Kaitlyn live? Only give the name of the city.*), the base model always uses ICL and answers *Kjheergg*, but the finetuned model reverts to IWL in some cases, indicating that the in-weight information can overwrite the in-context prediction present in the base model. Even when this does not happen, the probability of selecting the memorized answer increases compared to the base model, demonstrating that ICL can be transient even in real LLMs (see Table 3 in Appendix C). Additional details about these experiments are given in Appendix C.

## 5 CONCLUSION

In this paper we introduced a simple theoretical framework, motivated by induction heads (Olsson et al., 2022), where in-weight and in-context predictors are learned separately, but a gating mechanism learns to combine them given the query and the context. Intuitively, this mechanism suggests that in regions of the input space where the model is able to learn a predictor that generalizes well (i.e., it has seen enough samples to achieve low error relative to the noise and the complexity of the function to be learned), in-weight learning happens. On the other hand, in regions of the input space where data is scarce and in-weight learning is not possible, in-context learning emerges given there is enough diverse training data where the context is useful. Furthermore, training longer with more data from the latter part of the input space results in a more confident in-weight predictor, making the model shift to in-weight from in-context prediction. We demonstrated experimentally that a transformer trained on synthetic data follows similar patterns, and also showed that in-context learning can be overwritten by in-weight learning in real language models if they are made to memorize more information in their weights. While earlier work (Chan et al., 2022a) connected these phenomena to distributional properties of the data, we demonstrated that it is rather the learnability of the data (naturally affected by its distribution) which matters.

These results contribute to the understanding of when and why in-context learning is present in language models. Furthermore, they might lead to new ideas in designing training schedules for large language models; exploring this avenue is left for future work.

## ACKNOWLEDGEMENTS

We would like to thank Stephanie Chan and the anonymous reviewers for their insightful comments that helped improve the manuscript. We would also like to acknowledge the support from the Canada CIFAR AI Chairs program, the Alberta Machine Intelligence Institute (Amii), and the Natural Sciences and Engineering Research Council (NSERC) of Canada.

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

## A Proofs and additional theoretical results

We start by stating a usual generalization bound (Boucheron et al., 2012) for a learning algorithm for general convex losses (together a proof for completeness), and then present the missing proofs for Section 2.

**Proposition 4.** *Suppose we have $N$ examples in $S$ drawn independently from the data distribution $\mathcal{D}$, and let $S_x$ denote the subset of $S$ with $x$ as the query in the example. Let $|S_x| = N_x$. Let $\hat{g}$ be the empirical risk minimizer of $S$, i.e. for any $x$, $\hat{g}(x) = \arg\min_{\hat{y} \in \Delta_C} \frac{1}{N_x} \sum_{(\tilde{x},y) \in S_x} \ell(\hat{y}, y)$. Let $G_\infty \geq \|\nabla \ell(\cdot, y)\|_\infty$ be an upper bound on the infinity norm of the gradients, where $G_\infty \geq 1$. Then with probability at least $1 - \delta$, for any $x \in \mathcal{X}$, the risk of $\hat{g}(x)$ satisfies*

$$\mathbb{E}_y[\ell(\hat{g}(x), y)] \leq \min_{\hat{y} \in \Delta_C} \mathbb{E}_y[\ell(\hat{y}, y)] + 6G_\infty \sqrt{\log(2|\mathcal{X}|C/\delta)/N_x}.$$

*Proof of Proposition 4.* We show the generalization bound by online-to-batch conversion. Fix $x \in \mathcal{X}$, and consider the set of examples in $S$ that contain $x$ as the query. Let this set be $S_x$, where $|S_x| = N_x$, and in our setting the labels in $S_x$ are generated i.i.d. from the distribution $y^*(x)$. For $(\tilde{x}_i, y_i) \in S_x$, define $\ell_i(\cdot) = \ell(\cdot, y_i)$, and let the empirical minimizer be $\hat{w} = \arg\min_{w \in \Delta_C} \sum_{i=1}^{N_x} \ell_i(w)$. We will show that

$$R_x(\hat{w}) = \mathbb{E}[\ell(\hat{w}, y)] \leq \arg\min_{w \in \Delta_C} \mathbb{E}[\ell(w, y)] + 6G_\infty \sqrt{\frac{\log(2|\mathcal{X}|C/\delta)}{N_x}},$$

where the expectation is taken over $y$ drawn from $y^*(x)$. Given $\ell_1, \ldots, \ell_{N_x}$, let $w_1, \ldots, w_{N_x}$ be the output of the exponentiated gradient descent algorithm:

$$w_i = EG(\ell_1, \ldots, \ell_{N_x}).$$

Let $\bar{w} = \frac{1}{N_x} \sum_{i=1}^{N_x} w_i$ be the average of the outputs. Then with probability at least $1 - \delta$, by the online-to-batch conversion (Cesa-Bianchi et al., 2001), we have

$$R_x(\bar{w}) = \mathbb{E}\left[\ell\left(\frac{1}{N_x}\sum_{i=1}^{N_x} w_i, y\right)\right] \leq \mathbb{E}\left[\frac{1}{N_x}\sum_{i=1}^{N_x} \ell(w_i, y)\right]$$

$$\leq \arg\min_w \mathbb{E}[\ell(w, y)] + \frac{1}{N_x}\mathbb{E}\left[\text{Regret}_{N_x}(EG)\right] + 2G_\infty \sqrt{\frac{\log(2/\delta)}{N_x}}$$

The EG algorithm over the simplex has regret bounded by $2G_\infty \sqrt{2N_x \log C}$ (Hazan, 2023), and we conclude that

$$R_x(\bar{w}) \leq \arg\min_w R_x(w) + 6G_\infty \sqrt{\frac{\log(2C/\delta)}{N_x}}.$$

A uniform bound over all $x \in \mathcal{X}$ concludes the proof. $\qquad\square$

*Proof of Proposition 1.* Let $\tilde{x}$ be an example sequence, $y$ be the label, and $\mathcal{I} \subseteq [L]$ be the set of indices with irrelevant labels. Note that $|\mathcal{I}| = k$. Since $y$ is one-hot, it can be expressed as $y = e_c$ for some $c \in [C]$. We have

$$\|h(\tilde{x}) - y\|_1 = \sum_{i \neq c} h(\tilde{x})_i + 1 - h(\tilde{x})_c = 2\sum_{i \neq c} h(\tilde{x})_i$$

$$= 2(1 - \epsilon C)\sum_{l \in \mathcal{I}} \frac{\exp(-\|x - x^l\|_A)}{\sum_{i=1}^{L}\exp(-\|x - x^i\|_A)} + 2\epsilon(C - 1).$$

Note that by definition $\exp(-2\sqrt{B}) \leq \exp(-\|x - x^l\|_A) \leq 1$, since $\|x^l - x\|_A^2 = (x^l - x)^\top A(x^l - x) \leq \|x^l - x\|_2^2 \|A\|_2 \leq 4B$. Furthermore, we have $\sum_{i \notin \mathcal{I}} \exp(-\|x - x^i\|_A) = L - k$.

Since the function $\frac{x}{x+y}$ is nondecreasing in $x$ for $x > 0$ and $y \geq 0$, we conclude that

$$\frac{k}{k + (L - k)\exp(2\sqrt{B})} \leq \frac{\sum_{l \in \mathcal{I}}\exp(-\|x - x^l\|_A)}{\sum_{i \in \mathcal{I}}\exp(-\|x - x^i\|_A) + \sum_{i \notin \mathcal{I}}\exp(-\|x - x^i\|_A)} \leq \frac{k}{L}$$

The proposition follows by plugging the inequality into the previous bound. $\qquad\square$

*Proof of Corollary 1.* Let $\tilde{x}$ be an example sequence, $\mathcal{I} = \{l \in [L], y_l \neq y\}$ denote the set of labels in the context that are irrelevant, and $|\mathcal{I}| = k$. Suppose $y = e_c$ for some class $c$, we can lower bound the CE loss as follows:

$$
\mathbb{CE}(y, h(\tilde{x})) = -\log h(\tilde{x})_c = -\log\left(1 - \sum_{i \neq c} h(\tilde{x})_i\right)
$$

$$
\geq \sum_{i \neq c} h(\tilde{x})_i \qquad\qquad (-\log(1-x) \geq x)
$$

$$
\geq \frac{(1 - C\epsilon)k}{k + (L-k)\exp(2\sqrt{B})} + (C-1)\epsilon,
$$

where the last inequality follows from the previous proposition. In particular, if $k = L$, then $h(\tilde{x})_c = \epsilon$, and $\mathbb{CE}(y, h(\tilde{x})) = -\log \epsilon = \log\frac{1}{\epsilon}$. Note that for $\epsilon \in (0, 1)$, our regime of interest, $1 - \epsilon < -\log(\epsilon)$.

For the upper bound, note that the function $\frac{x}{x+y}$ is nondecreasing in $x$ for $x > 0$ and $y > 0$, and hence $h(\tilde{x})_c$ is minimized when $\sum_{i \neq c} h(\tilde{x})_i$ is maximized. We thus have

$$
\log h(\tilde{x})_c \geq \log\left((1 - \epsilon C)\frac{L-k}{L} + \epsilon\right),
$$

and the proposition follows by taking the negative on both sides. $\qquad\square$

*Proof of Proposition 2.* Define $\ell_t(\cdot) = \ell(\cdot, y_t)$, and let $g_t(\cdot)$ denote $g(\cdot; w_t)$, $h_t(\cdot) = h(\cdot; u_t)$. Let $g^*(\cdot) = g(\cdot; w^*)$ be the optimal IW predictor. We can decompose the regret of Algorithm 1 as follows:

$$
\sum_{t=1}^{N} \ell_t(f_t(\tilde{x}_t)) - \sum_{t=1}^{N} \ell_t(g^*(\tilde{x}_t))
$$

$$
\leq \sum_{t=1}^{N} \alpha_t(\tilde{x}_t)\ell_t(g_t(\tilde{x}_t)) + \sum_{t=1}^{N} (1 - \alpha_t(\tilde{x}_t))\ell_t(h_t(\tilde{x}_t)) - \sum_{t=1}^{N} \ell_t(g^*(\tilde{x}_t))
$$

$$
= \sum_{t=1}^{N} (\alpha_t(\tilde{x}_t) - \alpha^*(\tilde{x}_t))(\ell_t(g_t(\tilde{x}_t)) - \ell_t(h_t(\tilde{x}_t)))
$$

$$
+ \sum_{t=1}^{N} \alpha^*(\tilde{x}_t)\ell_t(g_t(\tilde{x}_t)) + (1 - \alpha^*(\tilde{x}_t))\ell_t(h_t(\tilde{x}_t)) - \ell_t(g^*(\tilde{x}_t))
$$

$$
\leq \text{Regret}_N(\mathcal{A}_\alpha) + \sum_{t=1}^{N} \alpha^*(\tilde{x}_t)(\ell_t(g_t(\tilde{x}_t)) - \ell_t(g^*(\tilde{x}_t))) + \sum_{t=1}^{N} (1 - \alpha^*(\tilde{x}_t))(\ell_t(h_t(\tilde{x}_t)) - \ell_t(g^*(\tilde{x}_t)))
$$

By definition, $\alpha^*(\tilde{x}) = 1$ for all $\tilde{x}$ such that $\sum_{\tilde{x}_t = \tilde{x}} \ell_t(h_t(\tilde{x}_t)) - \ell_t(g_t(\tilde{x}_t)) \geq 0$, and otherwise $\alpha^*(\tilde{x}) = 0$. Let $\mathcal{X}_{IWL}$ be the set of examples where $\alpha^*(\tilde{x}) = 1$. To bound the second sum, we have:

$$
\sum_{t=1}^{N} \alpha^*(\tilde{x}_t)(\ell_t(g_t(\tilde{x}_t)) - \ell_t(g^*(\tilde{x}_t))) = \sum_{t:\tilde{x}_t \in \mathcal{X}_{IWL}} \ell_t(g_t(\tilde{x}_t)) - \ell_t(g^*(\tilde{x}_t))
$$

Similarly, for the third sum,

$$
\sum_{t=1}^{N} (1 - \alpha^*(\tilde{x}_t))(\ell_t(h_t(\tilde{x}_t)) - \ell_t(g^*(\tilde{x}_t))) = \sum_{t:\tilde{x}_t \notin \mathcal{X}_{IWL}} \ell_t(h_t(\tilde{x}_t)) - \ell_t(g^*(\tilde{x}_t))
$$

$$
\leq \sum_{t:\tilde{x}_t \notin \mathcal{X}_{IWL}} \ell_t(g_t(\tilde{x}_t)) - \ell_t(g^*(\tilde{x}_t))
$$

Putting the three terms together, we have

$$\sum_{t=1}^{N} \ell_t(f_t(\tilde{x}_t)) - \sum_{t=1}^{N} \ell_t(g^*(\tilde{x}_t)) \leq \text{Regret}_N(\mathcal{A}_\alpha) + \sum_{t:\tilde{x}_t \in \mathcal{X}_{IWL}} \ell_t(g_t(\tilde{x}_t)) - \ell_t(g^*(\tilde{x}_t))$$

$$+ \sum_{t:\tilde{x}_t \notin \mathcal{X}_{IWL}} \ell_t(g_t(\tilde{x}_t)) - \ell_t(g^*(\tilde{x}_t))$$

$$= \text{Regret}_N(\mathcal{A}_\alpha) + \text{Regret}_N(\mathcal{A}_g)$$

$\square$

## A.1 EXTENSION TO CONTINUOUS INPUT SPACE

Similar bounds can be obtained for the continuous input space using standard results from statistical learning theory (Bousquet et al., 2003; Csiszár & Shields, 2004). Indeed, one can obtain generalization bounds for the IW predictor that decay with $O(1/\sqrt{N_x})$ (for convex losses) and $O(\log N_x / N_x)$ (for the cross-entropy loss) depend on some complexity measures of the function class, e.g., Rademacher complexity. These bounds still show that as the sample size increases, the excess risk converges to 0 in the limit. Our regret bounds in Section 2.3 automatically hold on continuous domains. However, to ensure that the regret for learning $\alpha$ can be meaningfully optimized, we need some additional assumptions (in the tabular setting this is an online learning problem with linear loss functions). A sufficient assumption is that the function $\alpha$ can be parametrized with a finite number of parameters such that the loss function is convex in these parameters and the resulting function class is rich enough to partition the input space into two sets such that IC prediction is better on one set and IW is better on the other.

## A.2 EXTENSION TO SELECTING BEST-IN-HINDSIGHT PREDICTOR

In some cases the best IW predictor is not better than the best IC predictor, and vice versa. Here we show that Algorithm 1 selects the best-in-hindsight predictor based on the observed losses.

**Proposition 5.** *Let $\tilde{x}_t$ and $x_t$ respectively denote the sequence and query at time $t$. Define the following as the regret of learning $g$, $h$, and $\alpha$ respectively:*

$$Regret_N(\mathcal{A}_g) = \sum_{t=1}^{N} \ell(g(x_t; w_t), y_t) - \sum_{t=1}^{N} \ell(g(x_t; w^*), y_t),$$

$$Regret_N(\mathcal{A}_h) = \sum_{t=1}^{N} \ell(h(\tilde{x}_t; u_t), y_t) - \sum_{t=1}^{N} \ell(h(\tilde{x}_t; u^*), y_t),$$

$$Regret_N(\mathcal{A}_\alpha) = \sum_{t=1}^{N} m_t(\alpha_t) - m_t(\alpha^*).$$

*Let $f^*(\tilde{x}_t) := f(\tilde{x}_t; w^*, u^*, \alpha^*)$ be the best-in-hindsight $f$. Suppose there are $N = N_{IWL} + N_{ICL}$ rounds with $N_{IWL}$ rounds of in-weight (IW) examples and $N_{ICL}$ rounds of in-context (IC) examples, decided based on comparing the errors of the best-in-hindsight IW and IC predictors, $\ell(g^*(x), y)$ and $\ell(h^*(\tilde{x}, y))$. Then, Algorithm 1 satisfies*

$$\sum_{t=1}^{N} \ell(f_t(\tilde{x}_t; \alpha_t), y_t) - \sum_{t=1}^{N} \ell(f^*(\tilde{x}_t), y_t) \leq Regret_N(\mathcal{A}_\alpha) + Regret_{N_{IWL}}(\mathcal{A}_g) + Regret_{N_{ICL}}(\mathcal{A}_h).$$

*Proof of Proposition 5.* Define $\ell_t(\cdot) = \ell(\cdot, y_t)$, and let $g_t(\cdot)$ denote $g(\cdot; w_t)$, $h_t(\cdot) = h(\cdot; u_t)$. Let $g^*(\cdot) = g(\cdot; w^*)$ be the best-in-hindsight IW predictor and $h^*(\cdot) = h(\cdot; u^*)$ be the best-in-hindsight IC predictor. Notice that $\alpha^*(\tilde{x}_t) = \mathbb{I}\{\tilde{x}_t \in \mathcal{X}_{IWL}\}$ for the best-in-hindsight predictor. Then, we can

decompose the regret of Algorithm 1 as follows:

$$\sum_{t=1}^{N} \ell_t(f_t(\tilde{x}_t)) - \sum_{t=1}^{N} \ell_t(f^*(\tilde{x}_t))$$

$$\leq \sum_{t=1}^{N} \alpha_t(\tilde{x}_t)\ell_t(g_t(\tilde{x}_t)) + \sum_{t=1}^{N}(1 - \alpha_t(\tilde{x}_t))\ell_t(h_t(\tilde{x}_t)) - \sum_{t=1}^{N} \ell_t(f^*(\tilde{x}_t))$$

$$= \sum_{t=1}^{N} \alpha_t(\tilde{x}_t)\ell_t(g_t(\tilde{x}_t)) + \sum_{t=1}^{N}(1 - \alpha_t(\tilde{x}_t))\ell_t(h_t(\tilde{x}_t)) - \sum_{t:\tilde{x}_t \in \mathcal{X}_{IWL}} \ell_t(f^*(\tilde{x}_t)) - \sum_{t:\tilde{x}_t \in \mathcal{X}_{ICL}} \ell_t(f^*(\tilde{x}_t))$$

$$= \sum_{t=1}^{N} \alpha_t(\tilde{x}_t)\ell_t(g_t(\tilde{x}_t)) + \sum_{t=1}^{N}(1 - \alpha_t(\tilde{x}_t))\ell_t(h_t(\tilde{x}_t)) - \underbrace{\sum_{t=1}^{N} \alpha^*(\tilde{x}_t)\ell_t(g^*(\tilde{x}_t))}_{t:\tilde{x}_t \in \mathcal{X}_{IWL}} - \underbrace{\sum_{t=1}^{N}(1 - \alpha^*(\tilde{x}_t))\ell_t(h^*(\tilde{x}_t))}_{t:\tilde{x}_t \in \mathcal{X}_{ICL}}$$

$$= \sum_{t=1}^{N} \alpha_t(\tilde{x}_t)\ell_t(g_t(\tilde{x}_t)) + \sum_{t=1}^{N}(1 - \alpha_t(\tilde{x}_t))\ell_t(h_t(\tilde{x}_t)) - \sum_{t=1}^{N} \alpha^*(\tilde{x}_t)\ell_t(g^*(\tilde{x}_t)) - \sum_{t=1}^{N}(1 - \alpha^*(\tilde{x}_t))\ell_t(h^*(\tilde{x}_t))$$

$$\underbrace{- \sum_{t=1}^{N} \alpha^*(\tilde{x}_t)(\ell_t(g_t(\tilde{x}_t)) - \ell_t(h_t(\tilde{x}_t))) + \sum_{t=1}^{N} \alpha^*(\tilde{x}_t)(\ell_t(g_t(\tilde{x}_t)) - \ell_t(h_t(\tilde{x}_t)))}_{=0}$$

$$= \underbrace{\sum_{t=1}^{N}(\alpha_t(\tilde{x}_t) - \alpha^*(\tilde{x}_t))(\ell_t(g_t(\tilde{x}_t)) - \ell_t(h_t(\tilde{x}_t)))}_{\text{Regret}_N(\alpha)}$$

$$+ \underbrace{\sum_{t=1}^{N}(1 - \alpha^*(\tilde{x}_t))(\ell_t(h_t(\tilde{x}_t)) - \ell_t(h^*(\tilde{x}_t)))}_{\text{Regret}_{N_{ICL}}(h)} + \underbrace{\sum_{t=1}^{N} \alpha^*(\tilde{x}_t)(\ell_t(g_t(\tilde{x}_t)) - \ell_t(g^*(\tilde{x}_t)))}_{\text{Regret}_{N_{IWL}}(h)} .$$

$\square$

## B  DETAILED EXPERIMENTAL RESULTS

Our code is available here: `https://github.com/chanb/icl_vs_iwl`.

### B.1  SYNTHETIC DATASET

In this section we investigate the parameters that may impact the emergence and transience of ICL. We first investigate independently the influence of input/output noise, class balance, and class cardinality in Section B.1.1. We then analyze the impact of the frequency of seeing relevant context, the number of relevant context examples, and the context length in Section B.1.2. A schematic diagram on data-generation for training and evaluation is provided in Figure 6.

### B.1.1  PARAMETERS IN BASE-LEVEL DISTRIBUTION

The base distribution is parameterized by four quantities: input noise $\sigma$, label noise $p_{label}$, number of low-frequency classes $|C_L|$, and probability of high-frequency class $p_{high}$. We first investigate the impact of $\sigma$ and $|C_L|$, which had been previously investigated by Chan et al. (2022a) and Reddy (2024). For the former, we vary the input noise $\Sigma = \sigma^2 I_d$, where $\sigma \in \{0.0, 0.02, 0.2, 0.4\}$, and train each model for 100K gradient steps to ensure convergence. Generally, we see that ICL is a transient phenomenon as we increase $N$ except for $\sigma = 0.4$ (Figure 7). As $\sigma$ increases the IW predictor requires more samples in order to achieve near-zero IBD error, whereas the IC predictor requires little samples to achieve similar error when conditioned on relevant contexts. The mismatch in convergence speed correlates with when the transformer is able to perform IWL on $C_H$ and ICL

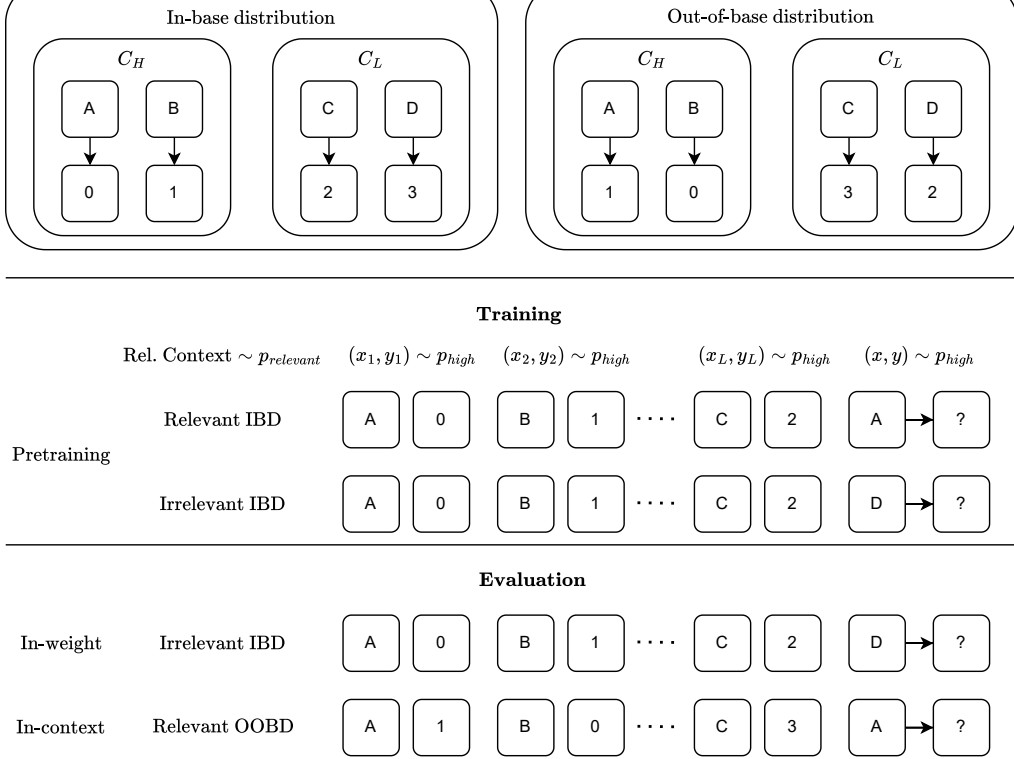

Figure 6: As an example our task maps a letter into a number. We consider two distributions—in-base distribution (IBD) and out-of-base distribution (OOBD). We generate various example sequences of length $L$ for pretraining, in-weight evaluation, and in-context evaluation. During pretraining, the model receives sequences generated by the in-base distribution. To train our transformer, we sample sequences with relevant context with probability $p_{relevant}$. To train our IW predictor and IC predictor, we use only respectively the relevant IBD sequences and the irrelevant IBD sequences. During in-weight evaluation, we consider only sequences with only irrelevant contexts, generated by the IBD. During in-context evaluation, we consider only sequences with only relevant contexts, generated by the OOBD. We further condition the evaluation to consider only high-frequency class $C_H$ and low-frequency class $C_L$.

on $C_L$ simultaneously. When $\sigma = 0.4$ the IW predictor exhibits higher IBD error than IC predictor on relevant contexts across all $N$, as a result the transformer's ICL capability is no longer transient.

For the latter experiment, we vary the number of low-frequency classes $|C_L| \in \{5, 45, 95, 495\}$, and train all models for 500k gradient steps. Similar to the findings in Chan et al. (2022a) and Reddy (2024), we observe that with increasing $|C_L|$ the transformer begins to exhibit ICL for $C_L$ while only performs IWL for $C_H$, which again indicates that a model can exhibit both ICL and IWL (Figure 8). Further observing the IBD errors of IW and IC predictors on $C_L$, we can see that the transformer transitions from ICL to IWL when IW predictor begins to exhibit lower errors than the IC predictor, aligning to our theoretical analysis.

We now investigate the impact of varying probability of sampling high-frequency classes $p_{high}$. We vary $p_{high} \in \{0.5, 0.9, 0.99\}$. The model exhibits ICL for slightly larger $N$ with balanced classes, compared to imbalanced classes (Figure 9, left). Looking at the IBD errors between IW and IC predictors, IW starts with higher error and reduces to near-zero error as $N$ increases, resulting in a better predictor than IC predictor on in-base distribution. This crossover occurs later on balanced classes as each class has effectively similar amount of samples, while $C_H$ will be observed more by the transformer on imbalanced classes. The IW predictor will converge faster on $C_H$ in the latter case and the transformer loses its ICL capability sooner.

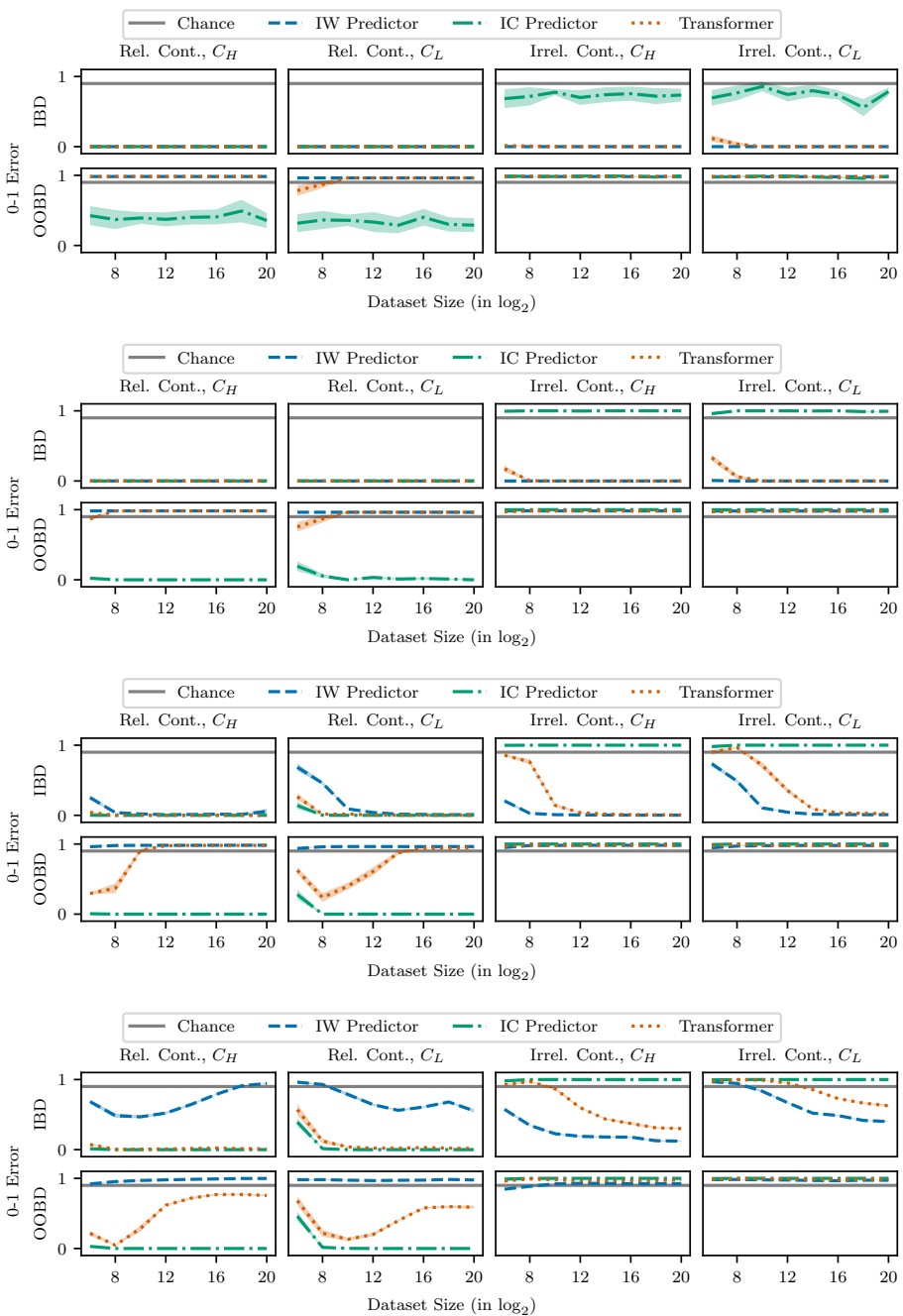

Figure 7: The validation 0-1 error of IC predictor, IW predictor, and transformer predictor as we vary input noise $\sigma \in \{0.0, 0.02, 0.2, 0.4\}$ on synthetic data, over five seeds. $L = 1, p_{high} = 0.9$, and $p_{relevant} = 0.9$ for training the transformer. For each variant, the top and bottom rows respectively correspond to IBD and OOBD errors.

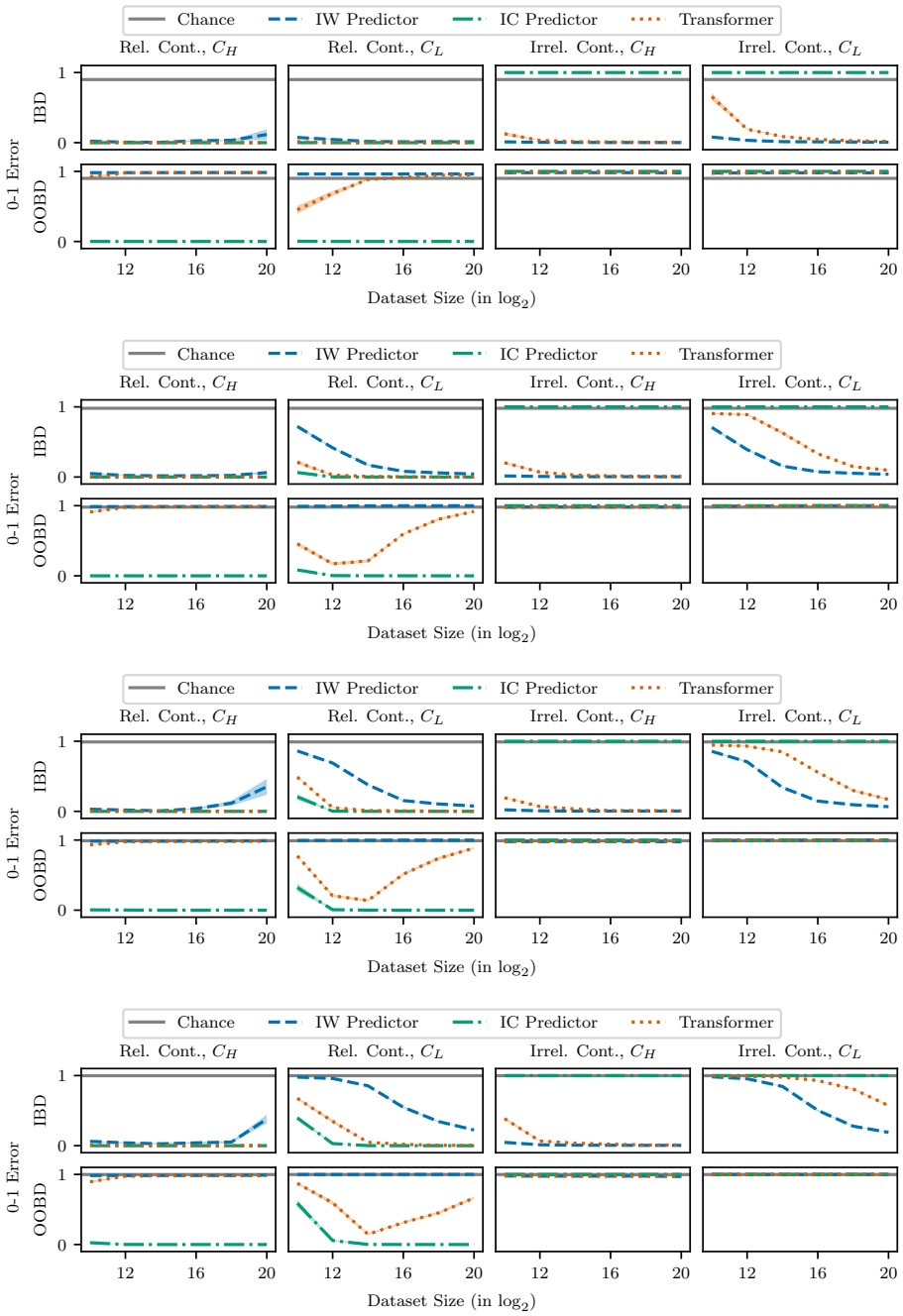

Figure 8: The validation 0-1 error of IC predictor, IW predictor, and transformer predictor as we vary the number of low-frequency classes $|C_L| \in \{5, 45, 95, 495\}$ on synthetic data, over five seeds. $L = 1, p_{high} = 0.9$, and $p_{relevant} = 0.9$ for training the transformer. For each variant, the top and bottom rows respectively correspond to IBD and OOBD errors.

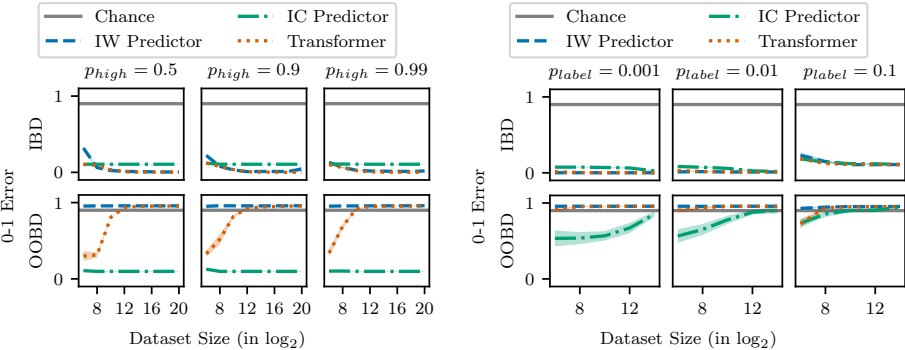

Figure 9: The validation 0-1 error of IC predictor, IW predictor, and transformer predictor as we vary the probability of high-frequency classes **(left)** and the label noise **(right)** on synthetic data, over five seeds. For each variant, the top and bottom rows respectively correspond to IBD and OOBD errors.

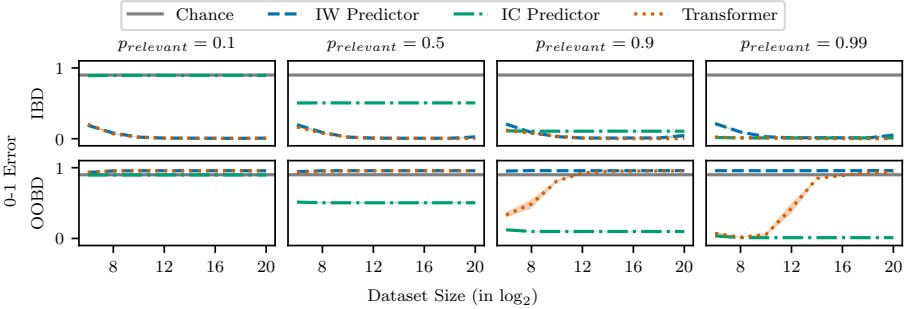

Figure 10: The validation 0-1 error of IC predictor, IW predictor, and transformer predictor as we vary the probability of sampling relevant contexts $p_{relevant}$ on synthetic data, over five seeds. For each variant, the top and bottom rows respectively correspond to IBD and OOBD errors.

Finally, we explore how label noise can impact the emergence of ICL, a setting that is not explicitly explored by existing work. We remove input noise (i.e. sampling only the prototype vector as the input) and vary the label noise $p_{label} \in \{0.001, 0.01, 0.1\}$. The noisy label is implemented such that with probability $1 - p_{label}$ we keep the original class $c$, otherwise we use $(c + 1) \mod |C|$. In this case the IW predictor achieves lower IBD error than IC predictor for small label noise, as a result the transformer exhibits minimal ICL capability (Figure 9, right). When $p_{label} = 0.1$ IC predictor is better than IW predictor. Once again our theory aligns with the experimental result whereby the transformer exhibits ICL until the errors of IC and IW predictors match.

### B.1.2 PARAMETERS IN CONTEXT-LEVEL DISTRIBUTION

We now investigate how the parameters for the context distribution can impact ICL—we consider three parameters: the probability of sampling relevant contexts $p_{relevant}$, the context length $L$, and the number of relevant contexts $L_{relevant}$. Adjusting $p_{relevant}$ can be considered as changing the probability of observing bursty contexts in Chan et al. (2022b). We vary $p_{relevant} \in \{0.1, 0.5, 0.9, 0.99\}$ in this experiment. Aligned with previous experiments and our theoretical analysis, the transformer exhibits stronger ICL when the IC predictor achieves smaller error than the IW predictor (Figure 10). As expected, when $p_{relevant}$ is small we see IC predictor to achieve high IBD error. Increasing $p_{relevant}$ results in a more accurate IC predictor, but the IW predictor can eventually outperform the IC predictor, resulting in the transient behavior of ICL in the transformer.

In NLP tasks the model is usually given multiple contexts in a sequence, some potentially act as distractors. From Figure 4 we can see that as we increase the number of contexts, even if we provide always relevant context, the IC predictor ends up exhibiting IWL capability. This suggests

the difficulty of learning ICL when variable relevant contexts are provided. We investigate in detail on whether the number of relevant examples impact the emergence of ICL. We fix $L = 4$ and vary the number of relevant examples $L_{relevant} \in \{1, 2, 3, 4\}$. When $L_{relevant} = 4$ the IC predictor achieves smaller IBD error than the IW predictor initially but the latter eventually achieves similar error with larger $N$ (Figure 12). However it is worth noting that with decreasing $L_{relevant}$ the initial IBD error of IC predictor increases, almost reaching the same error when $L_{relevant} = 1$. In that case the transformer never exhibits ICL. Indeed, as we decrease the number of relevant contexts, the transformer will have to be very precise in identifying the sole relevant context in addition to copying its label, which can be more challenging that learning just the IW predictor.

## B.2 OTHER MODEL ARCHITECTURES ON SYNTHETIC DATA

We further conduct experiments with other model architectures on the synthetic dataset. We replace the transformer in Section 3.1 with two other model architectures while keeping the training algorithm and other hyperparameters consistent: our stylized ICL model architecture defined in Section 2 and a RNN-based model.

**Stylized ICL model.**   As described in Section 2, the stylized model consists of three components: the IC predictor $h$, the IW predictor $g$, and the selector $\alpha$. Given a sequence $\tilde{x}$, the IC predictor $h(\tilde{x})$ applies a softmax over the negative $\ell_2$ distances between the query and the context inputs, followed by a dot-product with the context labels. Both the IW predictor $g$ and the selector $\alpha$ are implemented as a two-layer ReLU feedforward network with 64 hidden units. The IW predictor $g(x)$ takes only the query $x$ as the network input and outputs a probability distribution over the classes. The selector $\alpha$ takes on the sequence $\tilde{x}$ as the network input and outputs the probability of selecting $g$ over $h$, $p_{iwl} = \alpha(\tilde{x})$. Finally, the stylized model outputs a probability distribution over the classes computed by the convex combination of the distributions induced by IW and IC predictors: $p_{iwl}g(x) + (1 - p_{iwl})h(\tilde{x})$.

**RNN-based model.**   We replace the transformer blocks with a single layer of the gated recurrent unit (GRU) (Chung et al., 2015) with hidden state size of 64. Similar to the transformer, we first apply the input and output tokenizers, rollout the GRU by alternatively passing in context inputs and context labels, followed by the query. We then feed the final output of the GRU into a linear layer followed by a softmax.

**Results.**   We first consider the stylized model. Unlike the IC and IW transformers considered in Section 3, which are proxies to the stylized IC predictor $h$ and IW predictor $g$ defined in Section 2, we can directly analyze the interactions between $g$, $h$, and $\alpha$. Aligning with our theoretical findings in Section 2, when the IW predictor $g$ achieves better IBD performance against the IC predictor $h$ we can see that the selector $\alpha$ leans towards choosing $g$ (i.e., $p_{iwl} \to 1$ in Figures 13 and 14). Notably, Figure 14 shows that with increasing context length $L$, the IC predictor $h$ achieves higher IBD error, and the IW predictor $g$ can overtake $h$ with smaller dataset size $N$.

In general, it appears that the general trend on the emergence and transience of ICL with our stylized ICL model and GRU align with the findings from the transformer architecture in Section 3. With larger input noise, we see similar behaviours with our stylized model, the GRU model, and the transformer model, where ICL is no longer transient (Figures 13 and 15), suggesting that our theory can be generalized to other model architectures in addition to our stylized ICL model.

## B.3 VISUALIZING ATTENTION

We provide visualization on the attentions of the transformer trained with $N \in \{2^{10}, 2^{20}\}$ samples in Figure 16 on a 20 OOBD samples conditioned on relevant contexts. When $N = 2^{10}$ the first attention layer focuses on the context label (i.e., the bottom middle of the $3 \times 3$ grid), whereas when $N = 2^{20}$ the first attention layer focuses on the query (i.e., the bottom right of the $3 \times 3$ grid). The former predicts the OOBD labels, exhibiting ICL, while the latter predicts the IBD labels, exhibiting IWL.

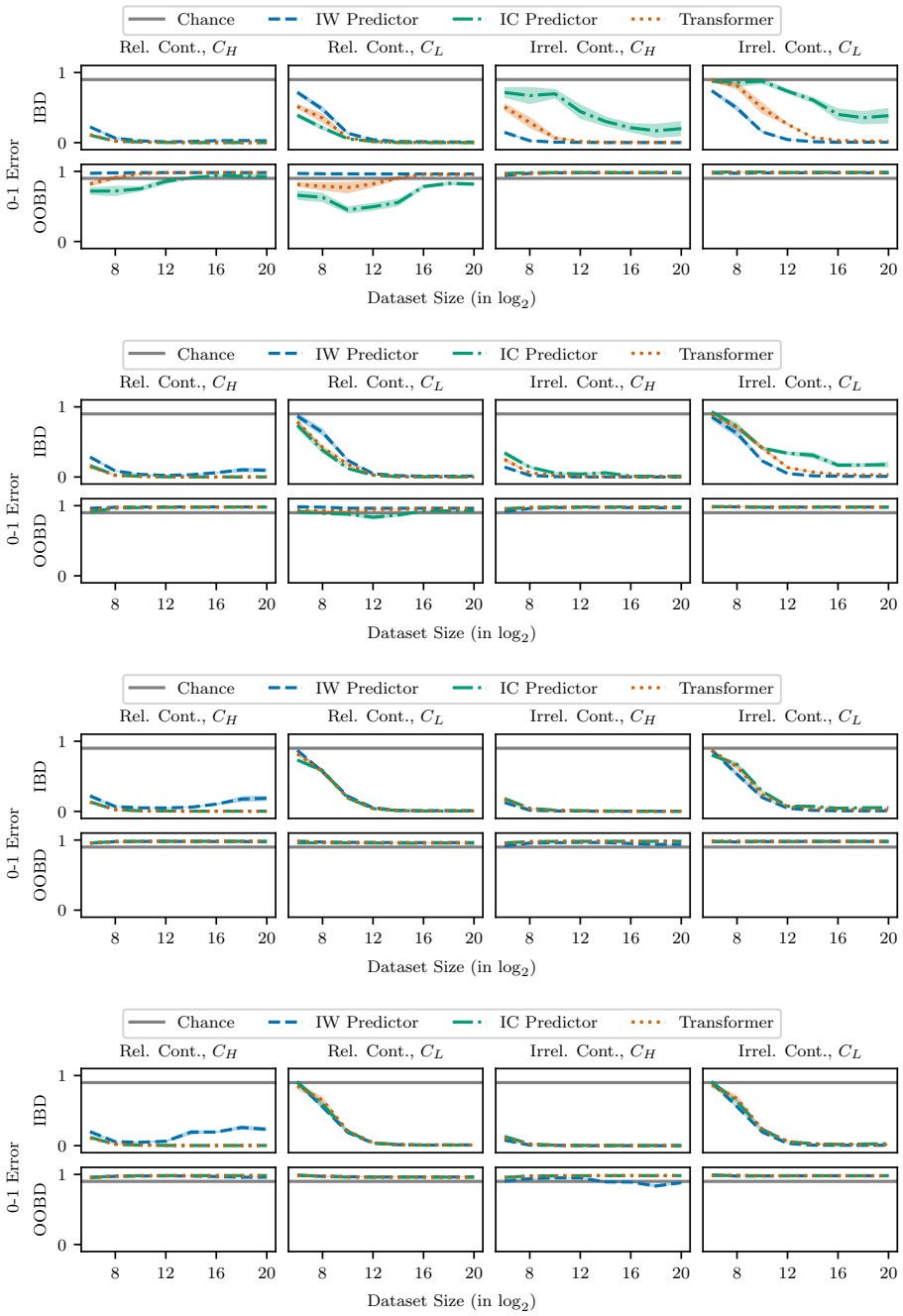

Figure 11: The validation 0-1 error of the IC predictor, the IW predictor, and the transformer predictor as we vary the context length $L \in \{2, 4, 8, 16\}$ on synthetic data, over five seeds. $L = 4, p_{high} = 0.9$, and $p_{relevant} = 0.9$ for training the transformer. For each variant, the top and bottom rows respectively correspond to IBD and OOBD errors.

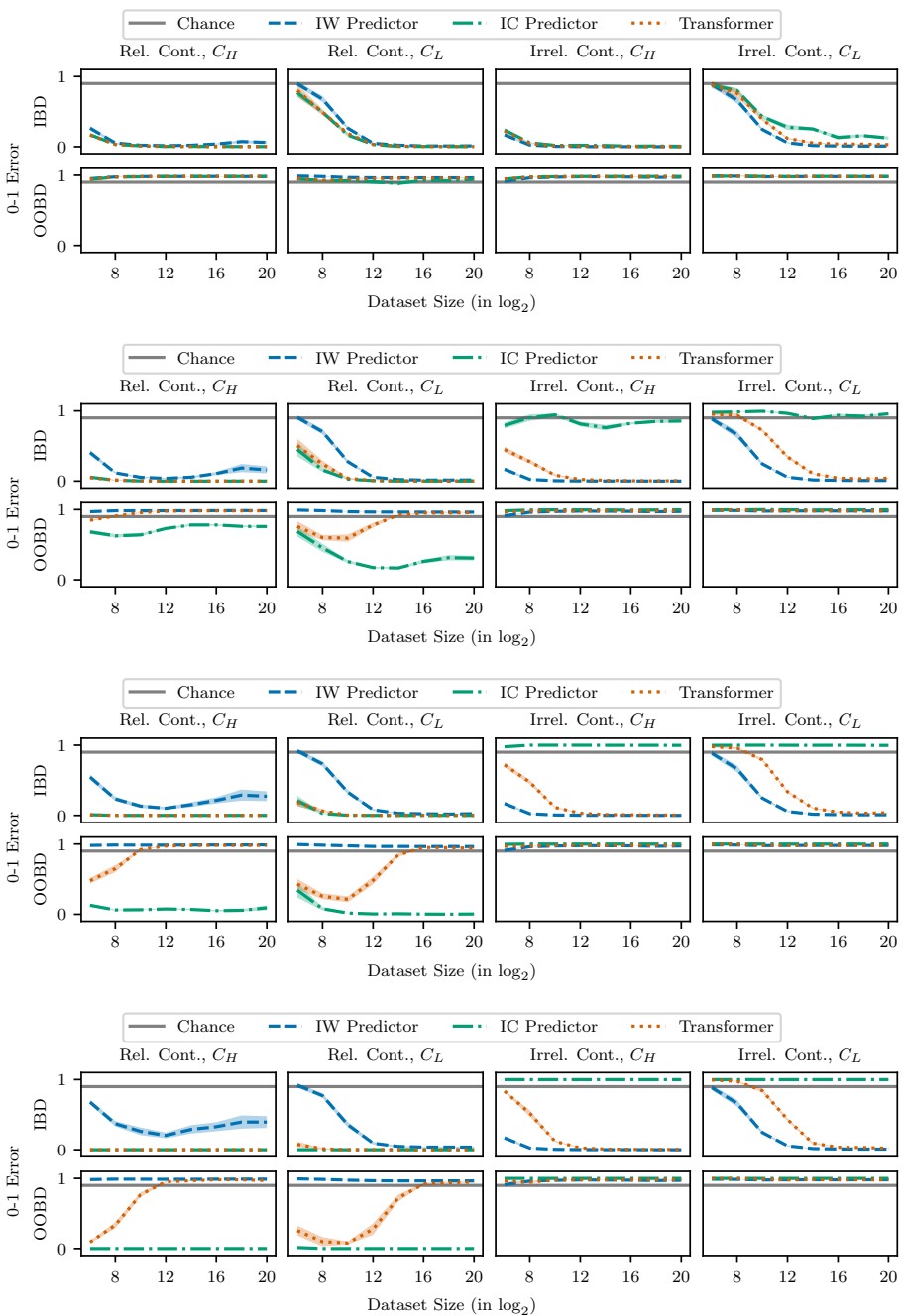

Figure 12: The validation 0-1 error of the IC predictor, the IW predictor, and the transformer predictor as we vary the number of relevant contexts $L_{relevant} \in \{1, 2, 3, 4\}$ on synthetic data, over five seeds. $L = 4, p_{high} = 0.9$, and $p_{relevant} = 0.9$ for training the transformer. For each variant, the top and bottom rows respectively correspond to IBD and OOBD errors.

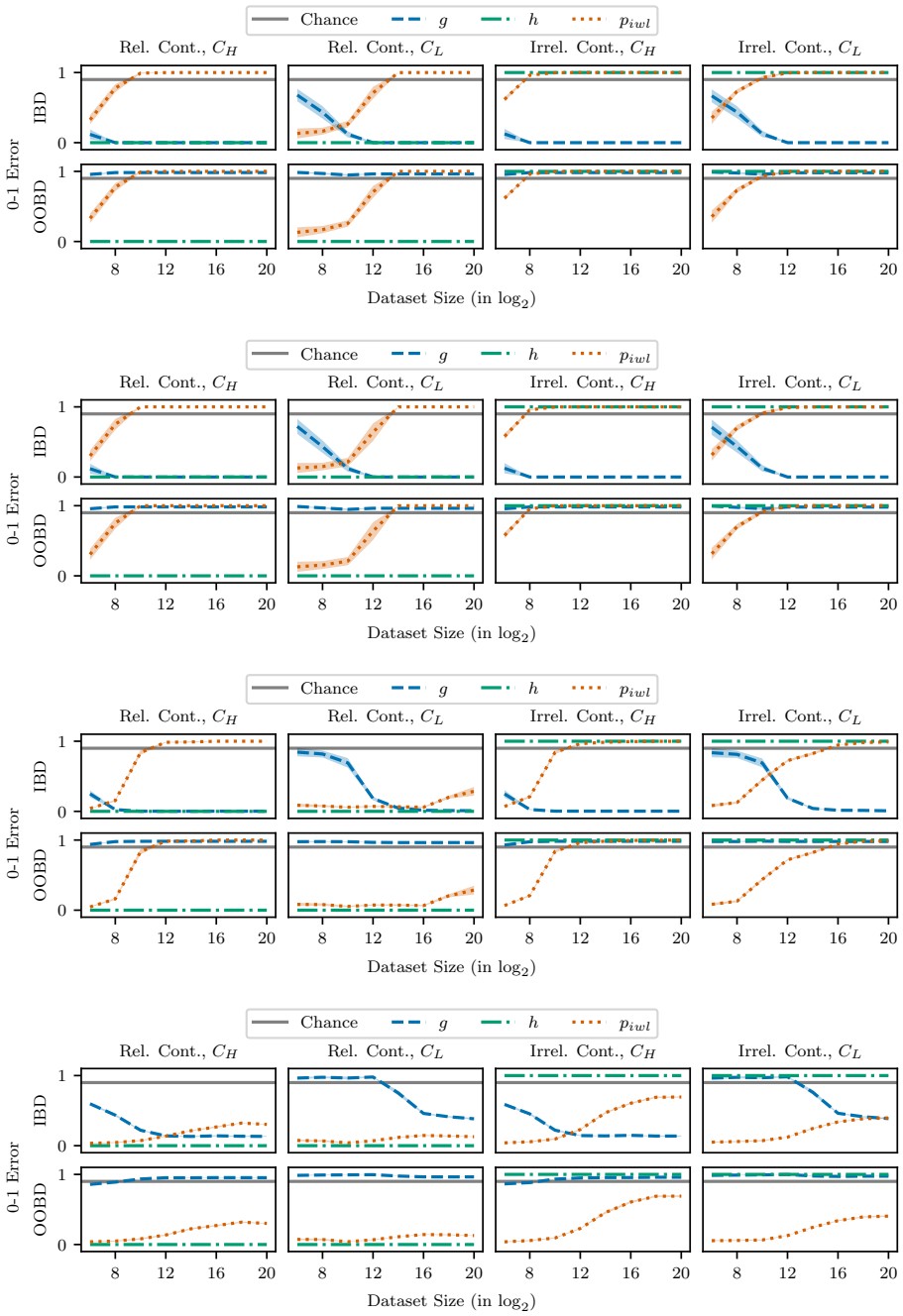

Figure 13: The validation 0-1 error of the IC predictor and the IW predictor, and the probability of selecting the in-weight predictor, i.e., $p_{iwl}$, as we vary input noise $\sigma \in \{0.0, 0.02, 0.2, 0.4\}$ on synthetic data, over five seeds. $L = 1, p_{high} = 0.9$, and $p_{relevant} = 0.9$ for training the transformer. For each variant, the top and bottom rows respectively correspond to IBD and OOBD errors.

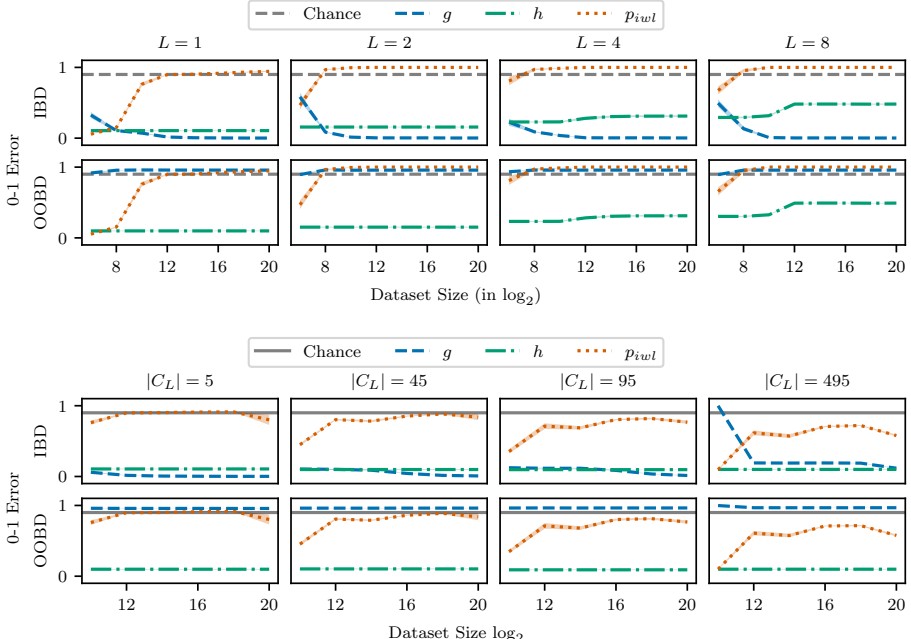

Figure 14: The validation 0-1 error of the IC predictor and the IW predictor, and probability of selecting in-weight predictor, i.e., $p_{iwl}$, as we vary context length $L$ and the number of low-frequency classes $|C_L|$ on synthetic data, over five seeds. $L = 1, p_{high} = 0.9$, and $p_{relevant} = 0.9$ for training the transformer. For each variant, the top and bottom rows respectively correspond to IBD and OOBD errors.

### B.4  OMNIGLOT

In this section we provide further analyses on the Omniglot experiments. We find similar trends as in Section 3.1, but with a few notable observations. First, even if we increase the image noise, ICL never emerges for $C_H$, while ICL always emerges for $C_L$ (Figure 17); in line with our theory, the IBD error of the "selected" IC learner is significantly lower for low-frequency classes. We notice that their IBD errors are almost identical on $C_H$ samples, with or without relevant contexts, suggesting that the IC predictor might have ended up learning IWL. Observing the IBD errors on $C_L$ samples, the IC predictor exhibits ICL. Similar to the synthetic setting, in this case there is a cross-over with small $\sigma$, whereby the IW predictor initially exhibits higher IBD error but eventually overtakes the IC predictor. Once again we see the transience of ICL in this scenario. The scenario where $\sigma = 1.0$ is the most surprising—even though the IW predictor never achieves lower IBD error than the IC predictor, the transformer still loses ICL with increasing $N$.

It also appears that for Omniglot, ICL can easily emerge for $C_L$ (Figure 18). Even when the context length $L$ increases and $L_{relevant}$ is low, the transformer can perform ICL. Nevertheless, ICL is consistently transient and of low quality, as we increase the number of samples.

While our result is related to the transient nature of ICL with increasing dataset sizes, we also demonstrate the transience of ICL with as the number of gradient updates increases. Figure 19, top, demonstrates that the transformer consistently learns ICL initially on samples from $C_L$ but always eventually loses such capability. On the other hand the transformer never learns ICL on samples from $C_H$. We also include the occurrence of low-frequency samples from $C_L$ over training (Figure 19, bottom). On average, each class from $C_L$ will be observed around 200 times after 100K iterations, while each class from $C_H$ will be observed similar amount of times after only 200 iterations. This suggests ICL, if it does appear, will diminish after 200 iterations on samples from $C_H$, hence we do not observe any ICL in Figure 19, top.

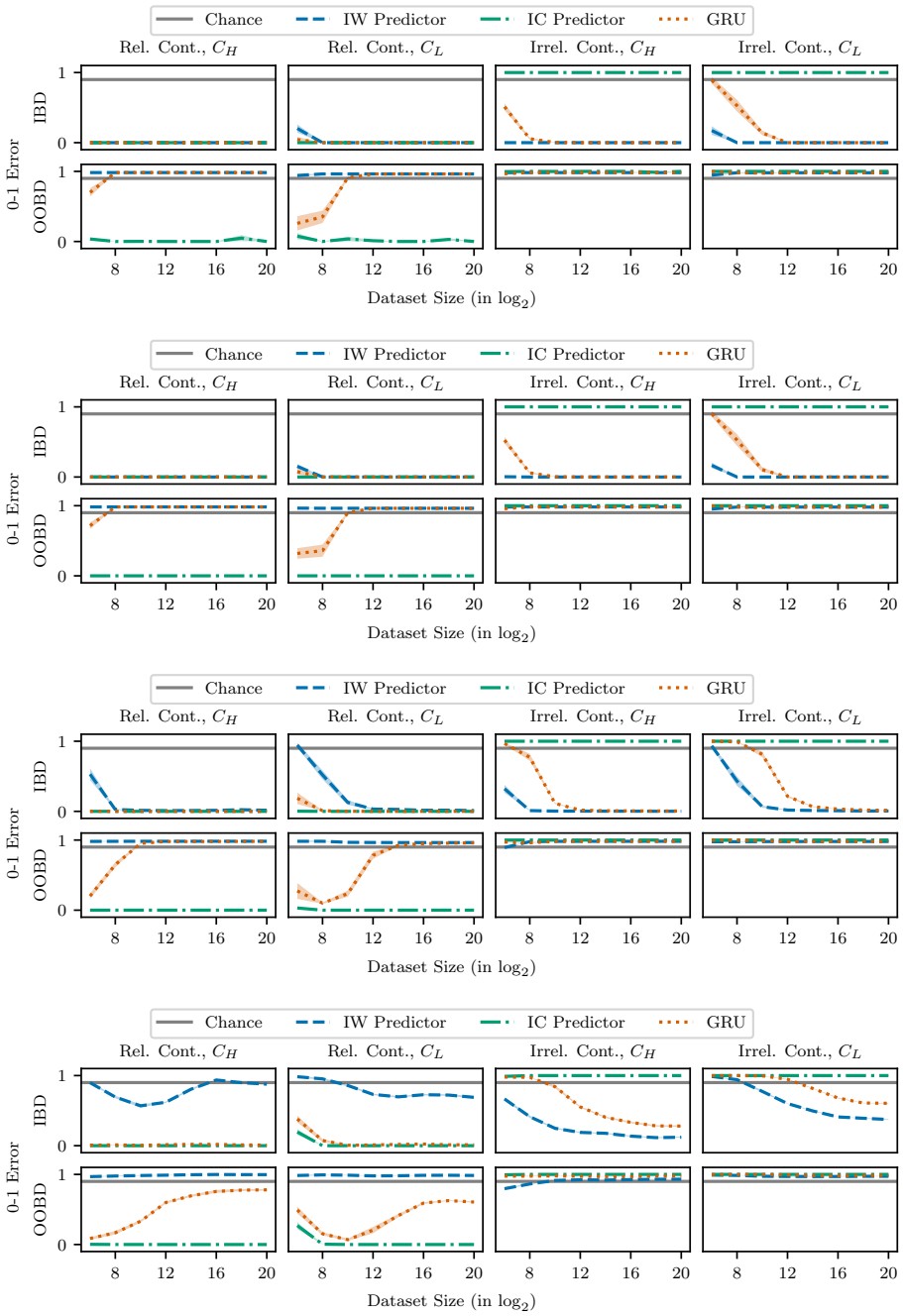

Figure 15: The validation 0-1 error of the IC predictor, the IW predictor, and the GRU predictor as we vary input noise $\sigma \in \{0.0, 0.02, 0.2, 0.4\}$ on synthetic data, over five seeds. $L = 1, p_{high} = 0.9$, and $p_{relevant} = 0.9$ for training the transformer. For each variant, the top and bottom rows respectively correspond to IBD and OOBD errors.

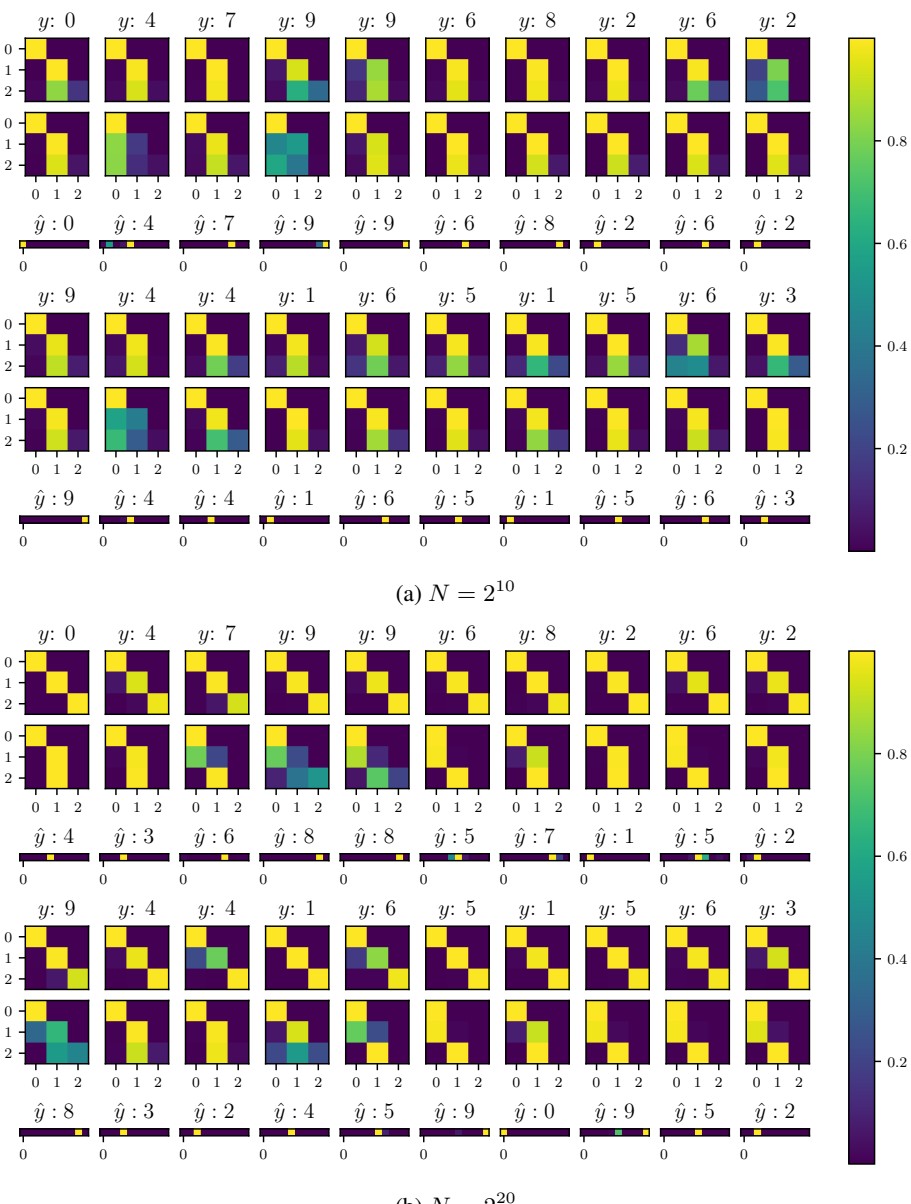

Figure 16: Images (a) and (b) correspond to the attention of the transformer trained with $N \in \{2^{10}, 2^{20}\}$ samples respectively, evaluated on 20 samples with relevant context. The transformers are provided with $L = 1$ context example. The top row corresponds to the first attention layer with $y$ being the target. The middle row corresponds to the second attention layer. The bottom row corresponds to probability of the model prediction, with $\hat{y}$ being the label with highest probability. We can see that the transformer trained with $N = 2^{10}$ exhibits ICL capabilities and its attention focuses on the context label, whereas the transformer trained with $N = 2^{20}$ no longer exhibits ICL capabilities and puts its attention on the query.

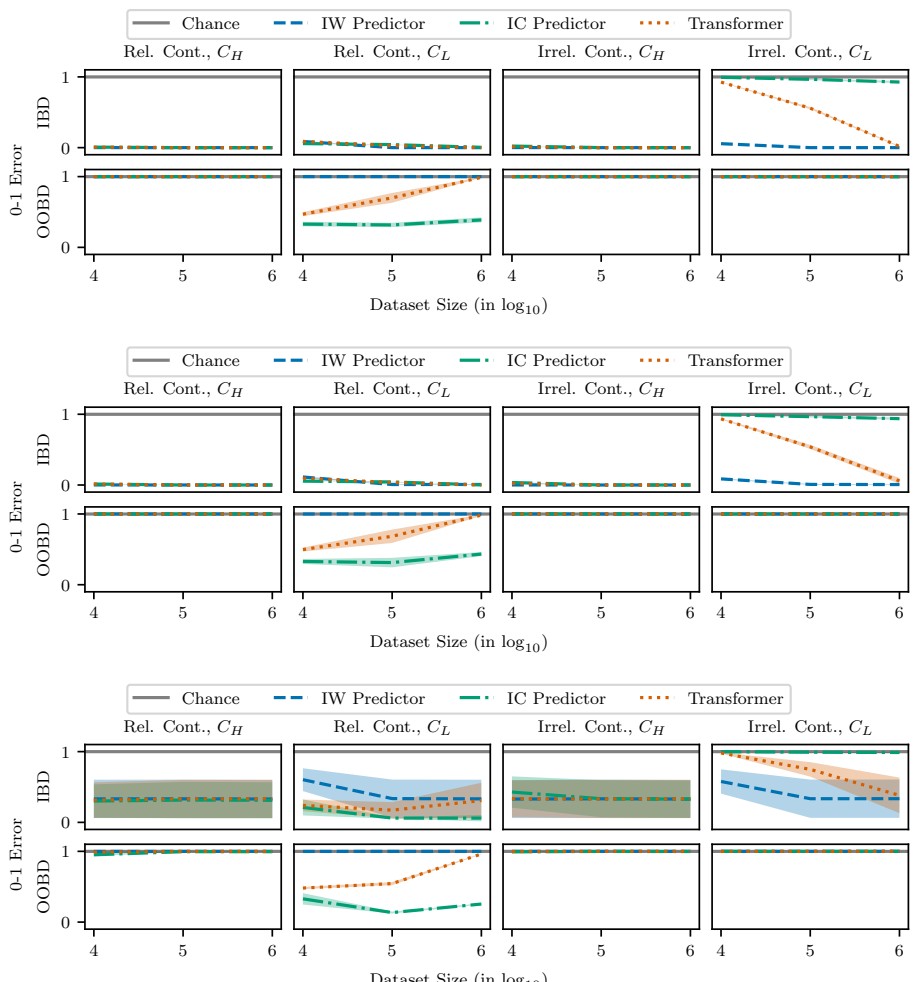

Figure 17: The validation 0-1 error of IC predictor, IW predictor, and transformer predictor as we vary input noise $\sigma \in \{0.0, 0.1, 1.0\}$ on Omniglot, over three seeds. We set $L = 2, p_{high} = 0.9$ and set $p_{relevant} = 0.9$ for training the transformer. For each variant, the top and bottom rows respectively correspond to in-base distribution and out-of-base distribution data.

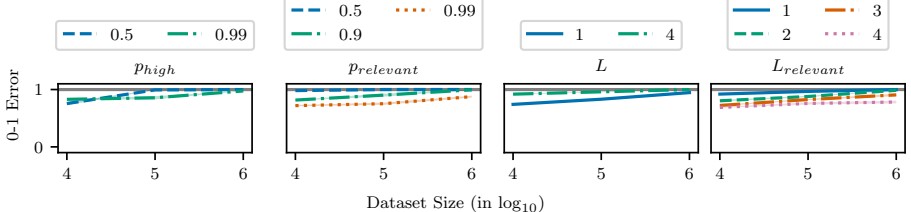

Figure 18: 0-1 OOBD error as a function of the dataset size $N$ on Omniglot, over three seeds. From left to right we vary: (1) the probability of sampling high-frequency classes $p_{high}$, (2) the probability of sampling relevant contexts $p_{relevant}$, (3) context length $L$, and (4) number of relevant contexts $L_{relevant}$ fixed at $L = 4$. The solid black line is random chance.

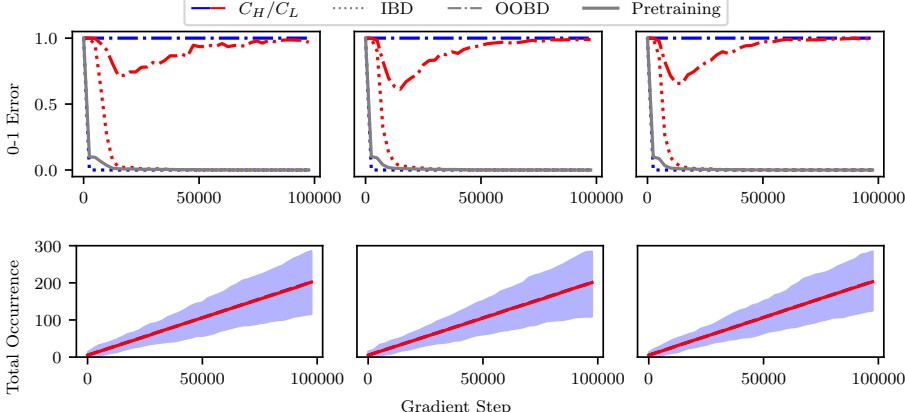

Figure 19: The top row is the 0-1 error of the transformer as we perform gradient updates on Omniglot. The bottom row is the total occurrence of each low-frequency class during training. The shaded region corresponds to max./min. occurrence. We set $L = 2, \sigma = 0, p_{high} = 0.9$ and set $p_{relevant} = 0.9$ for training the transformer. Each column corresponds to a specific seed. We observe that the transformer consistently learns to correctly predict $C_H$ quickly through IWL. The transformer also exhibits ICL capabilities on $C_L$ initially but ICL eventually diminishes.

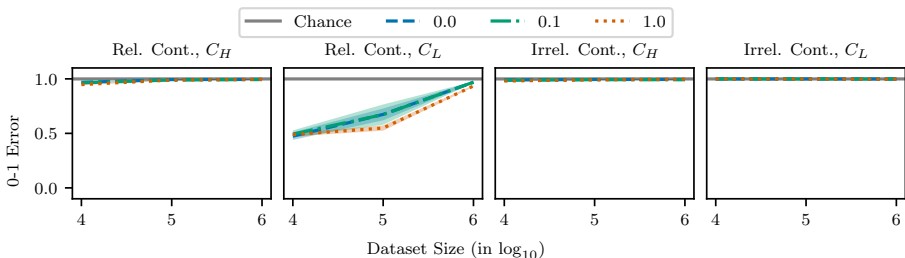

Figure 20: The ICL evaluation 0-1 errors with heldout images as we increase the number of samples $N$ on Omniglot, over three seeds. Regardless of the input noise, the transformer performs ICL initially but gradually performs more IWL as $N$ increases.

Finally, we evaluate the trained transformers on heldout images that are never seen in training. In this experiment we fix $L = 2$, vary $\sigma \in \{0.0, 0.1, 1.0\}$ and only provide one of 20 images per class in Omniglot for training the transformer. Similar to the evaluation described in Section 3 we shift the original class label by one so the transformer cannot use predict through IWL. Since $L = 2$, the transformer will need to identify the relevant context and the query through the inputs and copy its label. Figure 20 shows that even with no noise, the transformer is able to generalize to unseen images and perform ICL on $C_L$.

## C  DATA AND PROMPTS FOR LARGE LANGUAGE MODEL EXPERIMENTS

Here we provide the finetuning data (Table 1), and the evaluation prompts and results (Tables 2 and 3) mentioned in Section 4. Below we provide extra analyses on different prompts.

Table 1: The training data for finetuning the Gemini Nano 1 model. The prompts contain four real and four invented (random) names and city names, each, and force the model to perform IWL as it gives zero information regarding where the person resides in.

| Prompt | Answer |
|---|---|
| Where does Cxx1h live? Only give the name of the city. | Lkkl |
| Where does Vzbiik live? Only give the name of the city. | Plooqujhd |
| Where does Mmojkr live? Only give the name of the city. | Nwops |
| Where does Trrrrqe live? Only give the name of the city. | Qtbnaaa |
| Where does Benjamin live? Only give the name of the city. | Buenos Aires |
| Where does Liz live? Only give the name of the city. | London |
| Where does Kaitlyn live? Only give the name of the city. | Kingston |
| Where does Wiesa live? Only give the name of the city. | Warsaw |

### C.1  ADDITIONAL DISCUSSION OF EXPERIMENTS

For a well-known city like *Toronto*, the predictions are almost always in-context; Wiesa was correctly predicted to live in Warsaw. Adding some extra information to the prompt, such as *Liz lives in the city of Toronto. The weather is fantastic. Where does Liz live? Only give the name of the city.*, the probabilities of the correct answers are drastically reduced. Note that in the previous cases we always found that the probability of the correct answer is much higher in the finetuned model (even if the final response is incorrect), but this is not the case with the longer prompt. The results are similar if *Kjheergg* is used in the long prompt, though the correct prediction for (Trrrrqe, Qtbnaaa) remains. Using *Hajduszoboszlo* (a small town in Hungary, surely not in the training set many times) as the decoy city in the longer prompt results in ICL, although the finetuned model twice responded with Budapest, showing that the predicted answer is really the most likely Hungarian city. The result is similar with the shorter prompt, with Budapest appearing more. Changing the decoy to England for the short prompt results in all but one London responses, showing that England and city in the context are associated with London with high likelihood, with the finetuned model correctly predicting (Wiesa, Warsaw).

Finally, note that our experiments are similar to the flipped-label experiments of Wei et al. (2023), who examined how ICL is affected if the information in the context is contradictory to what the model memorized during training. Their experiments show that ICL can affect IWL in large models, but it is inconclusive for smaller models of similar size to Gemini Nano, as they do not seem to have sufficient in-weight knowledge. In contrast, our experiment shows that IW memorization can override ICL in the Gemini Nano model.

Table 2: The question-answer pairs and the corresponding predictions of the finetuned and the base models for the finetuning dataset, also showing the relative log-probability of possible answers. The shaded rows are the finetuned model predictions and the unshaded rows are the base model predictions. In this scenario the models must use IWL to predict the correct answer.

| Model | Question | Answer |
|---|---|---|
| Finetuned | Where does Cxx1h live? Only give the name of the city.
Lkkl -0.2666, Anytown, CA -55.3096 | Lkkl |
| Base | Where does Cxx1h live? Only give the name of the city.
Lkkl -40.5117, Anytown, CA -2.4959 | Anytown, CA |
| Finetuned | Where does Vzbiik live? Only give the name of the city.
Plooqujhd -45.9387, Bratislava -0.0659 | Bratislava |
| Base | Where does Vzbiik live? Only give the name of the city.
Plooqujhd -66.2485, Bratislava -1.3778 | Bratislava |
| Finetuned | Where does Mmojkr live? Only give the name of the city.
Nwops -6.0057, Milwaukee -0.5176, Los Angeles -12.1351 | Milwaukee |
| Base | Where does Mmojkr live? Only give the name of the city.
Nwops -32.8910, Milwaukee -4.8689, Los Angeles -1.9404 | Los Angeles |
| Finetuned | Where does Trrrrqe live? Only give the name of the city.
Qtbnaaa -0.0136, Toronto -9.0857 | Qtbnaaa |
| Base | Where does Trrrrqe live? Only give the name of the city.
Qtbnaaa -50.5695, Toronto -1.0771 | Toronto |
| Finetuned | Where does Benjamin live? Only give the name of the city.
Buenos Aires 0.0000, New York City -43.2574 | Buenos Aires |
| Base | Where does Benjamin live? Only give the name of the city.
Buenos Aires -16.4484, New York City -0.7329 | New York City |
| Finetuned | Where does Liz live? Only give the name of the city.
London 0.0000, Los Angeles -35.3694 | London |
| Base | Where does Liz live? Only give the name of the city.
London -2.6441, Los Angeles -0.2607 | Los Angeles |
| Finetuned | Where does Kaitlyn live? Only give the name of the city.
Kingston 0.0000, Seattle -45.2526 | Kingston |
| Base | Where does Kaitlyn live? Only give the name of the city.
Kingston -7.0968, Seattle -0.6796 | Seattle |
| Finetuned | Where does Wiesa live? Only give the name of the city.
Warsaw -0.0000, Frankfurt am Main -54.8158 | Warsaw |
| Base | Where does Wiesa live? Only give the name of the city.
Warsaw -3.0168, Frankfurt am Main -1.3085 | Frankfurt am Main |

Table 3: The question-answer pairs and the corresponding predictions of the finetuned and the base models for questions designed to test ICL capabilities, also showing the relative log-probability of possible answers. The shaded rows are the finetuned model predictions and the unshaded rows are the base model predictions. In this scenario the models must use ICL to predict the correct answer.

| Model | Question | Answer |
|---|---|---|
| Finetuned | Cxx1h lives in Kjheergg. Where does Cxx1h live? Only give the name of the city.
Kjheergg -1.8326, Lkkl -7.9295, Kxxh -0.4498 | Kxxh |
| Base | Cxx1h lives in Kjheergg. Where does Cxx1h live? Only give the name of the city.
Kjheergg -0.0000, Lkkl -45.5691, Kxxh -32.7908 | Kjheergg |
| Finetuned | Vzbiik lives in Kjheergg. Where does Vzbiik live? Only give the name of the city.
Kjheergg -0.2945, Plooqujhd -47.7992 | Kjheergg |
| Base | Vzbiik lives in Kjheergg. Where does Vzbiik live? Only give the name of the city.
Kjheergg -0.0002, Plooqujhd -62.2570 | Kjheergg |
| Finetuned | Mmojkr lives in Kjheergg. Where does Mmojkr live? Only give the name of the city.
Kjheergg -0.0382, Nwops -18.9681 | Kjheergg |
| Base | Mmojkr lives in Kjheergg. Where does Mmojkr live? Only give the name of the city.
Kjheergg -0.0001, Nwops -47.8115 | Kjheergg |
| Finetuned | Trrrrqe lives in Kjheergg. Where does Trrrrqe live? Only give the name of the city.
Kjheergg -21.6897, Qtbnaaa 0.0000 | **Qtbnaaa** |
| Base | Trrrrqe lives in Kjheergg. Where does Trrrrqe live? Only give the name of the city.
Kjheergg -0.0003, Qtbnaaa -52.3584 | Kjheergg |
| Finetuned | Benjamin lives in Kjheergg. Where does Benjamin live? Only give the name of the city.
Kjheergg -0.0367, Buenos Aires -25.9128 | Kjheergg |
| Base | Benjamin lives in Kjheergg. Where does Benjamin live? Only give the name of the city.
Kjheergg -0.0001, Buenos Aires -33.1918 | Kjheergg |
| Finetuned | Liz lives in Kjheergg. Where does Liz live? Only give the name of the city.
Kjheergg -0.0765, London -18.2949 | Kjheergg |
| Base | Liz lives in Kjheergg. Where does Liz live? Only give the name of the city.
Kjheergg -0.0000, London -24.7004 | Kjheergg |
| Finetuned | Kaitlyn lives in Kjheergg. Where does Kaitlyn live? Only give the name of the city.
Kjheergg -2.4694, Kingston -0.1156 | **Kingston** |
| Base | Kaitlyn lives in Kjheergg. Where does Kaitlyn live? Only give the name of the city.
Kjheergg -0.0001, Kingston -25.7090 | Kjheergg |
| Finetuned | Wiesa lives in Kjheergg. Where does Wiesa live? Only give the name of the city.
Kjheergg -1.5144, Warsaw -15.2768, Kjee -0.9055 | Kjee |
| Base | Wiesa lives in Kjheergg. Where does Wiesa live? Only give the name of the city.
Kjheergg -0.0001, Warsaw -33.9379, Kjee -15.4231 | Kjheergg |

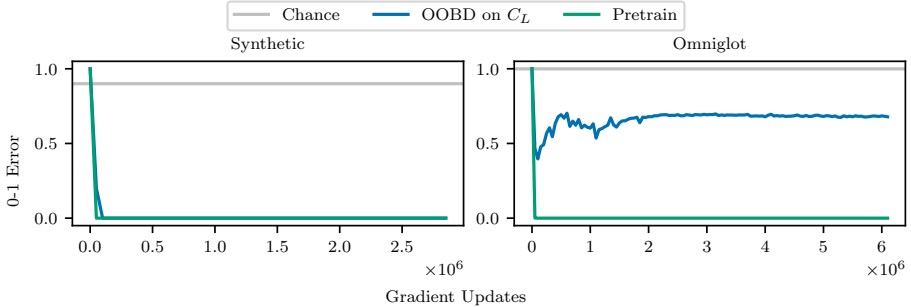

Figure 21: The 0-1 error of transformer trained over a large number of gradient updates with $N = 10^7$, $p_{high} = 0.9$, and $\sigma = 0.0$. The context sequence always includes relevant examples. **(Left)** $L = 1$ on synthetic data. **(Right)** $L = 2$ on Omniglot data. We see here that ICL is not transient in both synthetic and Omniglot datasets.

## D  TRANSIENCE OF ICL

Singh et al. (2023); Reddy (2024); Panwar et al. (2024) have previously suggested that ICL is transient with more training steps. We have conducted experiments on both our synthetic dataset and the Omniglot dataset where the transformer is always provided with relevant contexts (i.e. $p_{relevant} = 1$). Here, since the input noise $\sigma = 0.0$, previous experiments described in Appendices B.1 and B.4 have demonstrated that the transformer is capable of learning IWL to reach zero error. That is, both IWL and ICL can completely solve the tasks. In Figure 21 we see that the transformers in both cases have learned to predict in-context, and ICL has not completely disappeared even after millions of training steps. The Omniglot experiment aligns with the findings of Singh et al. (2023), as the ICL capability degrades after few million training steps but never completely disappears. On the other hand, the synthetic experiments suggests that the transience of ICL might not occur at all, contradicting the findings of Singh et al. (2023).

Our theory suggests that the transience of ICL may depend on the difficulty of the ICL and IWL tasks. This can be seen by analyzing the performance of our IC predictor (i.e., a predictor trained only with in-context data), as depicted in Figure 4. The tasks become easier as more examples in the context become relevant. When only one example is relevant, the IC predictor does not show any ICL capability, and hence learns fully to predict fully IW. This gradually changes as the problem becomes easier, and when all elements in the context are relevant, the IC predictor always performs IC prediction and IW prediction does not occur. Relating to the results in Figure 21, we find similar results—all contexts are relevant in the synthetic experiment and ICL is never transient, while half of the examples in the contexts are relevant in the Omniglot experiment and ICL is transient but some ICL capability remains in the model.

Generally, we can expect ICL to be transient when ICL is worse than IWL at the end. In our main experiments in Section 3, we have trained our transformers with mixed in-context and in-weight data. Note that the standard pretraining procedure of LLMs includes both in-context and in-weight data due to the sequential nature of the training, because at the beginning of any prompt or document there is no useful context (and there are typically many documents containing relevant contexts). In this case, the theoretical optimum performance (which is assumed to be achieved asymptotically) of the IWL predictor is better since we have samples where no relevant context is available. Nevertheless, for some inputs IWL and ICL can still have equally good performance. However, in this case the model may still have some bias towards IWL if some smoothness is imposed on $\alpha$, as because of this the inputs for which IW prediction is better will bias the selection of $\alpha$ for nearby inputs.

Finally, it is also possible that for every input both IWL and ICL can perform perfectly. For this case, the Omniglot experiment of Figure 21 and the experiment presented by Singh et al. (2023) both show that ICL may be transient (or more precisely that the ICL capability of the model decreases – we can see the same phenomenon in the performance of the IC predictor in Figure 11 for $L = 2$ or in Figure 12 for $L_{relevant} = 2$) while our synthetic experiment in Figure 21 shows that ICL is

persistent. In this case we do not know whether ICL or IWL is better (this is very hard to measure properly as a transformer trained with in-context data only can rather implement IWL as visible, e.g., in Figures 11 and 12); had we known this information we could check that the predictions of our model match reality for this edge case. Singh et al. (2023) suspected that the attention might be too soft in their case, which might limit the prediction performance of an IC learner using those attention heads, and in this case our theory would predict that ICL is transient. In our synthetic experiment in Figure 21, there is a single relevant context, and hence ICL needs to learn to predict the single label present in the input (which requires learning a mapping from the label embeddings to the label distributions), while IWL needs to learn to map the class representative to a label. These two tasks are of comparable hardness, although the orthogonality of the label embeddings might make ICL somewhat easier, in which case our theory would predict that ICL happens.

Consider another scenario where the model is provided with only in-context data but random labels. We can show that both IWL and ICL can ultimately be preferred depending on the correlation structure between the noise in the labels. For the case of fully correlated labels, when the possibly random label for the same example is identical between the context and the ground-truth response to the query, IC prediction can achieve 0 error while IW prediction will suffer because of the noise. If the randomness in the labels is independent, IWL may have an advantage: Consider the case of binary prediction where the ground truth for a given input has a main label with probability $1-p$ and a "flipped' label with probability $p$ with $p < 0.5$. If there is a single relevant context, the labels for the context and the query agree with probability $(1 - p)^2 + p^2$, leading to an ICL error probability of $2p(1 - p)$, while the error of the IW predictor always predicting the main label is $p$, and we have $p < 2p(1 - p)$ for $p < 0.5$.

