# OpenReview forum: "Toward Understanding In-context vs. In-weight Learning"
_ICLR.cc/2025/Conference — ICLR 2025 Poster_

### Official Review · Reviewer_VrYi · 2024-11-03

**Soundness:** 3
**Presentation:** 1
**Contribution:** 3
**Rating:** 6
**Confidence:** 3

**Summary:**

This paper tries to understand how trained Transformers choose between in-weights learning (IWL) and in-context learning (ICL). The paper is well grounded in recent literature studying the contrast of phenomenology and inference time behavior of IWL and ICL, especially with Singh et al. 2024’s finding that ICL can be transient. The paper first sets up a mathematical framework where vectors are input and noisy labels are used in a classification setup. The paper then discusses analytic predictors using ICL and IWL, and explains why IWL is asymptotically superior and why a neural network can find the optimal solution by gradient descent. The paper conducts experiments comparing a transformer’s validation loss to the ICL and IWL predictors to identify that noise level and fraction of rare tasks are critical variables deciding ICL/IWL. Finally, the paper shows, by fine tuning Gemini Nano 1, that fine tuning can induce loss of ICL abilities in real LLMs

**Strengths:**

Overall, the paper’s motivation is clear and the flow is consistent. Here is a list of strengths:

S1: The paper connects an observed phenomenon (ICL’s transience into IWL) with a synthetic model and theory.

S2: The intuitive interpretation that new training samples are effectively test samples under the assumptions made in Sec. 2 is useful.

S3: This work clearly demonstrates how noise level and the fraction of rare tasks can affect the ICL versus IWL.

S4: Unlike concurrent work exploring similar phenomena with similar methodology (https://openreview.net/forum?id=XgH1wfHSX8, https://openreview.net/forum?id=INyi7qUdjZ, https://openreview.net/forum?id=LbceJJc9h2), this work provides experiments with real language models. However see below for some criticism as well. Despite the criticism, the investigation of how well findings on synthetic data generalize to real models is valuable.

**Weaknesses:**

The paper has some weaknesses, listed below:

W1: Theory seems to rely on assumptions not met in realistic applications of ICL. The theory assumes that the data are drawn i.i.d, while this is usually not met in many realistic applications of ICL. Thus it is quite unclear whether new training samples can safely assumed as test samples in real LLM training

W2: The setup seems to perform a classification task where x,y are given in pairs. However, this *might* be not a good model of ICL in real LLMs as it does not involve a complex sequence-space computational structure. Recent works, e.g. Akurek et al. 2024 (https://arxiv.org/abs/2401.12973), seems to model the sequence space nature of ICL better. It seems like this seems important to understand ICL \textit{in Transformers} as Tong et al. 2024 (https://arxiv.org/abs/2405.15618 ) points out that MLPs can learn in context.

W3: I was not able to draw these conclusions in line 484 from the given Figure 7: “in line with our theory, the IBD error of the “selected” IC learner is significantly lower for low-frequency classes. Furthermore, both the IC and IW predictors achieve similar IBD errors on CH, suggesting that IWL more easily emerges on common classes even with larger input noise (Figure 7, right).” It would be great if this can be explained further

W4: While I agree with most conclusions drawn from the plots, it would be better to compute explicitly a ICL_vs_IWL ratio as a metric since currently it seems like these conclusions are drawn by matching the experiment curves with predictors by eye.

W5: It is intuitively hard to grasp the experiment. This is solely a presentation issue, it would be very helpful if there was a schematic diagram which describes all of $p_{relevant}$, $p_{high}$, OOBD, IBD, IC model, IW model, etc. The figures could also be improved, as mentioned below in questions.

W6: It is unclear how well the real LLM experiments justify the claims in the main text and the claim in Singh et al. 2024 that ICL can be transient *during pretraining*. The current experiments seem to suggest that fine-tuning can override ICL abilities, but it is very probable that the ability itself is still there while it is simply suppressed superficially. (See Jain et al 2023 (https://arxiv.org/abs/2311.12786, https://arxiv.org/abs/2410.1653 ). This would not be corresponding to the claim in the main text and in Singh et al. 2024, which I believe discusses a pre-training stage transience requiring orders of magnitude more steps than fine tuning.

**Questions:**

Questions
1. How could the real vs invented (name, city) pair result be explained? If the model is being fine-tuned, why can’t it memorize invented (name, city) pairs? Is there an intuition or is this simply the case?
2. How applicable is the theory to real LLMs?

Suggestions:
1. The current figures were quite confusing for me, the axis and plot labeling constantly changed through Fig 2,3,7,8. It would be great if the plot colors/style can be improved so that they can be better understood.
2. I think Figure 16 is main-text material, perhaps more so than Figure 8.

---

> ### Author Response · Authors · 2024-11-22
>
> We thank reviewer VrYi for their thoughtful reviews. We address your questions and comments here.
>
> > W1
>
> We agree that our model does not exactly fit the training of large language models at the token level. However, if one interprets examples to represent documents, then assuming an i.i.d. distribution during training fits the setting of large language models quite well. Also note that few shot learning is another example of in-context learning where assuming that the examples presented are i.i.d. is again realistic. Another setting for ICL is having rare classes during training and testing. In this case, the training and testing sequences can be drawn i.i.d., but there is a large number of rare classes such that each class is seen only a couple of times. In this case, ICL may emerge due to insufficient in-weight data. Note that we do not assume that the data in the context and query would be i.i.d., and our lower bound for ICL (Proposition 2) and upper bound for Algorithm 1 (Proposition 3) hold under much more general settings, without making any assumption (e.g. independence) about the data distribution.
>
> > W2
>
> We emphasize that the (input, label) setting still captures the sequential nature of ICL , since the ICL predictor is still a function of the entire sequence. However, we do not perform general sequence learning where the sequence length can vary. Our setting, also studied by Chan et al. (2022), is the simplest setting that allows us to readily investigate ICL vs. IWL in transformer networks. We can extend our study to problems with more complex sequence-space computational structure; indeed, this would correspond to having a more difficult concept class for IWL and ICL, as well as a different ICL predictor than the one instantiated in Section 2.2. We would also like to mention that Tong et al. (2024) is a concurrent work.
>
> > W3–W5
>
> Thank you for the suggestion. We have reorganized the plots to address the raised concerns to improve clarity and reduce clutter—we have delegated the less important content to Appendix B and emphasized only the important messages. We have also included a schematic diagram to demonstrate data generation, training, and evaluation (Figure 6).
>
> Regarding Figure 7 (now Figure 16 in the revised version), we can observe that when the input noise $\sigma \in \{0.0, 0.1\}$ the errors between the IW and IC predictors are similar on both the high-frequency ($C_H$) and low-frequency ($C_L$) classes, while when $\sigma = 1.0$, the IW error is clearly higher than the IC error on $C_L$. We note that for relevant context on $C_L$, the IW predictor can eventually achieve lower error than the IC predictor which is also when ICL begins to disappear.
>
> > W6 + Questions
>
> Please take a look at the section “Significance of the LLM experiments” in the general response. In particular, the LLM experiments show that on inputs with in-weight data, the IWL predictor can potentially override the ICL predictor. We do not claim that after fine-tuning, the LLM will perform IWL on all inputs, in fact, ICL is very much present in the model for other inputs. This is consistent with our claim in the main text, where we show that ICL and IWL can emerge on different parts of the input space.

---

> > ### Comment · Reviewer_VrYi · 2024-11-29
> >
> > >W1
> >
> > I am unsure about the statement "few shot learning is another example of in-context learning where assuming that the examples presented are i.i.d. is again realistic". Few shot learning prompts are often designed to be repeated examples in a very narrow domain compared to the pretraining corpus. e.g. One can repeat many word reversing tasks  (Brown et al 2020, [1]), to prompt a model to perform this task but this kind of enumeration is highly unlikely to be drawn from an iid draw.
> >
> > I might be confused here, proposition 3 is clearly stating "suppose (x,y) are drawn i.i.d.".
> >
> > >W2
> >
> > I get this, however the constrained setting does remain a weakness.
> >
> > >W3-5
> >
> > Thank you, Figure 6 really helps. But Figure 16 is an attention plot if I downloaded the most updated version?
> >
> > >W6
> >
> > I think the criticism on the LLM experiment stands.
> >
> > (this paragraph is a bit repetitive.) I understand what the LLM experiments is demonstrating, however Singh et al 2024's result is mainly about how ICL can be transient *during* pretraining. There is still a significant gap between showing that ICL can naturally emerge and dissapear during pretraining vs. showing that one can fine tune a model strong enough that the IWL circuit is more strongly activated. i.e. is it surprising that I can take a maximally ICL checkpoint from Singh et al 2024 and fine tune it so that it performs IWL to some inputs?
> >
> > As the global response mentions, it is indeed hard (obviously) to run a full pretraining experiment. I am definitively not suggesting that this should be done, but I think currently the LLM experiments are simply investigating into a very different topic than the rest of the manuscript, since it directly optimizes the model on a different distribution from the pre-training corpus. This setting doesn't seem to be connected the the rest of the manuscript, especially with the theory. This is more of a weakness in presentation/coherence issue, not one about the actual results.
> >
> > ---
> > ---
> > **Clarification:** Thank you for your reply on the weaknesses! While I think some of these might still benefit from a discussion, I must clarify that, as one can observe in my original scores: **Soundness**: 3 **Presentation**: 1 **Contribution**: 3, the main issue about the paper is its presentation and not its content.
> >
> > I think the issue of "How easily can one understand the main findings of the paper" which are in W3-5 has not been addressed. i.e. I do not think the presentation of the findings are made more clear in the updated manuscript.
> >
> > I will thus maintain my score, which is already positive.
> >
> > ---
> >
> > [1] https://arxiv.org/abs/2005.14165

---

> ### Author Response · Authors · 2024-12-03
> **Response to follow-up discussions**
>
> Thank you for the further discussions.
>
> > W1
>
> Please note that the i.i.d. assumption is not applied to the elements of the context and the query but to the whole context and the query, that is $(\tilde{x}_i, y_i)$ are i.i.d. for all $i$, but there can be arbitrary correlation among the examples $(x^1_i,y^1_i), (x^2_i,y^2_i),\ldots,(x^L_i,y^L_i),(x,y)$ appearing in the context and the query (which allows the setting to model few-shot learning).
> The i.i.d. assumption here means that we randomly sample few-shot examples for various problems. We agree that it is possible that the training data may not follow this exactly, for example, it may follow a two-step procedure where first a problem is selected randomly and then several samples are selected from the same problem), but we believe that the i.i.d. assumption still gives a useful model.
>
> Also note that our theory about ICL and IWL suggests (and also the experiments support this) that if some similar data is repeated too many times, the model will rather memorize the label than learn to predict based on the context, which suggests that ICL is more prominent the closer the data $(\tilde{x}_i, y_i)$ is to i.i.d.
>
> To see this more formally, assume that the data distribution follows a two step generation procedure $P_{d|t} (data|task) P_{t} (task)$; then the options may be (i) the i.i.d. case where every sample is generated independently by first sampling a task $t \sim P_t$ and then a data $d=(\tilde{x},y)$ from $P_{d|t}(\cdot | t)$, in which case we obtain i.i.d. samples from the marginal for $d$; and (ii) the case where we take multiple samples from fewer tasks, that is, first we sample independently a few tasks $t$ from $P_t$, then for each task $t$ we sample several data points from $P_{d|t}(\cdot | t)$. In the latter case, if the points from a given task are more similar, then the data generated in this way is more likely to encourage IWL than in the i.i.d. case of (i).
>
> To clarify, Proposition 3 in our initial response is now Proposition 2, where we have regret guarantees without distributional assumptions.
>
> > W2
>
> One more comment about this question: We lightly discuss the effect of sequential training (of real LLMs) compared to our model, arguing that it actually may contribute towards the transience of ICL; see the discussion around line 1770 in our new Appendix D about the transience of ICL.
>
> > W3-W5
>
> Should be Figure 17 now (apologies, we have uploaded another revision in the meantime).
>
> > W6
>
> In our view, the LLM experiment is an efficient simulation for pretraining, as the fine-tuning procedure we are using is identical to what is used in pretraining, so we are not specifically optimizing for IWL vs. ICL. Of course we changed the data distribution, as instead of mixing our fine-tuning data into the usual (pretraining) data distribution, we only used the data we wanted the model to memorize, which is the most sample-efficient way to increase the number of times the fine-tuning data appears in the training, as required by our theory (although it violates the random order used in pretraining). This reduced the training time needed for the memorization effect to show up; mixing our finetuning data with other “usual” data would have slowed down the training. Also note that we do not consider the effect that the model is post-trained (using supervised finetuning and RLHF) between the normal pretraining and our “extension” of it. Nevertheless, based on the above considerations we think that our LLM experiment is sufficiently similar to pretraining with a changed data distribution (where the fine-tuning data appears more), and hence it is able to simulate how IWL and ICL changes during training (where the original and the fine-tuned models play the role of an earlier and a later checkpoint in the training, respectively).
>
> Note that we also added a sentence to the beginning of Section 4 to clarify the role of the LLM experiment.
>
> > Clarity of the presentation
>
> We are sorry to hear that we have not managed to address your concerns about the clarity of the presentation. If you have any suggestions that would help here, we would be grateful to hear about them.

---

### Official Review · Reviewer_8YMd · 2024-11-04

**Soundness:** 4
**Presentation:** 3
**Contribution:** 3
**Rating:** 8
**Confidence:** 3

**Summary:**

Summary of the paper:

* The paper defines a sequential, tabular supervised classification task as a
  simplified setting in which to study the dynamics of in-context vs.
  in-weight learning.
* The paper considers three models in this setting:
  1. An 'in-weight learner' model capable of predicting labels based only on
     the most recent query.
  2. An 'in-context learner' model that predicts labels based instead on a
     sequence-dependent weighted combination of previous labels appearing in
     the sequence (specifically, the weights are given by a softmax
     distribution based on similarity to the previous queries).
  3. A contextual mixture model that can (learn to) choose based on the input
     sequence whether to predict according to the first model or the second
     model, or interpolate between the two.
* The paper describes a theoretical analysis of the performance of these
  three models in the synthetic task:
  * Deriving an upper bound on the expected test loss of the in-weight
    learner.
  * Deriving an upper and lower bound on the loss for a particular input of
    the in-context learner.
  * Giving a family of algorithms for learning all three models and bounding
    the regret of the combination model in terms of the regret of the
    learning algorithms used for learning some of the components.
* Based on the nature of these bounds, the paper argues that this model
  captures phenomena observed in prior empirical work on in-context learning
  and in-weight learning, namely that rare classes drive the emergence of
  in-context learning (Chan et al.) and that ICL disappears after enough
  training (Singh et al.).
* The paper also includes some experiments with small transformers on
  synthetic data and natural language data and comparisons of the theoretical
  model's qualitative predictions to the resulting empirical observations.
* There is also a small qualitative study of a LLM's dependence on knowledge
  from fine-tuning versus knowledge from its context.

I wrote a somewhat long review. Therefore, I also include a summary of my
review as follows.

* I think this paper presents some interesting ideas and results. For
  example, I like the sequential supervised classification task and I like
  the three-part model.
* However, I find the results do not achieve the paper's stated aims of
  capturing empirical phenomena to the degree claimed, due to what appear to
  me to be several issues, namely:
  * fundamental mismatches between the theoretical framework and results and
    the phenomena studied in prior work,
  * some limitations in the theoretical results that do not appear to have
    been acknowledged, and
  * a lack of connection between the theoretical results and the experiments.
* Overall, I consider the framing of the paper to be inaccurate and therefore
  in my judgement the paper should not be accepted in its current form,
  though I am certainly open to discussion and revisions to the framing to
  acknowledge and address these gaps as I think the work does have value.
* I also noted several questions and minor corrections, mainly regarding the
  theoretical sections.

**EDIT TO ADD:** Summary of discussion.

* The authors addressed my concerns about the framing through their responses and revisions:
  * They refined their claims about connections to prior work, resolving my concern about the mismatch between their theory and prior work.
  * They strengthened their theoretical results and acknowledged the remaining limitations, resolving my concern about the limitations of their theory.
  * They clarified the connection between their theory and experiments and added new synthetic experiments with a more direct connection, resolving my concern about the connection between their theory and experiments.
* With the authors having resolved all of my concerns, I am now pleased to be able to recommend the paper for acceptance, and raised my rating from 5 to 8.

**Strengths:**

The paper addresses an important topic in the science of deep learning,
namely the phenomena of the emergence and transience of in-context learning
(versus in-weight learning).

On this topic the paper presents an interesting learning setting involving an
in-context supervised classification task that captures several important
elements of in-context learning. At the same time, this setting is simpler in
some ways than in-context linear regression or other comparable synthetic
learning settings.

I think the overarching story developed in the paper (roughly, in my
understanding, that in-context learning arises as a heuristic that can reduce
loss in areas of the input distribution where in-weight learning has not
gathered enough experience to provide a specialised and more accurate
prediction) is an interesting and thought-provoking perspective.

I think the three-part model is an interesting and thought-provoking toy
model of in-context learning versus in-weight learning, and I think the
theoretical analysis is informative and supports the aforementioned
overarching story.

**Weaknesses:**

**W1. Mismatch with Singh et al.'s setting.**
In the introduction you write that your analysis "explains the findings of
Singh et al. (2023)," namely the transience of in-context learning, giving
way to in-weight learning. It doesn't seem to me that your model captures the
phenomenon observed by Singh et al. In my understanding, a fundamental
feature of the setting of Singh et al. is that there is no train-loss
incentive for an ICL solution over an IWL solution, because both paradigms
are equally capable of solving the task (yet, they observe, ICL still emerges
and still gives way to IWL in the end). In contrast, in your model, as you
show, based on your formulation of ICL and IWL there is a performance lower
bound to ICL which the upper bound on IWL performance is eventually able to
surpass.


**W2. Incomplete characterisation of the emergence of ICL for rare classes.**
On lines 207--209, you write "Under this [oracle] algorithm, ICL will first
emerge on rare classes ... This phenomenon is consistent with the empirical
observation that in-context learning emerges with rare classes in the
training data...".

It appears that this characterisation of the oracle algorithm is based on the
observation that, for some specific values, the IW upper bound starts higher
than the IC upper bound (figure 1), suggesting that in these cases the simple
model will use the IC sub-model rather than IW sub-model. However, this is
not implied by the theory as the IW bound is merely an upper bound on the IW
risk (I also note in a question below that the IC bounds are not of the IC
risk but of a particular sample of the risk).

I believe a more accurate characterisation would require the stronger
assumption that the IW risk happens to be higher than the IC risk for the
rare classes in question. This arrangement is permitted but not implied by
the comparison between upper bounds. It would be implied if you had a lower
bound on the IW risk and this was higher than an upper bound on the IC risk.
I am unsure whether such a bound is plausible. In my opinion this gap between
the model and the phenomenon should be noted explicitly as a limitation of
the analysis.

(Note that this issue is not localised to lines 207--209, but that appears to
be the most explicit discussion of the comparison between the phenomenon and
the theoretical framework. The favourable result of the comparison is a theme
throughout the rest of the paper.)


**W3. Incompletely addressed limitation of tabular analysis.**
There are several details in the theoretical framework and analysis that
assume a tabular setting. In footnote 1 you claim that the results can be
easily be extended to a continuous input space. It is not clear to me that
this extension would be easy, since, beyond (yes) easily extending the
definition of the task and the models, the bounds and the proofs seem to be
tightly coupled to the tabular setting. It is unclear to me whether
qualitatively similar bounds could be achieved in a continuous input setting.
On what basis have you made this assertion?


**W4. Specific model of in-context learning appears quite restrictive.**
The paper talks generally about in-context learning in transformers, but for
the three-part model your formulation of in-context learning appears
surprisingly restrictive, namely at first limiting the prediction to a convex
combination of context labels and then by the relevant theorem further
limiting to the specific modelling choice of using a softmax distribution
based on query similarity for weighting the labels.

It seems to me that mechanisms for in-context learning in a transformer are
unlikely to be limited in this way, and the ability to use alternative
mechanisms would generally serve to broaden the model class under
consideration and therefore put your performance lower bounds into question.
In other words, these restrictions on the ICL sub-model seem to be the source
of the conclusion that IWL is eventually preferable to ICL, which does not so
much answer the question of why IWL is eventually preferred as raise the
question of why we should believe that ICL is limited in this specific way.

**W5. Gap between theory and experimental study.**
The experimental methodology involves comparing one transformer's performance
with that of two other 'baseline' transformers, one trained on data that is
supposed to incentivise ICL and another trained on data that is supposed to
incentivise IWL. The comparisons between the 'middle' transformer and these
baselines are then used to corroborate the predictions of the theory.
However, the use of pretrained transformers as baselines creates a confound
that the 'IWL transformer' may not resemble the IWL model in the theory and
(even more plausibly) the 'ICL transformer' may not resemble the ICL model in
the theory. This would potentially undermine the connection between the
theory and the experimental results (and, in turn, the relevance of the
theory). This gap appears not to have been acknowledged in the paper.

Given that you have details models of these learners and have posited
specific training algorithms capable of achieving your regret bounds, and in
the synthetic data experiments your set up seems to match the theoretical
assumptions (apart from the tabular input space, though presumably the
synthetic data setting could be modified to satisfy this assumption along
with any others I have overlooked), it seems like it would have been possible
to implement your theoretical models directly and compare the performance of
these models to the baseline transformer (or the middle transformer).


**W6. Unclear relevance and novelty of LLM experiments.**
The LLM experiments are interesting in their own right, however I came away
feeling unimpressed for several reasons:

1. I felt you could have done more to connect these experiments and their
   results to the theoretical framework developed in section 2. At the moment
   I don't see what the LLM experiments say about the theory and I don't see
   what the theory says about the LLM experiments.

2. The total number of examples tested is small (if my understanding is
   correct and all examples are listed in the appendices). No statistical
   analysis has been performed on any specific questions about the results to
   see if the trends reported are statistically significant.

3. Though it is not my area, I have the feeling that there exists at least
   a small number of papers having already studied the topic of the
   competition between knowledge from fine-tuning and knowledge from contexts
   in determining the predictions of LLMs. (I regret I have not been able to
   locate any examples, but one that comes somewhat close is Wei et al.,
   2023, arXiv:2303.03846, which is actually cited in passing in the
   introduction but is not discussed in any detail). I would have liked to
   see you contextualise their experiment and results within this literature.

**Questions:**

Theory:

1. What parameterisation of $\alpha$ do you have in mind? Simply to learn a
   value for every $\tilde x$ (i.e., again tabular)?
   * If so, how can you expect it to know to route to the IC model when you
     haven't seen enough of an example for $g$ to be useful? Is it supposed to
     be initialised as biased towards routing to $h$?

2. In Proposition 1 (for example), over which distribution are the
   expectations of $y$ taken? Is $y \sim y^\ast(x)$?

3. In Proposition 2 and Corollary 1, if I am not mistaken, your bounds apply
   to a given sample of the expected risk, and they depend on the sample. For
   example, in corollary 1, you bound $\mathrm{CE}(h(\tilde x), y)$ for a
   particular $\tilde x$ and $y$. Since you later compare these bounds to the
   IW upper bound which is of the expected risk, could you comment on the
   relationship between these bounds and the expected risk for the IC learner?

4. In Corollary 1, I was confused by the line "Assume the labels $y^\ast$ are
   one-hot (deterministic)".
   * If I understand correctly, in the data generating process, the concrete
     *labels* $y$ in any given example are *already* one-hot, as stated on
     lines 120--121 "When we sample $x$ in the context of as the query, we
     sample $y=e_i$, the $i$-th standard basis vector, ..."
   * I thought, possibly you meant that the ground truth labelling *function*
     $y^\ast$ is assumed to be such that the ground truth label $y^\ast(x)$
     is deterministic? But this seems like too strong of an assumption, since
     you go on to talk about IWL being better with enough samples *if* the
     variance is sufficiently low, and if you were assuming the variance was
     zero for your ICL bound then it wouldn't make sense to say this.
   * In fact, the assumption that $y$ is one-hot appears to be used in the
     proof for proposition 2, whereas no additional assumption on $y$ or
     $y^\ast$ seems to be used in the proof of corollary 1.
   * So, I think this assumption is redundant. Is that correct?

5. In Corollary 1, you give both a lower bound in terms of $k$ and also an
   exact value in the case where $k=L$. There is no restriction on $k$ in the
   general case and so I assumed that if I put $k=L$ the lower bound
   should lower bound the exact value. However, if I am not mistaken, if $k=L$
   then the lower bound simplifies to $1-\epsilon$, which is larger than the
   exact value $-\log(\epsilon)$ for some values of $\epsilon$. Is there a
   mistake in the lower bound?

6. In the figure 1 caption, what is the "Big-O constant of the IW risk"? I saw
   no asymptotic analysis in the IW bound.

Notation and terminology:

1. Corollary 1: I invite you to consider using a roman font to denote cross
   entropy loss (as in "$\mathrm{CE}$") so as to prevent lexical ambiguity
   with the variable $C$ and an undefined variable $E$ for readers parsing
   the result statement.

Typos:

1. There are two separate reference list entries for Edelman et al. (2024).
2. There are two separate reference list entries for Zhang et al. (2024).
3. Line 90 "attention attention"
4. Line 148 "cdot" presumably should be "\cdot"
5. Line 304 "transient of ICL"
6. Line 345: It says "A generic model trained on data with $p_{high}$ is
   referred to as the *transformer.*" Is that meant to be $p_{relevant}$?
7. The proof of Proposition 1 uses notation $R_x$ for expected risk without
   having introduced this notation anywhere (as far as I can see).

---

> ### Author Response · Authors · 2024-11-22
> **Response for Weaknesses**
>
> We thank reviewer 8YMd for their detailed feedback, below we address their comments and questions. We have also fixed the typos, thank you for raising them.
>
> > W1
>
> Please see the section “The theoretically optimal performance of IC and IW predictors, and their selection” in the general response.
>
> > W2
>
> Please see sections “Claim on “Regarding the claim about ‘ICL emerging first in rare classes’” and “The theoretically optimal performance of IC and IW predictors, and their selection” in our general response.
>
> Regarding Figure 1, the bounds properly describe the typical trends of the error of the predictors; the figure only illustrates the different regimes which show up when we compare IWL with induction-head-type ICL (that is why we neglected the constants when plotting the graphs). The figure also illustrates the typical behavior that ICL becomes harder when the context length and the amount of irrelevant information increase (and can occur anywhere in the context).
>
> > W3
>
> Similar bounds can be obtained for the continuous input space using standard statistical learning theory results. Indeed, one can obtain generalization bounds for the IWL predictor that decay with $O(1/\sqrt{N_x})$ and depend on some complexity measures of the function class (e.g., Rademacher complexity) [1]. These bounds still show that as the sample size increases, the excess risk converges to 0 in the limit. Our regret bounds in Section 2.3 automatically hold  on continuous domains. However, to ensure that the regret for learning $\alpha$ can be meaningfully optimized, we need some additional assumptions (in the tabular setting this is an online learning problem with linear loss functions). A sufficient assumption is that the function $\alpha$ can be parametrized with a finite number of parameters such that the loss function is convex in these parameters and the resulting function class is rich enough to partition the input space into two sets such that IC prediction is better on one set and IW is better on the other. We are happy to discuss these assumptions in the paper, but we thought that the technicalities needed to make these precise (e.g., defining Rademacher complexities) could perhaps draw the attention from the underlying simple phenomenon.
>
> [1] Bousquet, Boucheron, Lugosi, Introduction to statistical learning theory. Advanced Lectures in Machine Learning, Springer, pp. 169–207, 2004
>
> > W4
>
> Please see the section “Criticism that the stylized model/data distribution is too simplistic in the theory” in the general response.
>
>  > W5
>
> Please see the section “Regarding the point that the IC and IW predictors in the experiments might not exhibit only ICL and IWL respectively” in the general response
>
> > W6
>
> Please see the section “Significance of the LLM experiments” in our general response
>
> > Comparison to Wei et al., 2023
>
> Thanks for bringing up this paper. Indeed, it looks at how the semantics (most similar to in-weight knowledge in our terminology) effects in-context learning (in much larger language models), but they consider much higher-level features of the models (such as size), and not the actual “strength” of in-weight knowledge/generalization as we do. Most similar to our investigation is their experiment comparing ICL abilities of a language model and its instruction-tuned version, arguing that instruction-tuning helps ICL – in our experiments this would be similar to providing in-context examples to boost the ICL capabilities of the model. Our experiments are similar to their flipped-label scenario, where their in-context examples are in contradiction to the models’ semantic knowledge, but they examine dynamics between IWL and ICL as a function of simple model parameters, while here we are interested in much more refined connections, and our LLM experiments aims to control directly the strength of the model’s in-weight (semantic in their terminology) knowledge, as much as possible given the unknown training data and the biases already encoded in the trained model.

---

> > ### Author Response · Authors · 2024-11-22
> > **Response for Questions**
> >
> > > Q1
> >
> > Yes, $\alpha$ is tabular given the context $\tilde{x}$. If $g$ hasn’t seen enough examples to be useful compared to $h$, then it will have a larger test error, and $\alpha(\tilde{x})$ will learn to put more mass on $h$.
> >
> > > Q2
> >
> > $y^*(x)$ defines the distribution of $y$. However, we use a one-hot representation of the labels, so instead of having label $i$, the label is $e_i$, the $i$th unit vector. Note that there is a slight abuse of the notation as we used $y^*$  to denote both the true label distribution and the optimal prediction in the bound of Proposition 1 (which are the same in case of the logarithmic loss used to train the models).
> >
> > > Q3
> >
> > The lower bound shows that given data with irrelevant labels in the context, under our assumptions, the stylized ICL predictor has bias that does not vanish even if the sample size grows without bound, resulting in a lower bound that is independent of the minimum achievable error achievable by an IW predictor. In contrast, the error of the IW predictor approaches the minimum error as the sample size grows.
> >
> > As we mention in Lines 212-214, when $y^*(x)$ has low noise, under some conditions the IWL minimum error is smaller than the ICL error lower bound. Here is a concrete example that we have included on lines 214-220: for simplicity, let $\epsilon = 0$ and $B=1$. Fix $L$, for any $k  \ge 1$, the minimum ICL error is bounded from below by $\frac{1}{1+ (L-1)e^2}$. Under the cross entropy loss, the minimum error for the IW predictor is the entropy of $y^*(x)$. Suppose $y^*(x)$ is supported on two labels. Then y has a binary distribution, whose entropy is a continuous function with minimum value 0. Therefore, if $y^*$ concentrates sufficiently on one of the possible outputs, the entropy is smaller than $\frac{1}{1+ (L-1)e^2}$, and under this $y^*(x)$ the minimum error of IWL is smaller than that of the ICL predictor.
> >
> > > Q4
> >
> > In Line 188, the assumption is on $y$ instead of $y^*$. The labels $y$ are one-hot, but the expectation is not. You are correct that no additional assumptions on $y^*$ are needed.
> >
> > > Q5
> >
> > $1-\epsilon$ is always smaller than $-log(\epsilon)$ for all $1 > \epsilon >0$. We have added this clarification in the proof in our revision.
> >
> > > Q6
> >
> > We take the constant to be 20 in Figure 1, but this constant, which summarizes the contribution of all parameters in the second term apart from $N_x$, does not substantially affect the graphs and the conclusions.

---

> > ### Comment · Reviewer_8YMd · 2024-11-26
> > **Resolved concerns**
> >
> > As I said, your response also addressed a number of my concerns, with the following notes/suggestions.
> >
> > **W3. Tabular vs. continuous input domains:**
> > Thank you for outlining this extension and explaining the simplification. It
> > would be valuable if you could add at least this much detail in an appendix
> > linked from the footnote.
> >
> > **W5. Gap between theory and experiment:**
> > I appreciate the experiments in the synthetic learning setting. Note I
> > haven't reviewed these new experimental results closely.
> > Thank you for acknowledging the potential gap
> > between your main-paper experiments with the learned proxy transformers. In
> > my opinion it would be helpful if this gap could be more prominently
> > acknowledged as a limitation in the main text.
> >
> > **W6. Significance of LLM experiments:**
> > Thank you for the detailed response. Some of this information would be worth
> > including in the main text or, if space cannot be made, in appendix C,
> > namely:
> >
> > * The key point that your experiment shows that, in comparison to however
> >   much memorization influences the answers of the base model due to
> >   pre-training, additional fine-tuning to memorize a fact can increase the
> >   extent to which the model appears to rely on memorisation; and how this
> >   aligns with your story about the preference for IWL being driven by the
> >   rarity of this conclusion in the data.
> > * The observation that the logits for the IWL answer always increase for the
> >   fine-tuned model in the prompt-with-conflicting-context setting.
> > * The disclaimer that you intend for the experiments to be taken as a small
> >   number of illustrative examples on a real (small) LLM, insufficient for
> >   drawing conclusions about statistical significance.
> > * The comparison with the flipped-label experiment in Wei et al. (2023) from
> >   your individual response.

---

> ### Comment · Reviewer_8YMd · 2024-11-26
> **Follow-up questions**
>
> Thank you for your clarifications. You have answered most of my questions but I have some remaining follow-up questions. In the interest of prioritisation, my score is more based on the 'weaknesses' discussion above (to which I will reply shortly) than these questions, and I will understand if you might not get time to address all of these questions.
>
> **Q1. Tabular alpha:** Thank you for clarifying that $\alpha$ is tabular. My
> follow-up question remains. Since $\alpha$ is tabular, the only way for it to
> learn to put mass on $h$ vs. $g$ given a context $\tilde x$ is for $\tilde x$
> to come up during training. But if this context comes up during training,
> that also allows (tabular) sub-model $g$ to memorise it. As soon as $\alpha$
> observes that $g$ achieves high loss on the context and starts putting mass
> on $h$, $g$'s loss will rapidly decrease, and it would actually be apropriate
> for $\alpha$ to put the mass back onto $g$. Is that right?
>
> **Q2. Sampling distribution for $y$:**
> Thanks for clarifying. I see this is outlined in the preliminaries section.
>
> **Q3. Example risk vs. expected risk:**
> I don't think you really answered my question. I'm concerned about the
> apparent comparison between:
>
> 1. the upper-bound on the expected risk of IWL, where the expectation is
>    taken over all $y$ and holds for any $x$, and
> 2. upper- and lower-bounds on the cross entropy for a given $x$ and $y$
>    assuming that $x$ has a certain number of irrelevant labels (which depends
>    on the concrete value of $y$.
>
> So the first bound is an average and the second is a function of $y$ (through
> the variable $k$ which implicitly depends on $y$).
>
> It seems that in order to perform a direct comparison with the IWL bound, you
> would want to compute the expected-risk version of the ICL bounds. Note that
> this would require comparing bounds for different values of $y$ and therefore
> different values of $k$. Does that make sense?
>
> **Q4 Assumption on $y$ in corollary 1:**
> Sorry, I either misread or misquoted the statement of the corollary. My real
> question is, why is it necessary to explicitly state the assumption that $y$
> is one-hot again in the statement of this corollary?
>
> If it's a stylistic choice, fine---I'm just checking that I am not missing
> something. I thought this assumption is also needed and used for proposition
> 2 (it's cited in the proof) but it wasn't stated explicitly in the statement
> of proposition 2.
>
> **Q5. Upper and lower bounds crossing:**
> Thanks for clarifying the intended range of $\epsilon$. It seems my original
> question was improperly motivated by looking at logarithm base 10, for which
> $-\log_{10}(\epsilon) < 1-\epsilon$ for some $\epsilon \in (0,1)$. My
> mistake.
>
> **Q6. The big-O constant in figure 1:**
> Thanks. Let me check my understanding:
>
> * The "second term" you refer to in your answer is
>     $6 G_\infty \sqrt{\log(2|\mathcal{X}|C/\delta)/N_x}$
>   which can be rewritten as
>     $6 G_\infty \sqrt{\log(2|\mathcal{X}|C/\delta)} \cdot 1/\sqrt{N_x}$.
> * As you observe on line 211, this term is in $O(1/\sqrt{N_x})$. The
>   corresponding "big-O constant" to which you refer in the figure is
>   therefore $6 G_\infty \sqrt{\log(2|\mathcal{X}|C/\delta)}$
>   (this is the bit I missed motivating my original question).
> * To avoid specifying each individual quantity determining this constant, you
>   just replace it with 20 for the figure, in other words you just plot the IW
>   risk bound as
>     (first term of IW risk bound) $+ 20/\sqrt{N_x}$.
> * It "does not substantially affect the graphs and the conclusions" in the
>   sense that regardless of the constant, the IW upper bound still falls below
>   the IC risk lower bound for large enough $N_x$ and $k$.
>
> Is that right?
>
> **Another typo:** On lines 346, 809, 1035, 1288, and 1295 (both lines near
> 1295) you have "transient of ICL", did you mean "transience of" or "transient
> nature of" in each case?

---

> ### Comment · Reviewer_8YMd · 2024-11-26
> **Remaining concerns (part 1 of 2)**
>
> Thank you for your response and revisions. You have allayed some of my
> concerns (W3, W5, W6) but unfortunately I maintain my concerns regarding
> the other issues (W1, W2, W4). The revisions have come some way to indirectly
> addressing these concerns but ultimately I am still unsure that the framing
> of the paper in the introduction is accurate and I would like to discuss
> further to see if we can resolve this.
>
> In your introduction, you mention that through your generalisation and regret
> bound analysis in your simple model:
>
> * you "identify conditions where in-context and in-weight learning emerge,"
> * these conditions are "consistent with" the empirical findings of Chan et
>   al. (2022a), and
> * the conditions "partially explain" transience (I note the revision from
>   saying they "explain" transience).
>
> In my understanding, this is the central case for the significance of the
> theoretical contribution of the paper (with the empirical contributions
> serving to support the theoretical contribution). Accordingly, the standard I
> think is appropriate for evaluating this paper is, to what extent are these
> claims substantiated soundly and explicitly in the main text?
>
> My remaining concerns center around whether or not these claims are in fact
> substantiated by your theoretical analysis. I have looked at the answers in
> the top-level comment and the new sections in the paper, but my core
> technical concerns have not been directly addressed.
>
> **First, let me check my understanding:** What are the "conditions" you
> identify, and how are they based on your model?
>
> 1. I think that when you refer to the conditions your analysis identifies,
>    you refer to the passage you clarified on lines 228--232:
>
>    > Under this algorithm, it is possible that initially IWL performs worse
>    > than ICL due to insufficient in-weight data. For example, when the
>    > training data contains a large number of rare classes, and each class is
>    > seen only a few times. Eventually, with enough in-weight data, the model
>    > can memorize the solution to achieve as good or even better prediction
>    > accuracy by using IWL on the query, rather than using the context for
>    > prediction and exhibiting ICL.
>
> 2. By "this algorithm" you refer to the optimal IWL--ICL mixture model, whose
>    predictions are based on ICL or IWL depending on which has lower expected
>    risk.
>
>    Therefore, if I understand correctly, this (the expected risk comparison)
>    is the "condition" you have identified. Is that right?
>
> 3. Then you interpet this condition: "it is possible that initially IWL
>    performs worse than ICL due to insufficient in-weight data."
>
>    If I understand correctly, you consider this claim supported by your
>    theoretical framework based on the fact that the IWL expected risk upper
>    bound is initially higher than the ICL risk upper bound. Right?
>
> 4. Then you continue to interpret, "with enough in-weight data, the model can
>    memorize the solution to achieve as good or even better prediction
>    accuracy by using IWL ... rather than ... ICL."
>
>    If I understand correctly, here you are referring to the fact that with
>    enough examples, the IWL risk upper bound falls below (or as low as) the
>    ICL risk lower bounds (for some $k$). Is that right?
>
> Assuming this is right (please clarify if you would put it differently), let
> me restate my concerns.
>
> **W1. Mismatch with "transience" phenomenon from Singh et al.:**
> You say this story "is consistent with the empirical observation that ICL ...
> can be transient" then cite  Singh et al. (2023) and point to the new
> Appendix D.
>
> I agree that your theoretical framework points to conditions under which IWL
> could arise during training (when the IWL lower bound goes under the ICL
> upper bound). One could call this "transience."
> However, I disagree that this is the "transience" studied by Singh et al.
> (2023).
> In my reading, the kind of "transience" studied in Singh et al. (2023) is
> specific to the setting where IWL has *no* risk advantage over IWL.
>
> I still don't think your theory matches up with this setting. As far as I can
> tell, your model makes no prediction about what a transformer should prefer
> in the case where IWL expected risk = ICL expected risk.
>
> Therefore, I am not sure in what sense your framework either offers a
> "partial explanation" or is even "consistent" with the findings of Singh et
> al. (2023), since it refers to a different setting.

---

> ### Comment · Reviewer_8YMd · 2024-11-26
> **Remaining concerns (part 2 of 2)**
>
> Notes:
>
> * I did see appendix D. You discuss two experiments in Singh et al.'s setting
>   where IWL and ICL have asymptotically equivalent risk. In the Omniglot
>   experiment, you say that you replicate the finding of Singh et al.. In the
>   synthetic experiment using your model, you say ICL does not turn out to be
>   transient.
>   To me, this appears to serve only to undermine the claim that your model
>   has bearing on Singh et al.'s type of transience (because it doesn't arise
>   in the closest variant of your model in an empirical experiment, though I
>   am not holding this against you).
> * As I said I still find your model informative. I think if you want to point
>   to prior work on a phenomenon like transience that your model is consistent
>   with, consider [Panwar et al. (2023)](https://arxiv.org/abs/2306.04891),
>   who study under the name "forgetting" a phenomenon like "transience" driven
>   instead by a difference in risk.
> * A minor point, the Appendix D hyperlink does not take me all the way to
>   appendix D. (Possibly this is due to the presence of figures, perhaps using
>   `\clearall` before the start of appendix D would fix this, not sure...)
>
>
> **W2: Incomplete characterisation of emergence for rare classes:**
> Now about the other interpretation of your analysis, in the rare classes
> regime. Here you say that in this setting, IWL performs worse than ICL due to
> insufficient in-weight data. As I noted, I understand this to be based on a
> comparison between the upper bound on the IWL expected risk and the upper
> bound on the ICL example risk (the IWL upper bound is larger for small
> $N_x$).
>
> My concern is that, of course, a larger upper bounds does not, in general,
> imply a larger value. You need to see a lower bound higher than an upper
> bound to infer that the former's value is higher than the latter's. Or, it
> would be sufficient to assume that the upper bound is actually tight.
>
> In your reply you mention that "the bounds properly describe the typical
> trends of the error of the predictors." What is this assertion based on?
>
> **W4. Specific model of in-context learning appears quite restrictive:**
> Finally, I'm still concerned about the concrete softmax-similarity
> induction-head ICL predictor as being too restrictive.
>
> You pointed out that your mixture model works for a black-box ICL predictor.
> I understand this but it does not address my concern, because your story for
> the ICL vs. IWL conditions of emergence depends on your risk bounds for the
> specific model, which are not shown to hold for general black box predictors.
>
> I understand that the softmax-similarity induction-head ICL predictor model
> is a small change from induction heads to allow approximate matches with
> prior tokens. However, I am concerned about this change because this change
> is load-bearing for the risk analysis. It is "load-bearing" in the sense that
> the risk lower bound (which is crucial for the paper's claims because it's
> what you compare to the IWL risk bound) specifically comes from considering
> the contributions to the output from all irrelevant pairs in the context.
> Yet, this lower bound is sensitive to the specifics of the ICL model. An
> exact-match induction head that ignored prior labels unless the inputs were
> equal to the prompt would have a different lower bound, for example. More
> generally, any larger family of ICL models (containing this family as a
> subset) would have a smaller or equal risk lower bound.
>
> I think it's fine to analyse some arbitrary model of ICL that permits
> analysis, and this model seems vaguely justified, but the choice is not that
> compelling to me and since the conclusions seem to depend on the choice, I
> would have liked to see the choice acknowledged as such so as to make it
> clear to readers that this is just 'one' model of ICL (and if your
> transformer performs ICL using a different mechanism, maybe this risk
> analysis won't apply).
>
> **Summary:** I've tried to lay out my understanding of the paper's claims and
> contents, and how this has given rise to my concerns, in detail. While my
> understanding is based on my best effort given available time I acknowledge
> that I might have misunderstood something about the argument or the paper's
> claims. I would consider raising my score if you can show me this is the
> case. Alternatively, if my understanding of the analysis is true, I would
> consider raising my score if you can adjust the introduction to make the
> claims match the conclusions of the analysis. I welcome further discussion.

---

> ### Author Response · Authors · 2024-11-27
> **Response to Remaining Questions**
>
> We greatly appreciate reviewer 8YMd for their thorough review and on-going discussions. We will first address the follow-up questions and later address the weaknesses tomorrow AoE.
>
> > Q1
>
> You are right that this would happen in the tabular case with no noise. However, if noise is present, the number of times a given input/context is seen matters. On the other hand, as we mentioned before, we chose the tabular setting only for convenience and our results are applicable more generally – given your concerns it seems to make sense to write out this generalization more properly. We discussed these generalizations in our response to W3 above. In short, the results of Section 2.3 are applicable much more generally (without essentially any restrictions), but are only useful if the regret terms in the upper bound can be optimized meaningfully. A sufficient assumption is that $\alpha$ is parametrized with a finite number of parameters such that the loss function is convex in these parameters and the resulting function class is rich enough to partition the input space into two sets such that IC prediction is better on one set and IW is better on the other. Note, however, that although proper regret bounds are not possible, regret-minimization algorithms (i.e., usual gradient-based optimization algorithms) perform well in practice even in the non-convex setting, so the bound of Proposition 3 based on regrets is still meaningful in such cases.
>
> > Q3
>
> You are correct that the risk of IWL is an expectation over $y$ given $x$, the query, since the output of the IWL predictor only depends on the query. However, the output of the ICL learner depends on the entire sequence $\tilde{x}$, and for some distributions of $\tilde{x}$ (depending on $k$), the ICL predictor has a higher expected risk.
>
> More specifically, we have
>
>
> $\mathbb{E}_{(\tilde{x}, y) \sim D} \left[ \ell(g(\tilde{x}), y) \right]$
>
> $= \mathbb{E}_{(\tilde{x}, y) \sim D} \left[ \ell(g(x), y) \right]$
>
> $ = \mathbb{E}_{(x, y) \sim D’} \left[ \ell(g(x), y) \right]$,
>
> where $D’$ is the marginal density of $(x, y)$. Our IWL risk upper bound holds pointwise for each $x$. For the ICL predictor, the risk is $\mathbb{E}_{(\tilde{x}, y) \sim D}\ell(h(\tilde{x}), y))$, and our bounds for ICL holds pointwise for each $\tilde{x}$ and $y$. Suppose $D$ is only supported on sequences with at least 1 irrelevant label, then the ICL predictor would have a minimum error that can potentially be higher than that of the IWL predictor. Note that under our data generating process, $y$ is sampled iid given the query $x$.
>
>
> > Q4
>
> It is a stylistic choice to include this explicitly in the statement, and now we have also added it to  Proposition 2; thanks for spotting this.
>
> > Q5
>
> To avoid misunderstanding, we have added a footnote about the base of the logarithm on page 4.
>
> > Q6
>
> Exactly. Of course, the final point where the IW upper bound falls below the IC lower bound depends on the actual value of the constant, but all we wanted to show in the figure was that this happens eventually. Please let us know if you think more explanation is needed about this in the text.
>
> > Typos
>
> Thank you, hopefully we have corrected all occurrences now.

---

> > ### Comment · Reviewer_8YMd · 2024-11-27
> > **Thanks for answers**
> >
> > Thanks, I am satisfied with these answers to my follow-up questions. I agree it would be helpful to include details on the generalization to the non-tabular case in an appendix or etc. Looking forward to continuing to discuss the remaining concerns.

---

> ### Author Response · Authors · 2024-11-28
> **Response to weaknesses**
>
> We would really like to thank reviewer 8YMd again for the in-depth discussion and suggestions.
>
> > W1
>
> Thanks for keeping this interesting question on the agenda. We think that we finally consolidated our findings properly, including a satisfactory explanation for the contradictory experiments: Our model predicts the transience of ICL whenever at some point IW prediction becomes better than IC prediction. This is easy to see to happen in cases where IWL is asymptotically better than ICL. However, when the two methods can both achieve the same asymptotic performance, it is hard to see which one can have any advantage in the large but finite sample regime. In fact, while Singh et al. (2023) shows an example where ICL is transient, in Figure 21 (left) we show an example where it is persistent and another where it is decreased over time but remains visible (this is similar to the experimental results of Singh et al.). Now we also provide some speculations for this case, which, together with our theory, correctly predicts the transience of ICL. We clarified these in the introduction (lines 64-68, and lines 71-74), at the end of Section 2.2, and in lines 1758-1792 of Appendix D.
>
> > W2
>
> To be more concrete, we have replaced Proposition 1 with a special bound for the cross-entropy loss, for which one can obtain faster rates than for general convex functions. The new rate, which comes with a lower bound that even matches the leading constant, is $\Theta(\log(N_x)/N_x)$. We have also changed Figure 1 to reflect this convergence rate. Of course, these rates are not directly applicable to training a complex function class like transformers, but they are suggestive and align qualitatively with some of the experimental findings.
>
> > W4
>
> While our ICL class $\mathcal{H}$ (used in the theory section) is indeed specific, it can be used to illustrate and analyze the potential difficulties with in-context learning:  In general, the difficulty of learning $h$ and the quality of the solution depends on the hardness of the in-context learning problem. Our simple ICL function class makes the “prefix-matching” aspect (Olsson et al., 2022) of the induction head explicit, and hence allows to quantify the difficulty of ICL in terms of the number of irrelevant contexts (one of many possible measures to quantify hardness). While this is explicit in our analysis (lower bounds) for $\mathcal{H}$ due to our simple structure, the same difficulty is also reflected in our experiments, where we vary the number of relevant contexts and also the context length. We can see that irrelevant labels indeed present a challenge for ICL as we show in our transformer experiments: when we have only 1 relevant label out of 4 labels, the transformer does not learn to do ICL at all (Figure 4, first column). In general with fewer relevant contexts or more irrelevant contexts, it is more difficult for ICL to emerge (see also Figures 11 and 12). Chan et al. (2022) and Singh et al. (2023) also made similar observations, and had to repeat relevant data in the context several times (“bursty samples”) to achieve in-context learning. Thus, from this perspective, we think that the function class we consider to demonstrate limitations is both simple and meaningful. Note that we have also added in lines 190-192 that our ICL predictor is only an example of a mechanism under which ICL can occur.
>
> **Suggestions**
>
> > W3
>
> Thank you for your suggestion. We have included them in our appendices A.1.
>
> > W5
>
> We added “qualitatively” to line 350 to strengthen this. We believe we have been clear about this---we don't claim that the theory would actually properly match practice.
>
> > W6
>
> We added the first two points and, for the third one, we have emphasized that we use a few examples to demonstrate that memorization can affect IW vs IC learning. Regarding comparison to Wei et al. (2023), it seems that for similar-sized models (the smaller GPT models) their experiment is quite inconclusive (Figure 2 in their paper) as even for no label flipping the prediction accuracy is around 50%, so it is not clear what kind of comparison could be made. We added a paragraph explaining this to the end of Appendix C.1.

---

> > ### Comment · Reviewer_8YMd · 2024-11-29
> > **Latest revisions/answers resolved my remaining concerns and I am raising my score to 8**
> >
> > Thank you for your further revisions and answers. The latest round of revisions has resolved my remaining concerns, with the below notes. I increased my score for 'soundness' since the claims and the theory now line up in my opinion, and I increased my rating from 5 to 8.
> >
> > Notes:
> >
> > * **W1:** I am satisfied with your resolution of the relationship to Singh et al. and the discussion in appendix D.
> > * **W2:** Nice, that the specialisation to cross entropy allows you to give a tight bound addresses my concern (I wasn't familiar with the Krichevsky–Trofimov estimator). Should the blue lines in figure 1 now be replaced with a range of $\pm K/N_x$ where $K$ is an arbitrary big Oh constant (if not already)?
> > * **W3:** Looks great.
> > * **W4:** I am satisfied with the revision to lines 190-192. I have come to see as separate theoretical contributions (1) that your general "mixture of two algorithms" model will prefer to route inputs to the sub-model that achieves the lowest risk on those inputs, and (2) the specific instantiation of the ICL sub-model as just an illustrative example, that does line up with your empirical observations. Under this framing I think it's adequate.
> > * **W5:** Sorry, I think I missed that your first revision already added what is now line 373: "These transformers only act as proxies for measuring whether it is possible to perform ICL and IWL in the idealized case." This is sufficient to address my remaining concerns about being explicit about this gap.
> > * **W6:** Looks great.
> >
> > Thank you again for responding thoughtfully and in detail to each of my concerns. I am pleased to now be able to recommend the paper for acceptance.

---

> > > ### Author Response · Authors · 2024-11-29
> > > **Response on the notes**
> > >
> > > Thank you very much for your constructive criticism and the effort put into reviewing our paper, which really helped improve the manuscript.
> > >
> > > Thank you for the extra notes---for W2 we note that the leading term is $\frac{\log N}{N}$ (with matching leading constants $\frac{C-1}{2}$ for both the upper bound and lower bound), and the $O(1/N)$ is a lower-order term. As a result we choose $K = 0$ here to illustrate only the leading term which is tight.

---

### Official Review · Reviewer_ueJX · 2024-11-04

**Soundness:** 3
**Presentation:** 2
**Contribution:** 3
**Rating:** 6
**Confidence:** 3

**Summary:**

[update: in response to the revision and author responses I have updated my score]

The study of in-context learning (ICL) has emerged in recent years as a tractable and yet important piece of the puzzle for understanding how transformers are able to achieve such good performance across such a wide range of data distributions. There is now a growing literature combining theoretical models with empirical study of ICL in synthetic settings, with the aim of understanding the mechanism by which ICL works, and why it emerges in the first place.

The present paper adds to this literature, by studying the tradeoff between memorisation (or in-weight learning) and ICL in a simple theoretical model and associated experiments, and how this tradeoff varies with the nature of the data distribution.

**Strengths:**

- Reasonable theoretical foundation for their arguments
- Well-designed experiments to test these theoretical predictions

Overall I think this has the makings of a good paper that makes a significant and original contribution to an important thread of research in the science of transformer generalization.

**Weaknesses:**

- At the moment I am not convinced that the experiments reported in Fig 2 really do support the theoretical predictions. I hope this is just my misunderstanding and that the authors can explain what I missed, and perhaps provide a clearer explanation in the paper.
- While I am pretty well-versed in this corner of the literature and so prepared to care about the contents of Fig 2, it still took me a while to process all the acronyms and setup.

**Questions:**

The main theoretical results seems to be that “ICL will first emerge on rare classes, and will eventually be replaced by the IW predictor on those classes” (line 207,208). This prediction is studied experimentally in the setting described in Figure 2, where three models are trained for 100K gradient steps (Appendix B.1.1) across a number of sample sizes N. The three models are all trained with a single (x,y) pair in the context (L = 1) and for the IW predictor the context is always “relevant” (so if I understand correctly the correct output given (x,y), x’ is to simply ignore x’ and copy the input y to the output) and for the IC predictor the context is always “irrelevant” (so given (x,y), x’ the correct output cannot be learned from the context, which should be ignored). The main transformer is trained to have 90% of its contexts relevant.

My reading of Fig 2 is that on the rare classes where IC is possible (second column, top row) the transformer behaves very similarly to the IC predictor from the beginning, and only converges to the IW predictor when the IC predictor does. The behavior on high-frequency classes seems qualitatively similar, and so I don’t see on what basis this is evidence for the “first emerge” part of the claim. Nor do I see how these experiments really support the idea that for large N the transformer switches to the IW predictor. A simpler explanation for this experimental data seems to be that, by construction, the transformer should be somewhere between the IC and IW predictor since its training distribution is between theirs, and for large N the IW and IC predictor behave similarly on relevant contexts.

It seems therefore that the crux of the argument might lie in the OOBD plots (Fig 2 second row). My reading here is that on the low-frequency classes OOBD performance of the general transformer is initially bad, improves, and then degrades, whereas for high-frequency classes the OOBD performance is initially roughly as good as it ever gets on the low-frequency classes, and then degrades. I do not see here the evidence for “ICL emerges first on rare classes”? The degradation of OOBD performance also seems to be much slower on the low-frequency classes than the high-frequency classes, which makes me unsure if the OOBD data shows any evidence for the “replaced by the IW predictor on those classes” claim either.

In short, I’m not sure how to read Fig 2 as evidence for the theoretical claims. This leaves me unsure of the soundness of the main experimental contribution of the paper.

Minor issues:

- “Cdot” line 149
- “Transient” -> “transience” line 303
- It’s a bit strange to see H defined twice in quick succession (differently) in 130 and 158

Factors that would improve my score:

- As already mentioned, clarifying the link between Fig 2 and the theoretical prediction.
- Compared to the other literature in this area the models are rather small (two layers, one head per layer) and the contexts are very short (e.g. L = 1 in Fig 2). Overall I am left with the sense that maybe these models are just not capable of learning the ICL solution the theory is talking about, and this could be confounding some of the other aspects of the paper. This is actually how I read Fig 6. To what extent is the choice of L = 1 to illustrate something theoretically, and to what extent is it just “the only length that works” as a result of the small model?

---

> ### Author Response · Authors · 2024-11-22
>
> We thank reviewer ueJX for their thoughtful reviews. We address your questions and comments here.
>
> > Descriptions of IC and IW predictors
>
> To avoid any misunderstanding, we would like to point out that there is a typo in the reviewer’s interpretation of the experimental setting in the “Question”, where the descriptions of the IW and IC predictors should be swapped.
>
> > On Figure 2 and the reviewer’s criticisms that : (i) ICL emerges first on rare classes first; (ii) for large data size the transformer switches to IW learning.
>
> Indeed, the OOBD plots (bottom row) in Figure 2 show the model’s ability to perform in-context learning: since the mapping from the inputs to the labels is changed (as described in lines 353-360), the only way to achieve good performance in this case is to use in-context learning where possible (i.e., when the context is relevant). As such, whenever we see high error rates for relevant contexts, it means that the model is not doing IC prediction. This supports our claim for (ii). In particular, the fact that the OOBD error of the transformer for relevant contexts (both for high- and low-frequency classes) becomes 1 for large training set sizes shows that the model is not able to do IC prediction, and the (close to) 0 IBD error (for all cases) shows that the model is able to do proper prediction for the base distribution, which – by exclusion – has to be IW prediction.
>
> Regarding (i), we indeed did not make our claim precise in lines 208-210; and now have rephrased. We refer the reviewer to the general response for “Regarding the claim about ‘ICL emerging first in rare classes’”. Supporting our revised statements, on the two leftmost columns of the bottom row of Figure 2 (OOBD, Rel. Cont., $C_H$ and $C_L$), the transformer has a significantly lower error than random guessing at the beginning, showing the presence of IC prediction. As $N$ increases, we start to see that the transformer begins to degrade in OOBD performance for relevant contexts, eventually losing this IC-prediction ability for both high- and low-frequency classes, while achieving close to zero error for in-distribution data, showing the emergence of IW prediction.
>
> > Experiments with larger context lengths
>
> We have conducted experiments with $L = 4$ while varying the number of relevant contexts (Figure 4 - please note that we have reorganized the figures and the references correspond to the revised version). Here increasing the number of relevant contexts, similar to burstiness in Chan et al. (2022) and Singh et al. (2023), can encourage the emergence of ICL as suggested by our theoretical findings as well. In the Omniglot experiments in Section 3.2 we have conducted experiments with $L = 2$ (Figure 5). Generally we see that the transformer exhibits some ICL capabilities on rare classes initially, which then disappears as $N$ increases. This corroborates the findings from Section 3.1 with the synthetic dataset using $L = 1$. Please also note that the original Figure 6 corresponded to cases where the relevant information appears only once in the context, which becomes very challenging for ICL even for relatively small context lengths (this has been observed in earlier work and is the reason why the relevant context is repeated several times in the experiments of, e.g., Chan et al., 2022a, and Singh et al., 2023).
>
> > Regarding using more transformer blocks and attention heads.
>
> While we believe that using larger models may further demonstrate how the theoretical results align with scale, Singh et al. (2024) has demonstrated that induction heads can indeed be implemented with only two transformer blocks with a single head per layer. We further note that Singh et al. (2024) uses Omniglot to validate their results as well. Consequently, we do not believe that there is any confounder induced by the (potential) lack of model expressiveness.

---

> > ### Comment · Reviewer_ueJX · 2024-11-25
> >
> > Thanks for the detailed response and for the updated paper.
> >
> > I think the reorganisation of the figures has helped and I'm grateful for the detailed explanation offered here in connection with my original questions. In line with my earlier comments, I have increased my score.

---

### Official Review · Reviewer_gQxH · 2024-11-05

**Soundness:** 2
**Presentation:** 3
**Contribution:** 2
**Rating:** 6
**Confidence:** 2

**Summary:**

This paper provides a theotical framework which studies in-context learning's emgerence and transient nature through a simple model which can adapatively switching between an in-weight predictor and an in-context predictor. The key insights from this paper shows that when the training data has long-tail, containg rare classes that appear infrequently, ICL tends to emerge. And if there are enough training data, IWL tends to dominate. Their experiments on both synthetic data and Ominiglot daat shows that ICL exhibit for rare classes and IWL for common ones.

**Strengths:**

1. The paper introduces a novel theoretical framework that investigates how a model selects between ICL and IWL. This framework offers insights into the conditions under which ICL emerges, especially highlighting the role of rare labels in promoting ICL.
2. The writing and presentation is clear.
3. this paper includes experiments in both synthetic and natural few-shot learning dataset and validated their findings.

**Weaknesses:**

1. This paper assumes a simple switch between the IWL and ICL based on error minimization but in real world LLM the transition may not be instantaneous and involves more complexities.
2. The in-context predictor is modeled as a convex combination of the labels in the context, weighted by input similarity. This raises questions because more commonly the in-context predictor should learn the relationship between inputs and outputs (i.e., the mapping from x to y rather than combining labels. This simplification might limit the applicability of the theoretical findings to real-world scenarios where the input-output relations are more complex.
3. The experiment with the real LLM, Gemini Nano 1 seems somewhat limited. It primarily tests the LLM's ability to follow in-context labels versus trained labels, which may be influenced by how different LLMs are instructed to trust new labels versus their trained knowledge. A more comprehensive and persuasive experiments could include various LLMs to assess the generality of the findings and provide deeper insights across different architectures and scales.

**Questions:**

A recent work[1] shows that ICL exhibits dual operating modes: task retrieval and task learning. As the number of correctly labeled examples grows, the model transitions from task retrieval to task learning, but initially, even with correct labels, it may retrieve the wrong task, leading to errors. Wondering how does your finding explain this dual-mode behavior?

[1]Dual Operating Modes of In-Context Learning. https://arxiv.org/abs/2402.18819

---

> ### Author Response · Authors · 2024-11-22
>
> We thank reviewer gQxH for their insightful feedback. Here we address your comments and questions.
>
> > W1
>
> Though real-world LLMs are more complex, we focus on the IWL vs. ICL aspect of in-context learning, as studied by Chan et al. (2022a), Singh et al. (2023), and the concurrent paper of Tong at el. (2024).  We propose a stylized theoretical model that is amenable to analysis; given the theoretical model, we describe a possible mechanism for the emergence of ICL vs. IWL, and support our hypothesis with experiments.
>
> Tong and Pehlevan. MLPs Learn In-Context. arXiv preprint arXiv:2405.15618 (2024).
>
> > W2
>
> We refer the reviewer to the general response regarding “Criticism that the stylized model/data distribution is too simplistic in the theory”.
>
> > W3
>
> Please see the section “Significance of the LLM experiments” in the general response.
>
> > Q1
>
> Thank you for bringing this paper to our attention. It considers a related problem: while we consider the difference of ICL and IWL, this paper considers different variants of IWL predictors, and examines the problem that different IWL predictors (skills) can be selected by the model given the input. The difference in our approaches is that while we model the behavior of an LLM with general predictors and a combiner, Lin and Lee (2024) considers the model of optimal Bayesian prediction in linear regression with the squared loss over Gaussian mixtures.  While they explain the potentially incorrect in-weight learning with task-similarity, we are mostly interested in understanding how the training process can affect where the model will apply in-context versus in-weight prediction, which is important in designing specific training methods to enable balancing factuality and grounding to the prompt.
>
> Lin and Lee. Dual Operating Modes of In-Context Learning. ICML 2024. https://arxiv.org/abs/2402.18819

---

> > ### Comment · Reviewer_gQxH · 2024-11-26
> >
> > thank you for replying to my concerns, it resolved most of my concerns and question. I raised my score to reflect this.

---

### Author Response · Authors · 2024-11-22
**General Response 1/3**

We would like to thank the reviewers for their thought-provoking and detailed reviews. Here we address common comments raised by the reviewers. All updates in the manuscripts are presented in blue text.

> Regarding the claim about “ICL emerging first in rare classes” (reviewers ueJX and 8YMd)

We believe the original statement on lines 208-210 is ambiguous and that might have caused some confusion. We have revised our statement to be “Under this algorithm, it is possible that initially IWL performs worse than ICL due to insufficient in-weight data. For example, when the training data contains a large number of rare classes, and each class is seen only a few times. Eventually, with enough in-weight data, the model can memorize the solution to achieve as good or even better prediction accuracy by using IWL on the query, rather than using the context for prediction and exhibiting ICL.” See lines 228-232.

> The theoretically optimal performance of IC and IW predictors, and their selection

We have included the following discussions in Appendix D regarding selecting between IC and IW predictors based on their theoretical performances and experimental results on transience of ICL.

**Mixed in-context and in-weight data:** First note that the ICL and IWL predictors in the experiments are just proxies and do not necessarily have the same abilities as (components of) the transformer model. Nevertheless, in the cases we consider, the theoretical optimum performance (which is assumed to be achieved asymptotically) of the IWL predictor is better since we have samples where no relevant context is available. Nevertheless, for some inputs IWL and ICL can still have equally good performance. However, in this case, the model may still have some bias towards ICL if some smoothness is imposed on $\alpha$, as because of this the inputs for which IW prediction is better will bias the selection of $\alpha$ for nearby inputs.

Note that the standard pretraining procedure of LLMs includes both in-context and in-weight data due to the sequential nature of the training, because at the beginning of any prompt or document there is no useful context (and there are typically many documents containing relevant contexts).

**Only in-context data, deterministic labels:** Considering the “fair” setting where there is always relevant context, both IW and IC prediction can achieve the same performance in the noiseless setting. While Singh et al. (2023) shows experiments that even in this case data repetition leads to IWL, our experiments, for example the ones reported in Figure 4, suggest that this may depend on the difficulty of the ICL and IWL tasks. This can be seen by analyzing the performance of our ICL predictor (i.e., a predictor trained only with in-context data), as depicted in Figure 4: in these experiments the context length is always 4 and the tasks become easier as more examples in the context become relevant. When only 1 example is relevant (first row), the model does not show any ICL capability, and hence learns fully IW. This gradually changes as the problem becomes easier, and when all elements in the context are relevant (last row), the model always performs IC prediction, and IW prediction does not occur in the scales shown. During the rebuttal period, we trained an IC model for a much longer period, both for the synthetic and Omniglot datasets. In the former $L = 1$ and in the latter $L = 2$. We can see in Figure 21 that ICL persists for both datasets after millions of training steps.

**Only in-context data, random labels:** For the case of random labels, simple calculations show that both IWL and ICL can ultimately be preferred depending on the correlation structure between the noise in the labels. For the case of fully correlated labels, when the possibly random label for the same example is identical between the context and the ground-truth response to the query, IC prediction can achieve 0 error while IW prediction will suffer because of the noise. If the randomness in the labels is independent, IW learning may have an advantage: Consider the case of binary prediction where the ground truth for a given input has a main label with probability $1-p$ and a ``flipped’ label with probability $p$ with $p<0.5$. If there is a single relevant context, the labels for the context and the query agree with probability $(1-p)^2+p^2$, leading to an ICL error probability of $2p(1-p)$, while the error of the IW predictor always predicting the main label is $p$, and we have $p<2p(1-p)$ for $p<0.5$.

---

> ### Author Response · Authors · 2024-11-22
> **General Response 2/3**
>
> > Criticism that the stylized model/data distribution is too simplistic in the theory (reviewers gQxH, 8YMd, and VrYi)
>
> Our main message is that ICL can emerge when it outperforms IWL on an input sequence, for example when the model does not have enough samples to learn in-weight. We use a simple model that can learn to weigh ICL vs. IWL. The ICL predictor in Section 2.2 is instantiated with the kernel predictor just for concreteness. However, our learning results in Section 2.3 do not assume a particular form of the ICL predictor and can be applied more generally (we revised the text introducing the specific form of $\mathcal{H}$ in line 187 accordingly) . In particular, the proposed algorithm can treat the ICL and IWL predictors as black boxes and can learn to use ICL vs. IWL depending on the regret of the predictors.
>
> Our particular ICL predictor is inspired by the induction-head mechanism where it chooses and copies the most similar token (Olsson et al., 2022), and this is one way to instantiate this. The ICL predictor is a natural generalization of the induction head: we do not assume exact match between the query and the examples, but instead weigh the labels by the similarity between the example and the query. Our combination of the ICL and IWL predictor closely follows the mechanistic working of a softmax attention model, and could be thought of as a more abstract model being applied at an arbitrary layer of a transformer, combining concepts rather than tokens.
>
> > Regarding the point that the IC and IW predictors in the experiments might not exhibit only ICL and IWL respectively (reviewers ueJX and 8YMd)
>
> There is no easy way (and it might actually be impossible) to separate the IW and IC predictor circuits of a transformer model, since they share significant architectural components. To estimate the respective performances, we trained two transformers, ICL and IWL in the paper, with only in-context and, respectively, in-weight data.  These transformers act as proxies for measuring whether it is possible to perform ICL and IWL in the idealized case.
>
> To further bridge the gap between the theory and experiments, we have added new experiments in Appendix B.2 that follow the stylized model (Figures 13 and 14). To further generalize to other model architectures we have also trained RNN-based models (Figure 15) on the same synthetic datasets, which shows similar trends as the results with the transformer architecture.
>
> > Clarification of the experiment figures
>
> In the revised version we improved the presentation of the experiments and their conclusions.  Specifically we have updated the axes and labels to be consistent across all figures. To reduce clutter we have also reorganized some results and moved the less informative results to Appendix B.

---

> > ### Author Response · Authors · 2024-11-22
> > **General Response 3/3**
> >
> > > Significance of the LLM experiments (reviewers gQxH, 8YMd, and VrYi)
> >
> > The main message of the paper is that memorization affects the emergence of in-context learning: if the model is able to predict a response well without context, it will tend to select a memorized “in-weight” response, and will revert to combining knowledge from the context otherwise. Training a reasonably good quality language model from scratch is very resource consuming, and to demonstrate our point, we need to ensure we have two different versions of an LLM to compare: one where some selected training data is learned several times and one where such extreme memorization has not taken place. The cheapest way to make the model memorize the response to a specific input is to memorize this information during fine-tuning, which is exactly what we did in our experiments.  Our experimental findings demonstrate that, for inputs with strong memorization, the ability to predict from the context is reduced, while for other inputs the ability to utilize the context in the prediction remains intact. The experiments become even more resource-heavy as we take larger models, so for practical considerations we had to use one of the smaller available models. (To avoid any confusion, please note that at query time no additional prompt or instruction is involved in the queries – of course the training of the model can affect how much the model prefers to use the context, but our experiments shows that additional memorization can steer this toward in-weight prediction.)
> >
> > As described in the paper, in our LLM finetuning experiments we only used very few examples to demonstrate that stronger in-weight knowledge of some facts affects the ICL ability of the model. Analyzing the logits we could see that this was the case all the time, that is, the probability of the memorized response has always increased. Although this limited experiment admittedly does not allow one to draw strong conclusions about statistical significance, that was not the goal because the model is expected to behave very differently for different queries given how much information about the names or cities were provided during training, giving uncontrolled biases we cannot account for. The reason for adding invented names and city names was to avoid some of these biases, but we can see that even memorizing these is harder because of the already existing biases in the model for using real city names.
> >
> > **Connection between theory and LLM experiments:** Analogous to the synthetic experiments, the LLM experiments investigate whether more in-weight data will induce IWL and reduce ICL on inputs with in-weight data. In particular, the LLM is fine-tuned with sentences that contain arbitrary (name, city) pairs, mimicking training the LLM on this in-weight dataset. Our theoretical results suggest that after seeing this dataset, the LLM will exhibit more IWL on the inputs (names) seen in the dataset and associate them with their respective cities. At evaluation time, the LLM is given inputs that contain a name and a different city from the one in the in-weight dataset. Our experiments show that instead of answering with the city in the context, the LLM will sometimes give the answer in the in-weight dataset, demonstrating clear IWL instead of ICL on those inputs.

---

### Comment · Area_Chair_Md8F · 2024-11-24

Dear Reviewers,

This is a gentle reminder that the authors have submitted their rebuttal, and the discussion period will conclude on November 26th AoE. To ensure a constructive and meaningful discussion, we kindly ask that you review the rebuttal as soon as possible and verify if your questions and comments have been adequately addressed.

We greatly appreciate your time, effort, and thoughtful contributions to this process.

Best regards,
AC

---

### Meta-Review · Area_Chair_Md8F · 2024-12-29

**Metareview:**

This paper investigates how transformers balance between in-context learning (ICL) and in-weight learning (IWL) capabilities. The key findings are that ICL emerges when IWL/pretraining has insufficient data, particularly for rare classes, and that ICL can become transient as more training data enables better IWL performance, at which point the LLM will ignore its context. The paper develops theoretical analysis through a simplified model and validates the findings through experiments on synthetic data, Omniglot, and a real LLM. The reviewers appreciated the theoretical framework and controlled experimental validation. The weaknesses were limited applicability of the theory and experiments to real-world LLMs due to certain simplifying assumptions (eg, mixture model). Overall, the reviewers were in agreement to accept the paper.

**Additional Comments On Reviewer Discussion:**

See above.

---

### Decision · Program_Chairs · 2025-01-22

Accept (Poster)